# ATAD2 drives melanoma growth and progression and inhibits ferroptosis

Ashok Mari [1,3], Kevin Graciano [1,2,3], Raj Kumar[1], Emily Giles [1], Patrick T Ball [1],
Revu V L Narayana [1] & Romi Gupta [1,2] ✉

## Abstract

**Melanoma is a highly metastatic form of skin cancer for which current therapies offer limited benefits. We show here that the histone reader ATAD2 is overexpressed in melanoma and predicts poor prognosis, and that the MAP kinase pathway, via the transcription factor E2F1, stimulates ATAD2 expression. Genetic or pharmacological inhibition of ATAD2 suppresses the growth and metastasis of BRAF and NRAS mutant melanoma. Mechanistically, we show that ATAD2 inhibition activates both distinct and common tumor-suppressive pathways in BRAF and NRAS mutant melanoma. In particular, we find that ATAD2 inhibition induces ferroptosis in both contexts by downregulating the ferroptosis suppressor GPX4. The ferroptosis inducer erastin also inhibits melanoma growth. Combining the ATAD2 inhibitor BAY-850 with the MEK inhibitor trametinib potently suppresses melanoma growth. Our study identifies ATAD2 as a key driver of melanoma and provides a rationale for targeting ATAD2 in conjunction with the MAPK pathway to treat melanoma.**

**Keywords** ATAD2; Melanoma; MAPK; NRAS; BRAF
**Subject Categories** Autophagy & Cell Death; Cancer

## Introduction

Melanoma is a highly aggressive form of skin cancer that accounts for over 75% of all skin cancer-related deaths (Davis et al, 2019). BRAF and NRAS mutant melanoma account for the majority of melanoma cases (Ghosh and Chin, 2009). In particular, mutations in the BRAF gene are detected in over 50–60% of melanomas, whereas mutations in the NRAS gene occur in over 20% of cases (Ghosh and Chin, 2009). These mutations result in the activation of the MAP kinase pathway, which promotes the proliferation and survival of melanoma cells (Schadendorf et al, 2015). Several inhibitors of mutant BRAF are used in the clinic in combination with MEK inhibitors for treating BRAF mutant melanomas (Gonzalez-Cao et al, 2021). For NRAS mutant melanoma,

treatment with MEK inhibitors along with other targeted agents or immunotherapy are used (Johnson and Puzanov, 2015). Although the current treatment regimen is expected to change with the development of new pan-RAS inhibitors that have shown clinical activity against RAS mutant cancers (Guruvaiah and Gupta, 2024; Punekar et al, 2022; Vidimar et al, 2020). However, even with these discoveries, melanoma remains a challenging disease to treat (Huang et al, 2024). Therefore, new and more effective therapies are needed to further improve treatment outcomes in the clinic.

In addition to genetic mutations, most cancers, including melanoma, undergo epigenetic alterations, such as changes in DNA methylation and histone modifications (Lee et al, 2014). These changes are largely driven by mutations or changes in the expression of DNA methylation or chromatin regulatory proteins (Yang et al, 2023). Histone reader proteins are crucial epigenetic regulators that recognize and bind to specific histone post-translational modifications (PTMs), thereby influencing chromatin structure and gene expression (Zheng et al, 2015). In melanoma, dysregulation of histone reader proteins has emerged as an important mechanism driving tumor progression and therapy resistance (Gallagher et al, 2015). For instance, the bromodomain and extraterminal (BET) family proteins, which bind acetylated histones, are often upregulated in melanoma and contribute to oncogene expression, including MYC (Wu et al, 2023). Targeting histone readers with small-molecule inhibitors has shown promise in preclinical studies (Huang et al, 2024), offering new potential therapeutic strategies for treating highly aggressive and drug-resistant melanoma.

ATPase family AAA domain-containing protein 2 (ATAD2) is a bromodomain-containing protein that plays a significant role in cancer by acting as a histone reader and transcriptional coactivator (Fu et al, 2023). It binds to acetylated histones, promoting the expression of genes involved in cell cycle regulation, proliferation, and metastasis (Fu et al, 2023). Overexpression of ATAD2 has been observed in various cancers, including breast, ovarian, lung, and liver cancers, and is often associated with poor prognosis and aggressive tumor behavior (Caron et al, 2010; Guruvaiah et al, 2023; Liu et al, 2019). ATAD2's ability to amplify oncogenic signaling pathways, such as MYC and E2F (Ciro et al, 2009), underscores its role as a critical driver of tumor progression. As a result, ATAD2 is emerging as a potential therapeutic target, with efforts underway to develop inhibitors to disrupt its cancer-promoting functions.

---

[1]Department of Biochemistry and Molecular Genetics, The University of Alabama at Birmingham, Birmingham, AL 35233, USA. [2]O'Neal Comprehensive Cancer Center, The University of Alabama at Birmingham, Birmingham, AL 35233, USA. [3]These authors contributed equally: Ashok Mari, Kevin Graciano. ✉E-mail: romigup@uab.edu

In this study, we identify ATAD2 as a driver of melanoma tumor growth and metastasis that is overexpressed in melanoma via the action of the MEK–ERK pathway in a transcription factor E2F1-dependent manner. We find that ATAD2 inhibition activates both distinct and common tumor-suppressive pathways in BRAF and NRAS mutant melanoma. In particular, we found that the inhibition of ATAD2 induces ferroptosis in both BRAF and NRAS mutant melanoma by inhibiting the expression of the ferroptosis suppressor GPX4. Furthermore, combinatorial targeting of melanoma by ATAD2 inhibitor BAY-850 with MEK inhibitor trametinib potently inhibits melanoma tumor growth. Collectively, these studies assign ATAD2 as a facilitator of melanoma growth and metastasis and showcase the promise of targeting ATAD2 either alone or in combination with MEK inhibitors for more effective melanoma therapy.

## Results

### ATAD2 is overexpressed in patient-derived melanoma samples, and its overexpression predicts poor prognosis in melanoma patients

Bromodomain proteins act as epigenetic readers by recognizing acetylated tails on histones, thereby promoting the transcription of specific target genes (Josling et al, 2012). In humans, there are around 60 identified bromodomains, categorized into eight subfamilies based on their structural similarities. Family IV of bromodomain-containing proteins comprises seven members: BRPF1, BRPF2, BRPF3, BRD7, BRD9, ATAD2, and ATAD2b (Lloyd and Glass, 2018). Despite the significant physiological roles of family IV bromodomains, their therapeutic potential remains largely unexplored in cancer, including melanoma. To determine if any of the Family IV bromodomain epigenetic readers are important in melanoma, we first examined if they are overexpressed at the mRNA level using GEPIA. Our analysis showed that out of seven members of Family IV of bromodomain-containing proteins, only ATAD2 mRNA expression was significantly higher in melanoma patient samples compared to normal skin in the TCGA dataset analyzed using GEPIA (Figs. 1A and EV1A) (Cancer Genome Atlas Research et al, 2013; Tang et al, 2017). Similar results were obtained in other melanoma dataset such as Talantov melanoma, further establishing that ATAD2 is overexpressed in melanoma patient samples compared to normal skin (Fig. 1B). ATAD2 expression was also observed to be higher in metastatic melanoma sample as compared to primary melanoma samples indicating its role in melanoma metastasis (Fig. 1B). To further determine if increased ATAD2 mRNA expression also correlates with increased protein level, we checked ATAD2 protein expression in melanoma samples and normal skin using the Human Protein Atlas. We observed that ATAD2 is strongly expressed in the majority of melanoma patient samples as compared to normal skin (Fig. EV1B,C). Based on these findings, we next investigated whether ATAD2 overexpression was clinically relevant and influenced patient survival. Analysis of the Human Protein Atlas dataset revealed that higher ATAD2 expression was associated with poor prognosis, with high ATAD2 levels associated with reduced patient survival in melanoma patients (Fig. 1C). Collectively, these results demonstrated that ATAD2 is overexpressed in melanoma samples, and its overexpression predicts poor prognosis in melanoma patients.

We next ascertained the mechanism by which ATAD2 expression is upregulated in melanoma. MAP kinase pathway activation due to mutations in the BRAF or NRAS genes occurs in over 75% of all melanomas (Arkenau et al, 2011). Therefore, we first investigated whether the MAP kinase pathway is involved in the upregulation of ATAD2 expression in melanoma. We treated BRAF mutant (A375 and M14) and NRAS mutant (SKMEL-103 and SKMEL-2) melanoma cell lines with the MEK inhibitor trametinib and measured the expression of ATAD2 mRNA and protein. We found that trametinib-treated melanoma cells showed downregulation of both *ATAD2* mRNA and protein (Fig. 1D,E). These results demonstrated that the MAP kinase pathway is necessary for ATAD2 overexpression in melanoma.

Since the observed effect of MAP kinase pathway inhibition on ATAD2 was at the mRNA level, we next investigated the transcription factors regulated by the MAP kinase pathway that may modulate ATAD2 expression in melanoma cells. To test this, we performed bioinformatic analyses by examining the DNA sequence of the ATAD2 promoter (~1 kb) to identify transcription factors (TFs) consensus DNA binding sites using the PROMO search (Messeguer et al, 2002). This analysis identified 24 TFs that based on PROMO search were predicted to bind to the ATAD2 promoter (Fig. EV1D). To prioritize TFs for further analysis, we analyzed the expression of these 24 TFs in melanoma TCGA data via the GEPIA. We asked whether any of these transcription factors similar to ATAD2, were also overexpressed in melanoma samples. We found that like ATAD2, only the transcription factor E2F1 was significantly overexpressed in melanoma samples (Fig. 1F). We also observed that E2F1 and ATAD2 showed a positive correlation in melanoma samples (Fig. EV1E). Based on these findings, we first asked if the transcription factor E2F1 is regulated by the MAP kinase pathway. We treated melanoma cells with trametinib and measured E2F1 expression. We found that like ATAD2, treatment with trametinib resulted in reduced E2F1 mRNA and protein levels in melanoma cells (Fig. 1G,H). To directly establish that E2F1 stimulates the expression of ATAD2 in melanoma, we knocked down the E2F1 expression using shRNA and measured ATAD2 expression. Melanoma cells expressing non-specific (NS) shRNA were used as controls. We found that *E2F1* knockdown led to reduced ATAD2 expression at both the mRNA and protein levels in melanoma cells (Fig. 1I,J). To further establish that ATAD2 is a direct target of E2F1, we performed a CUT&RUN assay. We observed significant enrichment of E2F1 on the ATAD2 promoter, which was reduced following *E2F1* knockdown (Fig. 1K). Collectively, these results demonstrated that the MAP kinase pathway via the transcription factor E2F1 stimulates the expression of ATAD2 in melanoma (Fig. 1L).

### ATAD2 inhibition suppresses melanoma growth and metastases in cell culture models of melanoma

Because we found that ATAD2 was overexpressed and predicted poor prognosis in melanoma patients, we asked if it also facilitates melanoma growth and metastasis. To test this, we used cell culture-based assays to measure the impact of ATAD2 inhibition on melanoma growth and metastases. First, we examined the effects of ATAD2 inhibition on melanoma growth in a cell culture model

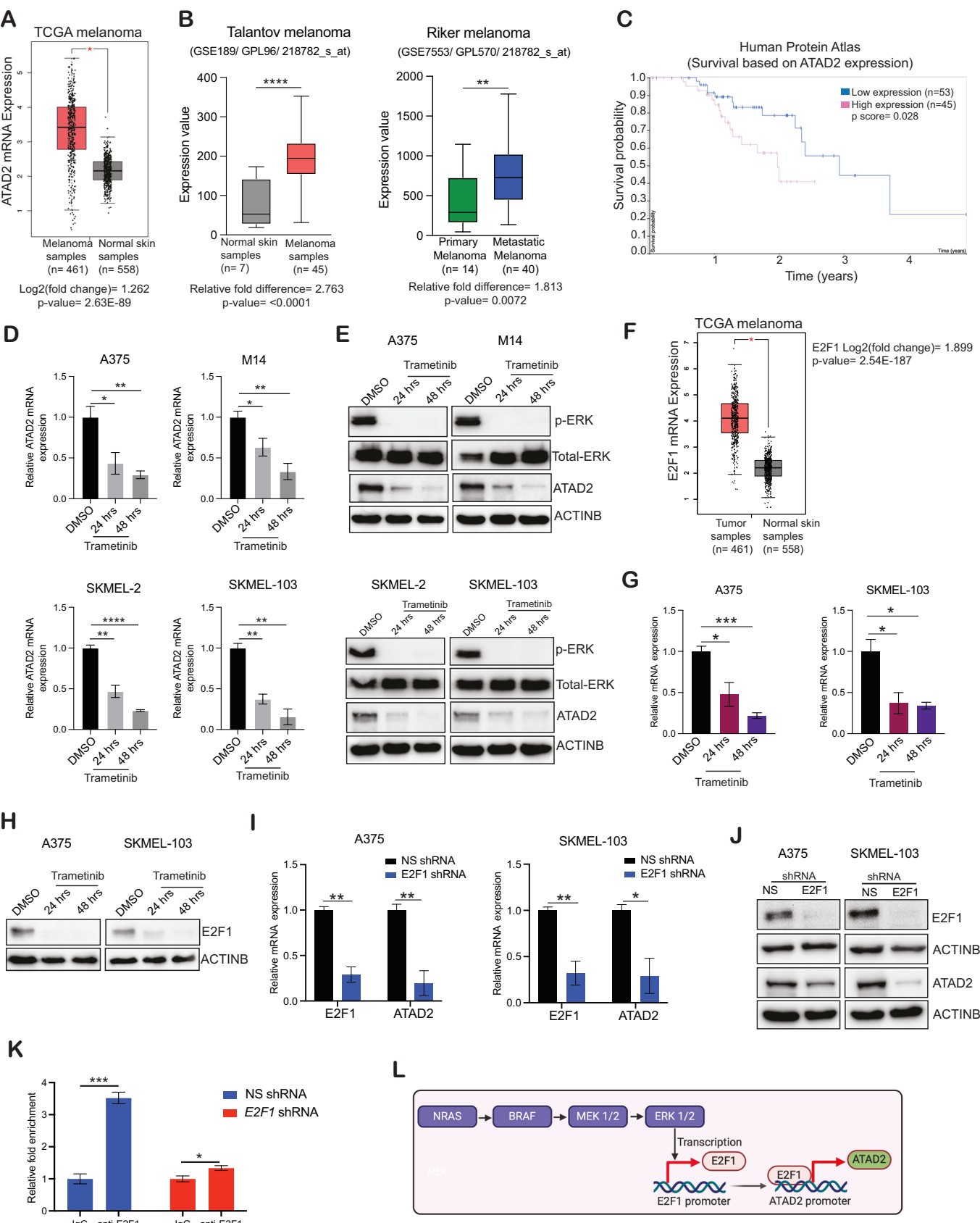

**Figure 1. ATAD2 is overexpressed in patient-derived melanoma samples and is regulated by the MAPK pathway.**

(A) The Gene Expression Profiling Interactive Analysis (GEPIA) was used for plotting *ATAD2* mRNA expression. Its mRNA expression is shown in melanoma patient samples ($n = 461$) and normal skin samples ($n = 558$). (B) *ATAD2* mRNA expression was plotted using Talantov dataset (normal skin and melanoma sample, ****$P = < 0.0001$) and Riker dataset (primary melanoma and metastatic melanoma sample, **$P = 0.0072$). (C) Survival analysis of melanoma patients in Human Protein Atlas datasets based on *ATAD2* expression (low, $n = 53$; high, $n = 45$) with $P = 0.028$. (D, E) The indicated melanoma cell lines were treated with either DMSO or the MEK inhibitor trametinib (200 nM) for 24 and 48 h. (D) *ATAD2* mRNA expression was measured using RT-qPCR and plotted as the level in trametinib-treated cells relative to that in DMSO-treated cells. ACTINB was used as a normalization control, *$P = 0.0391$, **$P = 0.0072$, *$P = 0.0493$, **$P = 0.0058$, from left to right—top panel, **$P = 0.0031$, ****$P = < 0.0001$, **$P = 0.0017$, **$P = 0.0017$ from left to right—bottom panel. (E) ATAD2 protein expression was measured using western blot analysis under the indicated conditions. ACTINB was used as a loading control. (F) *E2F1* mRNA expression was plotted using Gene Expression Profiling Interactive Analysis (GEPIA). (G, H) The indicated melanoma cell lines were treated with either DMSO or the MEK inhibitor trametinib (200 nM) for 24 and 48 h. (G) *E2F1* mRNA expression was measured using RT-qPCR and plotted as the level in trametinib-treated cells relative to that in DMSO-treated cells. ACTINB was used as a normalization control, *$P = 0.0299$, ***$P = 0.0004$, *$P = 0.0316$, *$P = 0.0121$, from left to right (H) E2F1 protein expression was measured using western blot analysis under the indicated conditions. ACTINB was used as a loading control. (I, J) A375 and SKMEL-103 cells expressing either non-specific (NS) shRNAs or shRNAs targeting *E2F1* were analyzed for *E2F1* and *ATAD2* mRNA expression using RT-qPCR. *E2F1* and *ATAD2* mRNA expression is plotted relative to NS shRNA-expressing cells. ACTINB was used as a normalization control, **$P = 0.0016$, **$P = 0.0061$, **$P = 0.0072$, *$P = 0.0243$, from left to right (I). A375 and SKMEL-103 cells expressing non-specific shRNA or shRNAs targeting *E2F1* were analyzed for the indicated proteins by immunoblotting. ACTINB was used as a loading control (J). (K) CUT-&-RUN analysis of E2F1 binding on *ATAD2* promoter in A375 expressing either non-specific (NS) shRNAs or *E2F1* shRNA. IgG was used as a negative control for CUT-&-RUN, and E2F1 fold-enrichment on *ATAD2* promoter plotted relative to IgG and is shown, ***$P = 0.0004$, *$P = 0.0455$, from left to right. (L) Schematics showing the regulation of ATAD2 by the MAPK pathway in a transcription factor E2F1-dependent manner. (B, D, G, I, K) $P$ value was calculated using unpaired Student's $t$ test using three independent replicates. (A, F) $P$ value was calculated using ANOVA (Analysis of Variance) for differential expression analysis between tumor and normal tissues. Source data is available online for this figure. Source data are available online for this figure.

using ATAD2-targeting small-molecule inhibitor BAY-850. BAY-850 is a potent and isoform-selective small-molecule ATAD2 inhibitor with an $IC_{50}$ of 166 nM. BAY-850 binds to the bromodomain of ATAD2, preventing it from interacting with acetylated histones (Kitahata et al, 2017). We treated BRAF mutant (A375 and M14) and NRAS mutant (SKMEL-103 and SKMEL-2) cells with BAY-850 and assessed cell viability using MTT assays. We found that BAY-850 inhibited the cell viability of both BRAF mutant and NRAS mutant melanoma cells (Fig. 2A). Contrary to this, normal human cells, such as primary human fibroblasts and melanocytes that express lower levels of ATAD2 as compared to melanoma cells, were significantly less sensitive to BAY-850 treatment as compared to the melanoma cell lines (Fig. EV1F,G).

We also examined the impact of BAY-850 on long-term melanoma cell survival using clonogenic assays. In this assay as well, BAY-850 inhibited the colony-forming ability of melanoma cells (Figs. 2B and EV2A). We next assessed the impact of BAY-850 on anchorage-independent growth of melanoma cells using the soft-agar assay. The soft-agar assay is a surrogate assay for tumor growth and has been used to evaluate the effects of treatments on in vivo tumor growth (Borowicz et al, 2014; Lin et al, 2014). Consistent with the results of MTT and clonogenic assays, BAY-850 treatment significantly inhibited melanoma cell growth in the soft-agar assay (Fig. 2C,D). Next, we performed the wound-healing assay to assess whether inhibition in cell viability also affects the ability of melanoma cells to migrate in the presence or absence of BAY-850 treatment. Compared with the control, treatment with BAY-850 significantly blocked melanoma cell migration (Fig. EV2B,C). We further confirmed that the inhibition of cell migration is a result of reduced cell viability, as changes in the expression of cellular migration genes did not occur at the sublethal dose of BAY-850 (Fig. EV2D,E).

To further bolster the results obtained using the pharmacological inhibition of ATAD2 with BAY-850, we leveraged a genetic approach. We knocked down the expression of ATAD2 using shRNAs in melanoma cell lines (A375 and SKMEL-103) while using non-specific (NS) shRNA-expressing cells as controls (Fig. 2E,F). We then performed a soft agar assay and a wound-

healing assay. We found that shRNA-mediated knockdown of ATAD2 in melanoma cells also resulted in reduced growth and migration (Figs. 2G,H and EV2F,G). In addition, we observed that ATAD2 overexpression partly rescued melanoma growth inhibition upon BAY-850 treatment (Fig. EV2H,I). Collectively, these results demonstrated that both pharmacological and genetic inhibition of ATAD2 suppresses melanoma tumor growth and metastases in cell culture.

## ATAD2 inhibition suppresses tumor growth and metastasis in the mouse model of melanoma

We next tested whether ATAD2 inhibition suppresses melanoma tumor growth and spontaneous metastasis in the mouse model of melanoma. To assess this, we labeled the metastatic BRAF mutant melanoma cell line A375-MA2 and the metastatic NRAS mutant melanoma cell line SKMEL-103 with the firefly luciferase gene (F-Luc). These F-Luc-labeled A375-MA2-F-Luc and SKMEL-103-F-Luc cells were subcutaneously injected into the flanks of immunodeficient NSG mice. These mice were then treated with vehicle or BAY-850, and subcutaneous tumor growth and spontaneous metastasis to distal organs were monitored by measuring tumor volumes and using the F-Luc-based bioluminescence imaging (Fig. 3A). We found that the BAY-850 treatment significantly suppressed tumor growth of BRAF mutant and NRAS mutant melanoma in mice compared to vehicle-treated mice (Fig. 3B–E,G,H). We also found that BAY-850 treatment significantly inhibited lung metastasis of BRAF mutant and NRAS mutant melanoma tumors in mice as compared to vehicle treatment (Fig. 3F,I). These results demonstrated that ATAD2 inhibitor BAY-850 effectively inhibits both melanoma tumor growth and metastasis in vivo in melanoma mouse models.

To further establish the clinical relevance of ATAD2 as a target for melanoma therapy, we tested the efficacy of BAY-850 in a human melanoma patient-derived xenograft (PDX)-based mouse model. PDX models of cancer closely resemble human patient's tumors and are used to determine the efficacy of investigational cancer therapeutic agents (Goto, 2020). We found that BAY-850

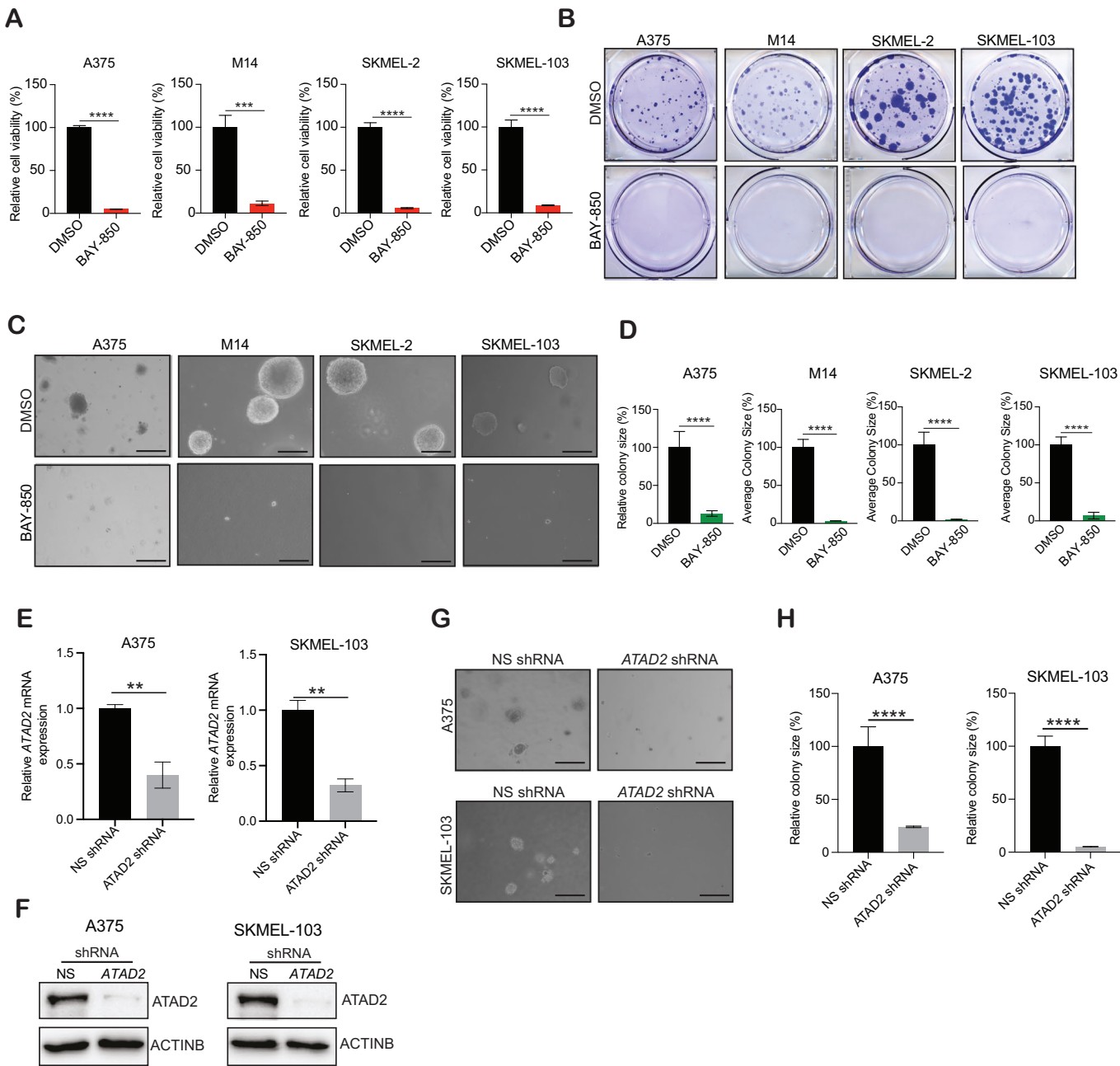

**Figure 2. ATAD2 inhibition suppresses melanoma growth in cell culture model.**

(A) The indicated melanoma cell lines were treated with 5 µM of BAY-850 for 3 days, and viability was assessed by 3-(4,5-dimethylthiazol-2-yl)-2,5-diphenyltetrazolium bromide (MTT) assay. Percentage cell viability in BAY-850-treated condition relative to the DMSO-treated condition is presented, ****$P = < 0.0001$, ***$P = 0.0008$, ****$P = < 0.0001$, ****$P = < 0.0001$, from left to right. (B) The indicated melanoma cell lines were treated with 5 µM of BAY-850 for 2–4 weeks. Cell survival was measured using clonogenic assays. Representative images are shown. (C) The indicated melanoma cell lines were treated with 5 µM of BAY-850 and analyzed for their ability to grow in soft-agar assays. Representative images are shown; scale bar, 500 µm. (D) Relative colony size (%) in BAY-850 treated relative to the DMSO-treated cells is plotted for the experiment shown in (C), ****$P = < 0.0001$, ***$P = < 0.0001$, ****$P = < 0.0001$, ***$P = < 0.0001$, from left to right. (E, F) A375 and SKMEL-103 cells expressing either non-specific (NS) shRNAs or shRNAs targeting *ATAD2* were analyzed for *ATAD2* mRNA expression using RT-qPCR. *ATAD2* mRNA expression is plotted relative to NS shRNA-expressing cells. ACTINB was used as a normalization control (E), **$P = 0.0081$, **$P = 0.0030$, from left to right. A375 and SKMEL-103 cells expressing non-specific shRNA or shRNAs targeting *ATAD2* were analyzed for the indicated proteins by immunoblotting (F). (G) The indicated melanoma cell lines expressing (NS) shRNAs or *ATAD2* shRNAs were analyzed for their abilities to grow in soft-agar assays. Representative images are shown; scale bar, 500 µm. (H) Relative colony size (%) in *ATAD2* shRNA expression cells relative to the control NS shRNA expressing cells shown in (G) is plotted, ****$P = < 0.0001$, ****$P = < 0.0001$, from left to right. (A, D, E, H) $P$ value was calculated using unpaired Student's $t$ test using three independent replicates. Source data is available online for this figure. Source data are available online for this figure.

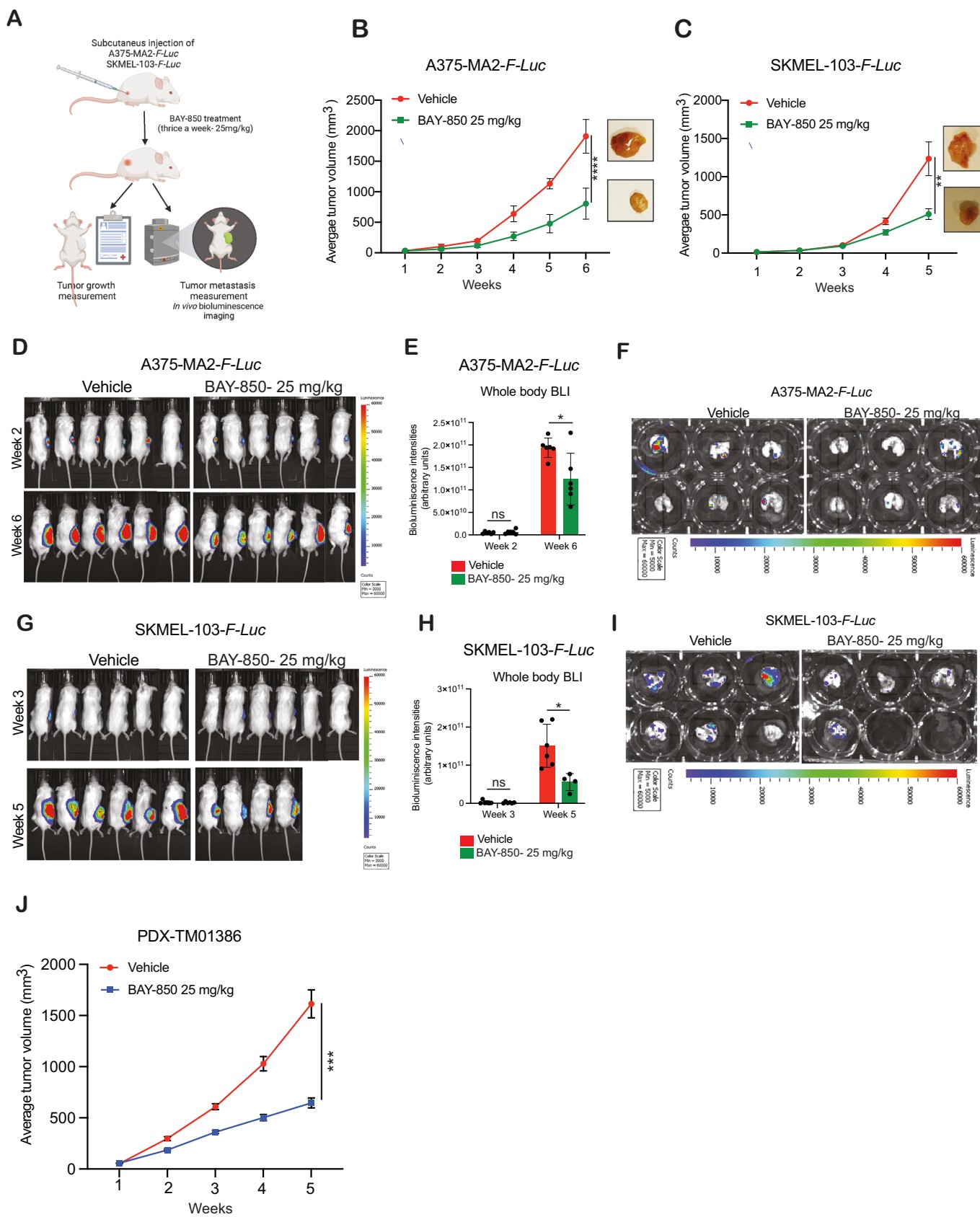

**Figure 3.** ATAD2 targeting inhibits melanoma tumor growth and progression in vivo.

(A) Schematics showing the procedure for performing the in vivo experiment. (B, C) Firefly luciferase–labeled A375-MA2-*F-Luc* and SKMEL-103-*F-Luc* cells were subcutaneously injected into the flanks of NSG mice (*n* = 6). The mice were administered vehicle or BAY-850 (25 mg/kg body weight) intraperitoneally thrice a week. The average tumor volume was measured weekly and plotted. Representative images of subcutaneous tumors at the experimental endpoint treated with either vehicle or BAY-850 is shown next to graph, ****$P$ = < 0.0001 for panel B and **$P$ = 0.0017 for (C). (D, G). Tumor growth and spontaneous metastasis was also assessed by imaging the mice using IVIS imaging. Whole body bioluminescence images of the mice treated with either vehicle or BAY-850 for the experiment shown in (B, C) is shown. (E, H) Representative bioluminescence intensities (BLI) for the whole body was measured and plotted from the mice images treated with either vehicle and BAY-850 shown in (D, G), ns $P$ = 0.5814, *$P$ = 0.0194 for (E) and ns $P$ = 0.7018, *$P$ = 0.0133 for (H). (F, I). Bioluminescence images for lungs obtained from the mice treated with either vehicle or BAY-850 shown in (D, G). (J). PDXs were subcutaneously injected into the flanks of NSG mice (*n* = 5). The mice were then treated with vehicle or BAY-850 (25 mg/kg body weight) intraperitoneally thrice a week, and tumor growth was measured. The average tumor volumes were measured and plotted each week, ***$P$ = 0.0002. (B, C, E, H, J) $P$ value was calculated using unpaired Student's *t* test using three independent replicates. Source data is available online for this figure. Source data are available online for this figure.

treatment significantly inhibited melanoma PDX growth in mice compared with vehicle-treated mice (Fig. 3J). Collectively, these studies demonstrated that the ATAD2 inhibitor BAY-850 blocks melanoma tumor growth and metastasis in mice, including in a clinically relevant PDX-based model of melanoma.

## ATAD2 inhibition induces ferroptosis in melanoma cells

Since ATAD2 is a histone reader, therefore, to understand the mechanism by which ATAD2 facilitates melanoma tumor growth and metastasis, we performed RNA-sequencing (RNA-seq) analysis and investigated global changes in the transcriptome. For RNA sequencing analysis, we treated melanoma cells (A375 [BRAF mutant] and SKMEL-103 [NRAS mutant]) with either DMSO or BAY-850. In A375 cells, BAY-850 treatment resulted in the significant (≥1.5-fold and $P$ value < 0.05) differential expression of 2904 genes (either upregulated or downregulated) compared to DMSO-treated A375 cells (Fig. EV3A,B; Dataset EV1). In case of SKMEL-103 cells, BAY-850 treatment resulted in the significant (≥1.5-fold and $P$ value < 0.05) differential expression of 3321 genes (either upregulated or downregulated) compared to DMSO-treated SKMEL-103 cells (Fig. EV3C,D; Dataset EV2). Using the list of differentially expressed genes, we performed KEGG pathway enrichment analysis and identified pathways to which differentially expressed genes were enriched (Fig. EV3E,F). We found enrichment of various signaling pathways based on gene ratio and $P$ values in A375 and SKMEL-103 cell lines following BAY-850 treatment (Fig. EV3E,F). Notably, cellular senescence was enriched in A375 cells (Fig. EV3E, Dataset EV3), while the AMPK signaling pathway was enriched in SKMEL-103 cells (Fig. EV3F; Dataset EV4). Both cellular senescence induction (Schmitt et al, 2022) and activation of the AMPK pathway (Schneider and Gartenhaus, 2010) have been shown to exert tumor-suppressive effects in melanoma and other cancer types. Based on these findings, we investigated the effect of senescence induction in BRAF mutant cells and AMPK pathway activation in NRAS mutant cells upon BAY-850 treatment.

We first measured senescence induction in BRAF mutant cells upon BAY-850 treatment using the senescence-associated beta-gal (SA β-gal) assay as a readout. As anticipated, we found that the treatment of BRAF mutant melanoma cells (A375 and M14) with BAY-850 induced cellular senescence (Fig. EV4A). IL6 is a known senescent associated secretory factor and is produced by senescent cells (Herbstein et al, 2024), therefore we also measured IL6 level upon BAY-850 treatment in these cells. We found higher IL6 levels

in A375 and M14 cells upon BAY-850 treatment as compared to DMSO treatment (Fig. EV4B). We next investigated the mechanism by which inhibition of ATAD2 results in senescence induction in BRAF mutant melanoma cells. To determine this, we analyzed our RNA-seq data and found an increase in CDKN1A expression upon BAY-850 treatment (Dataset EV1). CDKN1A also known as p21, is a cyclin-dependent kinase inhibitor 1A that can induce cellular senescence induction (Gorgoulis et al, 2019; Wang et al, 2022). Consistent with the results of our RNA-seq, we observed increased expression of CDKN1A at both the mRNA and protein levels upon BAY-850 treatment in A375 and M14 cells (Fig. EV4C,D). To further establish that CDKN1A is a direct target of ATAD2, we performed a CUT&RUN assay. We observed significant enrichment of ATAD2 on the CDKN1A promoter, which was reduced following BAY-850 treatment (Fig. EV4E), indicating that ATAD2 contributes to CDKN1A repression. In addition, we investigated whether inhibition of CDKN1A using UC2288, a CDKN1A-specific inhibitor (Wettersten et al, 2013) could rescue CDKN1A-dependent senescence induction upon BAY-850 treatment in melanoma cells. We observed that treatment with UC2288 led to a partial, yet significant, rescue of senescence induction in A375 and M14 cells upon treatment with BAY-850 (Fig. EV4F). These results demonstrated that one of the mechanisms by which the ATAD2 inhibitor BAY-850 inhibits the growth of BRAF mutant melanoma is through senescence induction via CDKN1A.

We next investigated NRAS mutant cell lines. Since the AMPK signaling pathway was enriched in RNA-sequencing dataset obtained from SKMEL-103 cells (Fig. EV3F; Dataset EV4), we examined the activation of the AMPK pathway and its impact on NRAS mutant cells (SKMEL-103 and SKMEL-2) following BAY-850 treatment. We observed increased phospho-AMPK levels in SKMEL-103 and SKMEL-2 cells following BAY-850 treatment (Fig. EV4G). Since AMPK activation has been linked to the induction of apoptosis, particularly in cancer cells, through several mechanisms including inhibition of the mTOR pathway, regulation of p53 activity, and modulation of Bcl-2 family proteins (Jones et al, 2005; Kilbride et al, 2010), we next measured caspase 3 level as an indicator of apoptosis. Consistently, we found increased apoptosis in BAY-850-treated SKMEL-103 and SKMEL-2 cells (Fig. EV4H). To confirm the role of AMPK in inducing apoptosis in NRAS mutant melanoma cells, we first treated the SKMEL-103 and SKMEL-2 cells with the AMPK agonist AICAR and measured apoptosis. Similar to BAY-850 treatment, AICAR also effectively induced apoptosis in SKMEL-103 and SKMEL-2 cells (Fig. EV4I). In addition, we treated SKMEL-103 and SKMEL-2 cells with

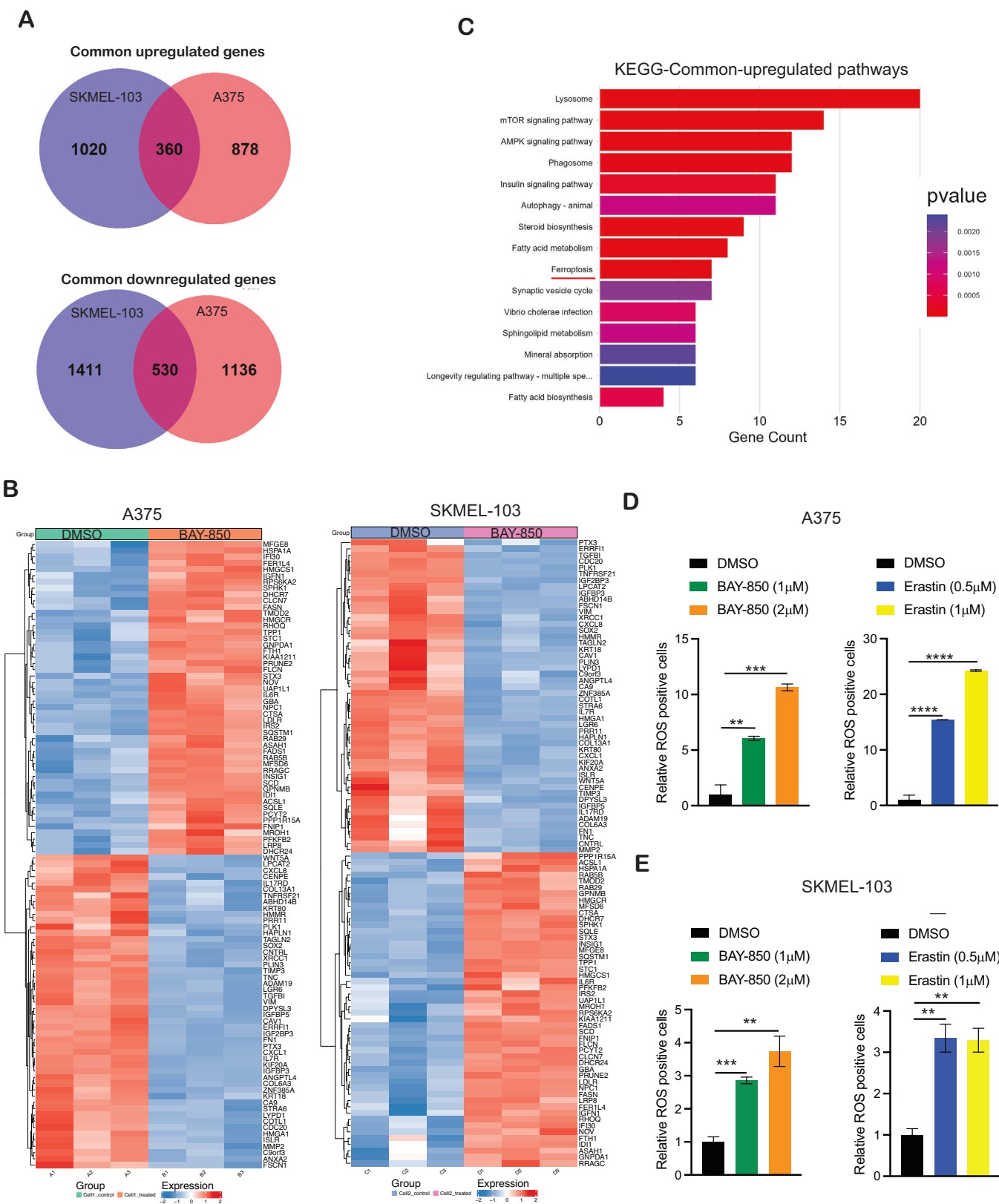

Figure 4. **ATAD2 targeting results in ferroptosis induction in melanoma cells.**

(A) Venn diagram showing common upregulated and downregulated genes in both A375 and SKMEL-103 cells upon treatment with either DMSO or ATAD2 inhibitor BAY-850 5 μM for 48 h. (B) Heatmap showing the top common 50 upregulated and top common 50 downregulated genes in both A375 and SKMEL-103 cells treated with either DMSO or ATAD2 inhibitor BAY-850 5 μM for 48 h. (C) Common pathways significantly upregulated based on gene expression changes were analyzed using KEGG pathway enrichment analysis, and the top 15 significantly altered functional pathways are presented. (D, E) ROS level was measured using CellROX™ Green Flow Cytometry Assay Kit. Relative ROS-positive cells in each condition is shown, $**P = 0.0047$, $***P = 0.0005$, $****P = < 0.0001$, $****P = < 0.0001$, from left to right (top, D) and $***P = 0.0005$, $**P = 0.0048$, $**P = 0.0032$, $**P = 0.0022$, from left to right (bottom, E). (C) P value was calculated using Quasi-Likelihood (QL) F-Tests. (D, E) P value was calculated using unpaired Student's $t$ test using three independent replicates. Source data is available online for this figure. Source data are available online for this figure.

BAY-850 and AMPK inhibitor Compound C. We observed that Compound C treatment partially rescued BAY-850-dependent AMPK upregulation-mediated apoptosis induction in SKMEL-103 and SKMEL-2 cells (Fig. EV4J). These results demonstrated that AMPK activation in NRAS mutant melanoma following BAY-850 treatment represents one of the tumor-suppressive mechanisms by which ATAD2 suppression induces apoptosis and inhibits their growth.

We then asked whether ATAD2, in addition to regulating different tumor-suppressive pathways, also modulates common tumor-suppressive pathways in both BRAF and NRAS mutant melanoma. To identify such common tumor-suppressive pathways, we analyzed genes that were commonly altered in both BRAF mutant A375 and NRAS mutant SKMEL-103 cell lines upon BAY-850 treatment. We found that a total of 360 genes were significantly upregulated (≥1.5-fold and P value < 0.05) and 530 genes that were significantly downregulated (P value < 0.05) in both BRAF mutant A375 and NRAS mutant SKMEL-103 cell lines following BAY-850 treatment (Figs. 4A,B and EV4K). We then performed KEGG pathway enrichment analysis to identify common pathways in which these common differentially regulated genes were enriched (Dataset EV5) based in gene ratio and P values. This analysis revealed enrichment of pathways associated with the lysosome, autophagy, and others. Notably, ferroptosis emerged as a key functional pathway that was significantly upregulated following BAY-850 treatment (Fig. 4C; Dataset EV5). Ferroptosis is a form of regulated cell death characterized by the accumulation of lipid peroxides, driven by iron and reactive oxygen species (ROS) (Pu et al, 2022). It plays a critical role in various diseases, including cancer, and has emerged as a promising therapeutic strategy in cancer treatment, with different approaches being developed to activate this pathway for clinical benefit (Pu et al, 2022). Notably, the role of ATAD2 in regulating ferroptosis in cancer cells, particularly melanoma, has not been previously documented. Therefore, we investigated the ferroptosis pathway as a potential mediator of ATAD2 function in melanoma.

We first measured intracellular reactive oxygen species (ROS) levels in BAY-850-treated melanoma cells using FACS cytometry-based assay for the detection of oxidative stress. As a positive control, the ferroptosis inducer erastin was used. ROS production is a key biomarker of ferroptosis induction (Xie et al, 2016)). We observed that the treatment of cells with BAY-850 resulted in significant upregulation of ROS similar to erastin treatment in melanoma cells (Fig. 4D,E). We used another method to confirm our results by quantitatively measuring ROS levels in melanoma cells (A375 and SKMEL-103) using a cellular ROS detection assay kit. In this assay also, we observed cellular ROS was increased upon

BAY-850 treatment (Fig. EV5A). We next investigated the mechanism by which BAY-850 treatment resulted in ferroptosis induction. We checked the expression of genes that are known to promote ferroptosis in the RNA sequencing data. Our analysis revealed that BAY-850 treatment decreased the expression of glutathione peroxidase 4 (GPX4) (Datasets EV1 and EV2). GPX4 is an enzyme that is critical in maintaining cellular redox balance and reduces lipid hydroperoxides to non-toxic lipid alcohols, using glutathione as a cofactor. This prevents lipid peroxidation and protects cells from ferroptosis, a type of iron-dependent oxidative cell death (Yang et al, 2014; Yang and Stockwell, 2016). We further validated the RNA sequencing results and measured both mRNA and protein expression levels of GPX4 upon BAY-850 treatment. We observed that BAY-850 treatment led to a decrease in mRNA and protein expression of GPX4 (Fig. 5A,B). GPX4 tends to be overexpressed in melanoma patient samples compared to normal samples (Fig. EV5B). Based on these results, we asked if ATAD2 directly binds to the GPX4 promoter. Therefore, we performed CUT&RUN assay to monitor ATAD2 enrichment on the GPX4 promoter in BAY-850-treated or control DMSO-treated conditions in melanoma cells. We observed increased enrichment of ATAD2 on the GPX4 promoter in melanoma cells, which was decreased following BAY-850 treatment (Fig. 5C). To confirm the role of GPX4 in mediating ferroptosis in BAY-850-treated melanoma cells, we overexpressed GPX4 in melanoma cells and then treated the cells with BAY-850 (Fig. EV5C). We observed that overexpression of GPX4 rescued BAY-850-dependent ferroptosis induction and cell viability in melanoma cells (Fig. EV5D,E), confirming that inhibition of GPX4 in melanoma cells is one of the mechanisms by which BAY-850 treatment contributes to ferroptosis induction and tumor growth inhibition.

Based on these results, we further investigated if ferroptosis inducer erastin (Dolma et al, 2003) could inhibit melanoma tumor growth like the ATAD2 inhibitor BAY-850. We observed that erastin could effectively inhibit melanoma tumor growth identifying it as promising therapeutic strategy for treating melanoma (Fig. 5D,E). Collectively, these results demonstrated that ATAD2 loss results in the downregulation of GPX4, which leads to ferroptosis induction and tumor growth suppression in melanoma.

## Combination of ATAD2 inhibitor and MEK inhibitor more potently inhibit melanoma

MEK inhibitors have shown efficacy in treating both BRAF mutant and NRAS mutant melanoma, and we found that the MAP kinase pathway was necessary for ATAD2 expression. In the past, targeting multiple nodes of the same pathway such as

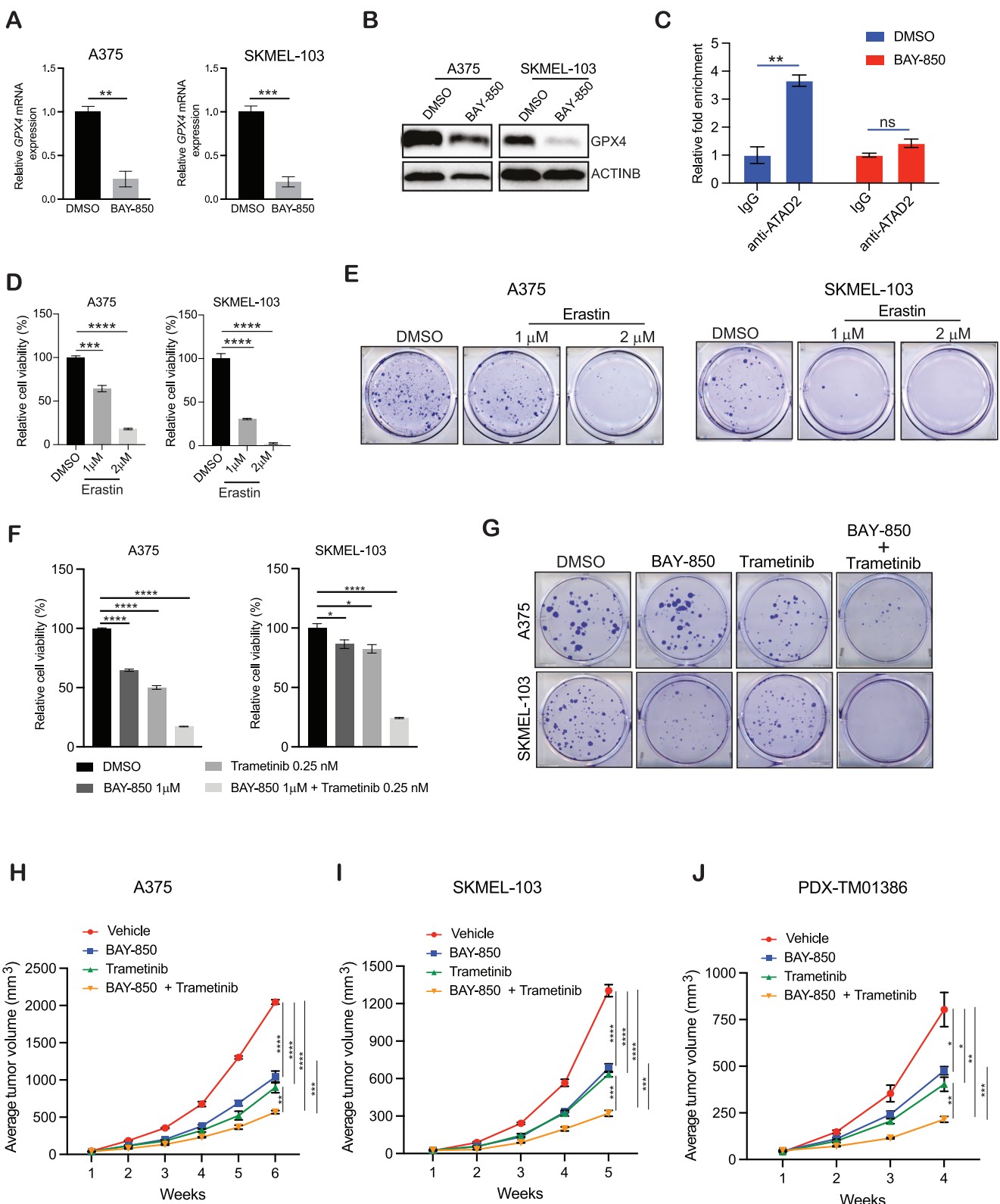

**Figure 5. Ferroptosis inducer and ATAD2 co-targeting with trametinib cause potent melanoma tumor growth inhibition.**

(A, B) The indicated melanoma cell lines were treated with either DMSO or ATAD2 inhibitor BAY-850 5 µM for 48 h. (A) *GPX4* mRNA expression was measured using RT-qPCR and plotted as the level in BAY-850-treated cells relative to that in DMSO-treated cells. ACTINB was used as a normalization control, **$P = 0.0022$, ***$P = 0.0008$, from left to right. (B) GPX4 protein expression was measured using western blot analysis under the indicated conditions. ACTINB was used as a loading control. (C) CUT-&-RUN analysis of ATAD2 binding on *GPX4* promoter in A375 expressing either treated with DMSO or BAY-850. IgG was used as a negative control for CUT-&-RUN, and fold-enrichment plotted relative to IgG is shown, **$P = 0.0018$, ns $P = 0.0646$, from left to right. (D) The indicated melanoma cell lines were treated with various concentrations of ferroptosis inducer erastin for 3 days, and viability was assessed by 3-(4,5-dimethylthiazol-2-yl)-2,5-diphenyltetrazolium bromide (MTT) assay. Relative cell viability in the treated condition relative to the DMSO-treated condition is presented, ***$P = 0.0001$, ****$P = < 0.0001$, ****$P = < 0.0001$, ****$P = < 0.0001$, from left to right. (E) The indicated melanoma cell lines were treated with the indicated concentrations of erastin for 2–4 weeks. Cell survival was measured using clonogenic assays. Representative images are shown. (F) The indicated melanoma cell lines were treated with either DMSO or BAY-850 1 µM alone or trametinib 0.25 nM alone or combination of BAY-850 1 µM and trametinib 0.25 nM for 3 days, and viability was assessed by 3-(4,5-dimethylthiazol-2-yl)-2,5-diphenyltetrazolium bromide (MTT) assay. Relative cell viability in treated condition relative to DMSO-treated condition is presented, ****$P = < 0.0001$, ****$P = < 0.0001$, ****$P = < 0.0001$, *$P = 0.0403$, *$P = 0.0145$, ****$P = < 0.0001$, from left to right. (G) The indicated melanoma cell lines were treated with either DMSO or BAY-850 1 µM alone or trametinib 0.25 nM alone or combination of BAY-850 1 µM and trametinib 0.25 nM for 2–4 weeks. Cell survival was measured using clonogenic assays. Representative images are shown. (H, I) A375 and SKMEL-103 cells were subcutaneously injected into the flanks of NSG mice ($n = 5$). The mice were administered either vehicle or BAY-850 (25 mg/kg body weight) or trametinib (0.5 mg/kg body weight) or combination of both BAY-850 (25 mg/kg body weight) and trametinib (0.5 mg/kg body weight) intraperitoneally thrice a week. The average tumor volume was measured weekly and plotted. ****$P = < 0.0001$, ****$P = < 0.0001$, ****$P = < 0.0001$, ***$P = 0.0002$, **$P = 0.0074$, from top to bottom for A375 and ****$P = < 0.0001$, ****$P = < 0.0001$, ****$P = < 0.0001$, ***$P = 0.0001$, ***$P = 0.0004$, from top to bottom for SKMEL-103. (J) PDX was subcutaneously injected into the flanks of NSG mice ($n = 5$). The mice were administered either vehicle or BAY-850 (25 mg/kg body weight) or trametinib (0.5 mg/kg body weight) or combination of both BAY-850 (25 mg/kg body weight) and trametinib (0.5 mg/kg body weight) intraperitoneally thrice a week. The average tumor volume was measured weekly and plotted, *$P = 0.0458$, *$P = 0.0186$, **$P = 0.0014$, ***$P = 0.0001$, **$P = 0.0045$, from top to bottom. (A, C, D, F, H–J) $P$ value was calculated using unpaired Student's $t$ test using three independent replicates. Source data is available online for this figure. Source data are available online for this figure.

co-targeting mutant BRAF along with a MEK or ERK inhibitor, has been shown to provide better efficacy in the clinic for metastatic melanoma treatment (Fernandez-Montalvan et al, 2017; Long et al, 2014). Therefore, we asked if a MEK inhibitor could be combined with the ATAD2 inhibitor for better therapeutic efficacy. We tested the effect of ATAD2 inhibitor BAY-850 in combination with the MEK inhibitor trametinib in cell culture, subcutaneous xenografts, as well as a melanoma PDX model. We first performed both short-term cell viability MTT assay and a long-term viability clonogenic assay to measure the effect of BAY-850 in combination with trametinib on melanoma tumor growth. We observed that combined treatment with BAY-850 and trametinib resulted in stronger melanoma tumor growth inhibition than either drug alone when used in suboptimal dose (Figs. 5F,G and EV5F) in both MTT and clonogenic assays. In addition, treatment with ATAD2 inhibitor BAY-850 along with MEK inhibitor trametinib inhibited the growth of subcutaneous cell line xenograft as well as melanoma PDX model (Fig. 5H–J) in vivo. Collectively, these results demonstrated that the MEK inhibitor trametinib can be combined with the ATAD2 inhibitor BAY-850 for more effective suppression of melanoma.

Resistance to targeted therapy is commonly observed in the clinic. Therefore, it was likely that resistance to BAY-850 will eventually develop, when used for treating melanoma. To proactively investigate the potential mechanisms of resistance to BAY-850, we conducted a series of experiments. First, we generated BAY-850-resistant melanoma cell lines by treating them continuously with BAY-850 over an extended period (Fig. 6A) and then checked the changes in phosphorylation of receptor tyrosine kinases (RTKs) in BAY-850-resistant human melanoma cell lines. The decision to examine RTKs was guided by previous studies showing that RTK pathways play an important role in drug resistance in cancer cells (Karunaraj et al, 2025; Olender et al, 2019; Yadav et al, 2012). Numerous RTK-targeting drugs have been developed over the years and are used in the clinic to treat a wide variety of cancers (Tomuleasa et al, 2024). Both A375 and SKMEL-103 BAY-850-resistant cells were analyzed for changes in

phosphorylation of human RTKs using a human RTK array capable of detecting phosphorylation changes across 49 RTKs. We found that phosphorylation of FGFR3, PDGFRβ, ROR1, and EphB3 was increased in BAY-850-resistant A375 cells, while phosphorylation of ROR1, EphA2, and DDR2 was increased in BAY-850-resistant SKMEL-103 cells (Fig. 6B–D). Consistent with these findings, we also observed increased levels of phosphorylated MEK1/2 and ERK1/2 in BAY-850-resistant melanoma cells (Fig. 6E). These results suggested that one of the mechanisms contributing to BAY-850 resistance may involve enhanced signaling through various RTKs and an activated MAP kinase pathway, which can be targeted to overcome resistance. These results also support our rationale for combining MEK inhibitors with BAY-850 to prevent or delay the development of resistance in melanoma.

A series of previous elegant studies have reported phenotypic and cellular state changes in melanoma in response to targeted and immune-based therapies, as well as the association of various cellular states to response and resistance to these therapies (Karras et al, 2022; Pozniak et al, 2024; Rambow et al, 2018). Based on these studies, we investigated whether ATAD2 expression is associated with specific cellular states. To this end, we analyzed ATAD2 expression in datasets from in Rambow et al, (Rambow et al, 2018), Pozniak et al, (Pozniak et al, 2024) and Karras et al, (Karras et al, 2022). We found that, in the Rambow et al dataset, ATAD2 expression was significantly enriched in invasive cell state compared to other cellular states (pigmented, "starved"-like melanoma cells (SMCs) and neural crest stem cells (NCSC)) (Fig. EV5G).

Furthermore, in the Pozniak et al, dataset, we observed that ATAD2 expression was significantly higher in the mitotic cell state compared to other cellular states (Fig. EV5H). The mitotic cell state is a highly proliferative state observed in multiple cancers. Finally, analysis of the Karras et al, dataset, revealed that ATAD2 expression was highest in the neural crest (NC) cell state compared to other cellular states (Fig. EV5I). Notably, stem-like cell state defined in this study had the second-highest ATAD2 expression level after NC cell state.

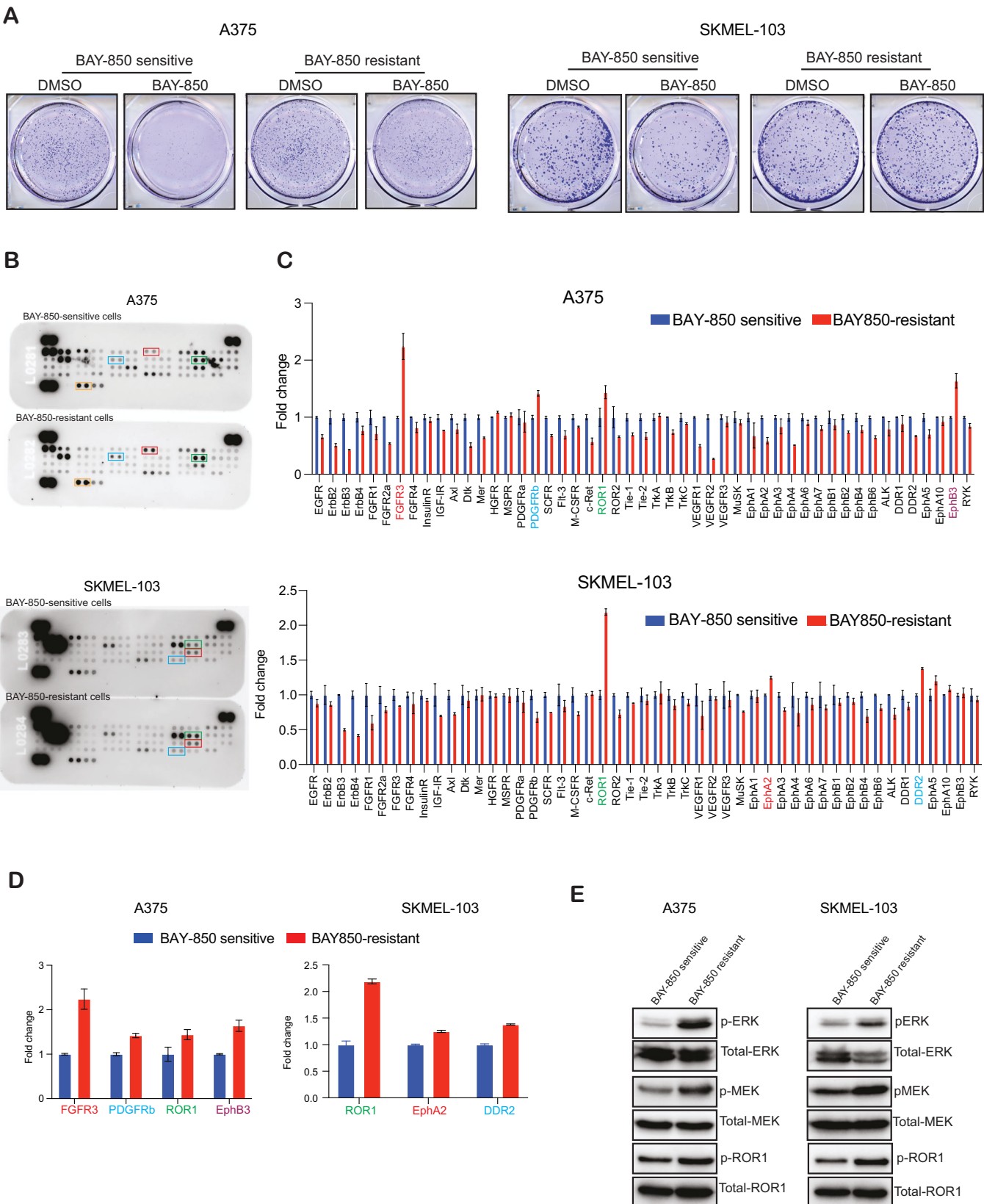

**Figure 6. Elucidating mechanisms of resistance to ATAD2 inhibitor BAY-850 in melanoma cells.**

(A) The indicated melanoma cell lines were treated with BAY-850 for 10 weeks to generate BAY-850-resistant colonies, and their sensitivity and resistance were assessed by performing a clonogenic assay with and without BAY-850 treatment. Representative images are shown. (B–D) Human phospho-RTK arrays were used to identify changes in RTK phosphorylation in BAY-850 inhibitor-resistant A375 and SKMEL-103 cells. Proteome profiler human phospho-RTK array membranes showing relative RTK phosphorylation in BAY-850 inhibitor-resistant cells and BAY-850 inhibitor-sensitive cells (B). Spot intensity was used to calculate the fold changes and plot individual RTK phosphorylation in BAY-850 inhibitor-resistant cells relative to BAY-850 inhibitor-sensitive cells (C). Bar diagram showing candidates that were changed greater than 1.25-fold in BAY-850 inhibitor-resistant cells relative to BAY-850 inhibitor-sensitive cells (D). (E) Indicated proteins were analyzed in BAY-850 inhibitor-resistant and sensitive cells. ACTINB was used as the loading control. Source data is available online for this figure. Source data are available online for this figure.

Previous studies have also shown that an invasive or mesenchymal-like gene expression signature is characterized by low levels of MITF and SOX10 and high levels of AXL (Kemper et al, 2014; Konieczkowski et al, 2014; Muller et al, 2014; Shaffer et al, 2017). This cellular state has also been shown to be intrinsically resistant to MAPK-inhibition. Since ATAD2 inhibition cooperates with MEK inhibitors, it is possible that ATAD2 may regulate invasive or other melanoma-specific cellular states. Future detailed studies will be required to elucidate these connections to invasive and other cellular states.

## Discussion

Although BRAF/MEK pathway-targeting therapies and cancer immunotherapies specifically immune checkpoint blockade therapies provide strong clinical benefits in a subset of melanoma patients, many do not achieve durable benefits from these treatments, highlighting the need for new therapeutic agents (Welsh et al, 2016). Here, we demonstrate that ATAD2 is overexpressed in melanoma, drives the growth of melanoma by suppressing ferroptosis, and its inhibition suppresses melanoma growth and progression (Fig. 7).

Oncogenic mutations in NRAS and BRAF genes in melanoma stimulate the MAP kinase (MAPK) pathway that in turn activates the expression and activity of several transcription factors, which in part mediates the oncogenic effect of the activated MAPK pathway (Boulton et al, 1991; Guo et al, 2020). Related to this, a previous study showed that the RAS–RAF–MEK–ERK pathway stimulates the expression of KLF4 through E2F1 and thereby promotes melanoma growth (Riverso et al, 2017). We observed that the MEK–ERK pathway via the action of E2F1, activates ATAD2 expression in melanoma. Previous studies in breast cancer, colorectal cancer, endometrioid carcinoma, and gastric cancer have shown that overexpression of ATAD2 confers poor prognosis in patients (Hou et al, 2016; Kalashnikova et al, 2010; Krakstad et al, 2015; Wang et al, 2023), including melanoma (Baggiolini et al, 2021). Consistent with these studies, we also found that ATAD2 overexpression predicts poor prognosis in melanoma patients.

ATAD2 is increasingly recognized for its role in tumor growth and metastasis across various cancers (Cao et al, 2021; Guruvaiah et al, 2023; Zheng et al, 2015). Overexpression of ATAD2 has been linked to enhanced proliferation, invasion, and metastasis in various cancers, including ovarian cancer, esophageal squamous cell carcinoma (ESCC), clear cell renal cell carcinoma (ccRCC), and cervical cancer (Cao et al, 2021; Guruvaiah et al, 2023; Wu et al, 2023; Zheng et al, 2015). We found that ATAD2 is necessary for melanoma tumor growth and metastasis. Since ATAD2 inhibition

suppressed primary tumor growth thus it is likely that the reduced metastatic growth is in part due to ATAD2 inhibition-induced tumor suppression.

A previous study identified ATAD2 as a melanoma competence factor that forms a complex with SOX10, enabling the expression of downstream oncogenic and neural crest programs (Baggiolini et al, 2021). In our study, treatment with the ATAD2 inhibitor BAY-850 did not affect SOX10 levels or the expression of SOX10-regulated genes in NRAS mutant melanoma cell lines. In contrast, BRAF mutant cell lines showed increased expression of SOX10 and its downstream targets following BAY-850 treatment. Interestingly, our study thus identified another layer of regulation, in which we found that ATAD2 potentially directly regulates SOX10 expression in BRAF mutant melanoma cell line. These findings suggest that the impact of ATAD2 inhibition on SOX10 and the neural crest transcriptional program depends on the mutational context, with selective upregulation occurring in BRAF mutant, but not NRAS mutant melanoma.

Previous studies have also reported that phenotypic changes in melanoma in response to targeted and immune-based therapies, including dedifferentiation and loss of pigmentation, are associated with ATAD2 (Baggiolini et al, 2021). Furthermore, our analysis of previously published single-cell RNA-seq datasets (Rambow et al, Pozniak et al, and Karras et al) revealed enrichment of ATAD2 expression in distinct cellular states. However, further in-depth investigation is required to fully elucidate the role of ATAD2 in melanoma cell state plasticity and to establish its importance in tumor growth, metastasis, and regulation of therapeutic response.

In our study, we observed that ATAD2 inhibition upregulated distinct and common tumor-suppressive pathways in melanoma based on their genotypes. For instance, ATAD2 inhibition via BAY-850 in BRAF mutant cells upregulated cellular senescence, which was dependent upon the cyclin-dependent kinase inhibitor 1A (CDKN1A or p21) expression, while in NRAS mutant melanoma cell lines, ATAD2 inhibition promoted apoptosis, which was dependent upon upregulation of the AMPK pathway. One of the limitations of our study is that it has a limited number of cell lines with different genetic backgrounds for studying distinct tumor-suppressive pathway activation following ATAD2 inhibition. Therefore, future independent study with a larger cell line panel with BRAF and NRAS mutations will be required to further generalize these findings.

In addition, Ferroptosis was identified as a commonly activated tumor-suppressive pathway in both BRAF and NRAS mutant melanoma upon ATAD2 inhibition, which was mediated by the downregulation of GPX4 expression. Other pathways identified through KEGG pathway analysis, such as autophagy and lysosomal function, were not validated in our follow-up experiments in the

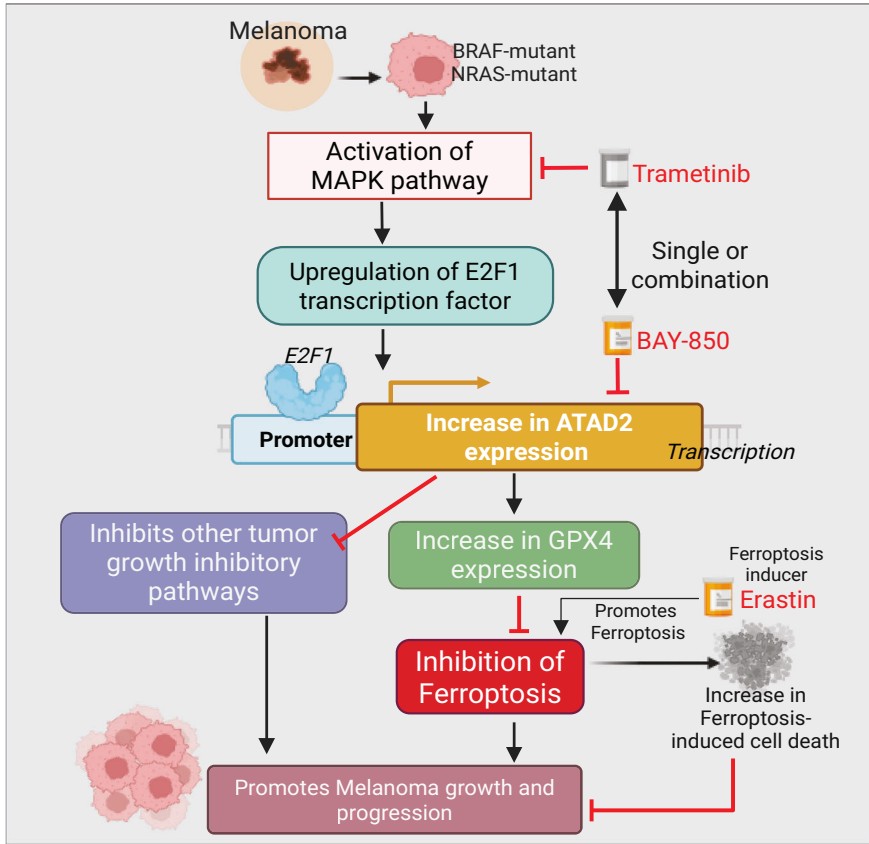

**Figure 7. Model showing the role of ATAD2 in melanoma and its potential as a therapeutic target for melanoma therapy.**

ATPase family AAA domain-containing 2 (ATAD2) is regulated by the MAPK pathway via E2F1 transcription factor in melanoma. Targeting ATAD2 leads to ferroptosis induction in melanoma cells by inhibiting GPX4 expression, contributing to tumor growth suppression. Thus, ATAD2 can be targeted via small-molecule inhibitors, either alone or in combination with a MAPK inhibitor to provide effective melanoma therapy. Figure was created using BioRender. Source data are available online for this figure.

tested conditions. In addition, metabolic pathways, including fatty acid metabolism and steroid biosynthesis, were also identified in the KEGG analysis. Previous studies have shown that ATAD2 is involved in metabolic reprogramming and the regulation of nutritional stress pathways. For instance, it promotes glycolysis and tumor progression in clear cell renal cell carcinoma via c-Myc (Wu et al, 2023), increases [18F]fluorodeoxyglucose uptake through the AKT–GLUT1/HK2 pathway in lung adenocarcinoma (Sun et al, 2019), and contributes to chemoresistance under hypoxia by inhibiting cell cycle progression (Wu et al, 2023). Both AKT signaling (Porstmann et al, 2005) and GLUT1/HK2-regulated glucose metabolism have been shown to influence fatty acid metabolism and steroid biosynthesis (Parhofer, 2015; Xiao et al, 2022). However, to date, no previous study has directly linked ATAD2 to the regulation of ferroptosis. Thus, our study is the first to identify ATAD2 as a regulator of ferroptosis in melanoma. Ferroptosis is a key tumor-suppressive mechanism, critical for cellular homeostasis and increasingly recognized as a promising treatment option.

Our study primarily focuses on BRAF and NRAS mutant melanomas, as these represent the most common and clinically relevant mutations in melanoma. While other mutations such as KIT, NF1, and others do occur, they are significantly less frequent

(Davis et al, 2018; Siroy et al, 2015; Thielmann et al, 2021). Future studies are needed to further investigate the effects of ATAD2 inhibitors and their mechanisms in other genotypes in greater detail.

Our findings have significant clinical implications, as ATAD2 can be pharmacologically targeted using small-molecule inhibitors. This is based on results which shows that ATAD2 inhibitor BAY-850 either alone or in combination with the MEK inhibitor trametinib blocked melanoma tumor growth and metastasis in cell culture and mouse models. It is noteworthy that BAY-850 is less efficacious than trametinib as a standalone therapy. The combination of BAY-850 with trametinib exhibited a potent tumor growth-inhibitory effect in melanoma. Furthermore, the combination treatment with BAY-850 and trametinib significantly reduced and delayed the emergence of drug-resistant colonies compared to trametinib alone. Together, these findings indicate that ATAD2 inhibition, in the context of trametinib treatment, exerts both short-term and long-term growth-inhibitory effects and can suppress the development of resistance. In addition, since the loss of ATAD2 induced ferroptosis, it is likely that ferroptosis inducers can also be used either alone or in combination with ATAD2 inhibitors to enhance its therapeutic benefits. Collectively, these results demonstrate the potential of targeting ATAD2 either alone

or in combination with other drugs as an alternative way to treat melanoma.

Furthermore, we noted that cells which became resistant to BAY-850 showed increased phosphorylation of several receptor tyrosine kinases (RTKs) as well as activation of the MAPK pathway. This further supports our rationale for combining MEK inhibitors and potentially other RTK inhibitors with BAY-850 to prevent or delay the development of resistance in melanoma to ATAD2 targeting. Such a combination could not only enhance therapeutic efficacy by simultaneously targeting different pathways thereby reducing the likelihood of tumor adaptability, it may also allow to lower the required doses of each agent, thus minimizing drug-related toxicity. Future clinical trials with ATAD2 inhibitors either alone or in combinations that we have identified from our studies will be necessary to fully understand the impact and efficacy of these agents in melanoma treatment.

# Methods

### Reagents and tools table

| Reagent or resource | Source | Identifier |
|---|---|---|
| **Antibodies** | | |
| ACTINB | Cell Signaling | Cat# 4970 |
| ATAD2 | Abcam | Cat# ab244431 |
| Phospho-ERK | Cell Signaling | Cat# 4376 |
| Total-ERK | Cell Signaling | Cat# 4695 |
| E2F1 | Cell Signaling | Cat# 3742 |
| CDKN1A | Cell Signaling | Cat# 2947 |
| Phospho-AMPK | Cell Signaling | Cat# 2531 |
| Total-AMPK | Cell Signaling | Cat# 2532 |
| GPX4 | Cell Signaling | Cat# 52455 |
| V5-tag | Cell Signaling | Cat# 13202 |
| **Chemicals, peptides, and recombinant proteins** | | |
| DMEM | GIBCO | Cat# 11965-092 |
| RPMI | GIBCO | Cat# 11875-093 |
| Fetal bovine serum | GIBCO | Cat# 10437-028 |
| Trypsin-EDTA | GIBCO | Cat# 25200-056 |
| Penicillin–streptomycin | GIBCO | Cat# 15140-122 |
| Effectene Transfection Reagent | QIAGEN | Cat# 301427 |
| Methylthiazole tetrazolium (MTT) | Sigma-Aldrich | Cat# M5655 |
| Agarose, Low gelling | Sigma-Aldrich | Cat# A9045 |
| BAY-850 | Sigma, MedChem | Cat# SML2748, Cat# HY-119254 |
| Trametinib | SelleckChem | Cat# GSK1120212 |
| Erastin | SelleckChem | Cat# S7242 |
| Compound C/ Dorsomorphin | SelleckChem | Cat# S7840 |
| AICAR | SelleckChem | Cat# S1802 |
| UC2288 | SelleckChem | Cat# E1190 |

| Reagent or resource | Source | Identifier |
|---|---|---|
| Camptothesin | SelleckChem | Cat# S1288 |
| Human TGF-beta1 recombinant Protein, PeproTech | ThermoFisher Scientific | Cat# 100-21C-50UG |
| **Critical commercial assays** | | |
| In vitro cellular senescence assay | Cell Signaling Technology. | Cat# 25833 |
| Cellular Reactive Oxygen Species Detection Assay Kit | Abcam | Cat# ab186027 |
| Caspase 3 colorimetric assay kit | Sigma | Cat# CASP3C |
| CellROX Green flow Cytometry Assay Kit | Life Technologies | Cat# C10492 |
| CUT&RUN Assay Kit | Cell Signaling | Cat#86652 |
| Human IL-6 ELISA Kit | ThermoFisher Scientific | Cat# EH2IL6 |
| **Deposited data** | | |
| RNA-Seq performed with A375 and SKMEL-103 cells treated with either DMSO or BAY-850 | This paper | GEO: GSE255437 |
| **Experimental models: cell lines/organoids/PDXs** | | |
| 293 T | ATCC | ATCC CRL-3216 |
| A375 | ATCC | ATCC CRL-1619 |
| M14 | CYTION | Cat# 302163 |
| SKMEL-103 | Sigma | Cat# SCC439 |
| SKMEL-2 | ATCC | ATCC HTB-68 |
| Melanoma PDX | Jackson Laboratory | TM01386 |
| Primary human fibroblast cells | ATCC | ATCC PCS-201-012 |
| **Experimental models: organisms/strains** | | |
| Mouse: NSG mice | Jackson Laboratory | Stock No. 005557 |
| **shRNAs** | | |
| ATAD2 shRNA | Sigma-Aldrich | TRC clone ID: TRCN0000158771 |
| E2F1 shRNA | Sigma-Aldrich | TRC clone ID: TRCN0000000251 |
| **Oligonucleotides** | | |
| | **Forward primer** | **Reverse primer** |
| Human ATAD2 | AAGGAAGTTGA AACCTACCACCG | GCAAGTTGCTC CGTTATTTCCA |
| Human E2F1 | ACGCTATGAGA CCTCACTGAA | TCCTGGGTCA ACCCCTCAAG |
| Human CDKN1A | TGTCCGTCAG AACCCATGC | AAAGTCGA AGTTCCATCGCTC |
| Human GPX4 | GAGGCAAGACC GAAGTAAACTAC | CCGAACTGGT TACACGGGAA |
| Human ACTINB | GTCTTCCCCT CCATCGTGGG | CCTCTCTTGCTC TGGGCCTC |
| CDKN1A promoter CUT&RUN primers | GAGGAGGGA AGTGCCCTCC | CGCCGAGCC AGCTGAGCCT |

| Reagent or resource | Source | Identifier |
|---|---|---|
| ATAD2 promoter CUT&RUN primers | GAGCGCGCAGAGGCCTCC | GAGCTCGGCG GAAGGAGAC |
| ACTINB promoter CUT&RUN primers | TCTTGGCTGG GCGTGACTGT | AAGGTGGGCT CTACAGGGCA |
| **Recombinant DNA** | | |
| Plasmid: piggyBac GFP-Luc | Ding et al, 2005 (PMID:16096065) | N/A |
| Plasmid: Act-PBase | Ding et al, 2005 (PMID:16096065) | N/A |
| Plasmid: psPAX2 | Addgene | Cat#12260, RRID:Addgene_12260 |
| Plasmid: pMD2.G | Addgene | Cat#12259, RRID:Addgene_12259 |
| Plasmid: GPX4 ORF | Horizon discovery | Cat#OHS6085-21383245 |
| Plasmid: GPX4 ORF | Origene | Cat#RC218291L3 |
| **Software and algorithms** | | |
| Prism 10.0 | GraphPad | www.graphpad.com/ scientific.software/ prism |
| ImageJ | https://imagej.nih.gov/ij | N/A |

## Cell culture

Melanoma cell lines (A375, M14, SKMEL-2, SKMEL-103), HEK-293T and primary human fibroblast cells were purchased from American Type Culture Collection (ATCC), Sigma and Cytion as listed in Reagents and Tools Table and maintained in a humidified atmosphere of 5% $CO_2$ at 37 °C in Dulbecco's modified Eagle medium (DMEM; Life Technologies, Carlsbad, CA, USA) or Roswell Park Memorial Institute-1640 Medium (RPMI)-1640 Medium; (Life Technologies), each supplemented with 10% fetal bovine serum and 1% penicillin/streptomycin (both from Life Technologies). Information regarding cell culture reagents is listed in the Reagents and Tools Table.

## Chemical inhibitors

BAY-850 (Cat. No.: Cat# SML2748) was purchased from Sigma for cell culture experiments and from MedChem (Cat. No.: HY-119254) for in vivo experiments. Trametinib (Cat. No.: GSK1120212), Erastin (Cat. No.: S7242), AICAR (Cat. No.: S1802), UC2288 (Cat. No.: E1190), Camptothesin (Cat. No.: S1288), and Compound C/Dorsomorphin (Cat. No.: S7840), were purchased from Selleck Chemical LLC, and dissolved for cell culture and in vivo experiments as suggested in the data sheet. Relevant information is provided in the Reagents and Tools Table. The treatment conditions are described in the corresponding Figure legends.

## shRNA, lentivirus preparation, and stable cell line generation

Gene-specific shRNA (*ATAD2* and *E2F1*) and non-specific (NS) control shRNA were obtained from Open Biosystems. The catalog numbers for the shRNAs are provided in the Reagents and Tools Table. For GPX4 and ATAD2 overexpression, CCSB-broad lentiviral expressing human GPX4 was obtained from Horizon

Discovery, and the ATAD2 construct was obtained from Origene. Lentiviral particles were generated by co-transfecting HEK-293T cells with gene-specific or NS shRNA plasmids and lentiviral packaging plasmids- PDM2.G and pSPAX2. Effectene Transfection Reagent (Qiagen, Hilden, Germany) was utilized for every lentiviral transduction. After 48 h, the lentivirus-containing supernatants were harvested, filtered, and used for infections. Melanoma cells were infected with shRNA lentiviral particles in 12-well plates to generate stable cell lines. For cell selection, an appropriate concentration of 0.5 μg/mL puromycin was used.

## MTT assay

For MTT assay, $2 \times 10^3$ of melanoma cells were plated in a 100 μl volume in 96-well plates. After 24 h, DMSO or inhibitors at the concentration shown in figures were mixed in 100 μl of medium and added to the cells. After 3 days of inhibitor treatment, the cell viability was evaluated. To do this, 20 μl of 5 mg/ml MTT solution dissolved in 1× PBS was added to each well and incubated for 2 h at 37 °C incubator. The MTT (1-(4,5-dimethylthiazol-2-yl)-3,5-diphe-nylformazan) solution was removed gently, and 100 μl of DMSO was added. After mixing well by pipetting, absorbance was measured at 590 and 630 nm. An average was calculated for both readings, and then the measurement at 630 nm was subtracted from that at 590 nm. The relative cell viability was plotted with respect to control DMSO-treated cells. Statistical analysis was performed using unpaired Student's *t* tests in the GraphPad Prism 7 software.

## Clonogenic assay

For the Clonogenic assay, melanoma cells were seeded in a six-well plate at $1 \times 10^3$ numbers. Cells were seeded in triplicate and incubated for 24 h, after which the cells were treated with either vehicle DMSO or inhibitors at the concentration shown in the figures every week. After 3–4 weeks, colonies were fixed using a fixing solution containing 50% methanol and 10% acetic acid and then stained with 0.05% Coomassie blue (Sigma-Aldrich, St. Louis, MO, USA). Representative images for each sample under the indicated condition are shown.

## Soft-agar assay

Soft-agar assays were performed by seeding $5 \times 10^3$ melanoma cells mixed with 0.4% low-melting-point agarose (Sigma-Aldrich) layered on top of 0.8% agarose. After 24 h, they were treated with either vehicle DMSO or inhibitors as described in figures. After 3–6 weeks of treatment, colonies were stained with a 0.05% crystal violet solution and imaged using a microscope. Colony size was calculated using ImageJ software (https://imagej.nih.gov/ij/) and plotted as the percent relative colony size in treatment condition as compared with the control DMSO-treated condition. Statistical analysis was performed using unpaired Student's *t* tests in the GraphPad Prism 7 software.

## Wound-healing assay

Melanoma cell lines were seeded in a six-well plate at a density of $2 \times 10^5$ cells per well and were grown until fully confluent. A scratch was then created using a sterile 20 μl pipette tip, and an image was

taken at 0-h time point. After that, the cells were either treated with DMSO or BAY-850 (5 µM). Cell migration or healing into the wound was monitored at 0, 24, and 48 h using light microscopy. Quantification of wound healing for each condition was performed using ImageJ software (https://imagej.nih.gov/ij/). Statistical analysis was performed using unpaired Student's $t$ tests in the GraphPad Prism 7 software.

## In vitro cellular senescence assay

Cellular senescence was measured using a senescence-β-galactosidase activity assay kit (fluorescence, plate-based # 25833; Cell Signaling) following the manufacturer's protocol in a 96-well format. Cell lysate was prepared as per the manufacturer's protocol. To perform the assay, 50 µl of cell lysate was transferred into a 96-well plate, and 50 µl of freshly prepared 2× assay buffer was added to it. The mixture was then incubated at 37 °C for 1–3 h, protected from light. After incubation, 50 µl of the reaction mixture was transferred into a black opaque 96-well plate, and the reaction was stopped by adding 200 µl of senescence stop solution to each well. Fluorescence was measured using a plate reader set to an excitation wavelength of 360 nm and an emission wavelength of 465 nm. Relative senescence in the treated condition with respect to the control DMSO-treated condition is plotted. Statistical analysis was performed using unpaired Student's $t$ tests in the GraphPad Prism 7 software.

## Caspase 3 activity assay for apoptosis measurement

Caspase 3 activity assay was performed using Caspase 3 colorimetric assay kit (CASP3C, Sigma-Aldrich) following the manufacturer's protocol under conditions indicated in figure legends. Caspase 3 activity assays were performed in 100 µl volume in a 96-well plate format using the Biotek Synergy MX Multi Format Microplate Reader (Biotek), and measurements were performed at 405 nm. The Caspase 3 Colorimetric Assay Kit is based on the hydrolysis of acetyl-Asp-Glu-Val-Asp p-nitroanilide (Ac-DEVD-pNA) by caspase 3, resulting in the release of the p-nitroaniline (pNA) moiety. p-Nitroaniline is detected at 405 nm using the Biotek Synergy MX Multi Format Microplate Reader (Biotek). The relative caspase 3 level was plotted in the inhibitor-treated condition with respect to the control DMSO-treated condition. Statistical analysis was performed using unpaired Student's $t$ tests in the GraphPad Prism 7 software.

## ROS measurement using CellROX Green Flow Cytometry assay kit

The cells were treated with different inhibitors shown in the figures for 48 h. ROS production by A375 and SKMEL-103 cell lines was measured using CellROX Green Flow Cytometry assay kit (C10492, Thermo-Fisher Scientific) according to the manufacturer's protocol. Relative ROS-positive cells in the inhibitor-treated condition were plotted with respect to the control DMSO-treated condition.

## ROS measurement using the cellular reactive oxygen species detection assay kit

The level of ROS in melanoma cells was detected using the Cellular Reactive Oxygen Species Detection Assay Kit (ab186027, Abcam,

Cambridge, UK), following the manufacturer's protocol. In brief, cells were plated (4000 cells per well) in a 96-well plate overnight. Subsequently, cells were treated with DMSO, or BAY-850 or erastin at the shown concentration for 48 h and then stained with ROS red stock solution for 1 h at room temperature. After incubation, Biotek Synergy MX Multi Format Microplate Reader (Biotek) was used to detect the intensity of the fluorescence at 520 nm (excitation) and 605 nm (emission) wavelengths. The relative fold change in ROS level was plotted in inhibitor-treated condition with respect to control DMSO-treated condition. Statistical analysis was performed using unpaired Student's $t$ tests in the GraphPad Prism 7 software.

## IL-6 measurement using ELISA

Melanoma cells were seeded in six-well tissue culture plates and treated with either BAY-850 (5 µM) or control DMSO for 48 h. The cell culture plates were centrifuged, and the supernatants is collected, and IL-6 level in the supernatant as measured using Human IL-6 ELISA Kit (ThermoFisher Scientific, EH2IL6), following the manufacturer's protocol.

## Cleavage under targets & release using nuclease (CUT&RUN) assay

CUT&RUN assays were performed with A375 cells using the CUT&RUN Assay Kit (Cat#86652; Cell Signaling Technology, Danvers, MA, USA) according to the manufacturer's instructions. Briefly, $2 \times 10^5$ cells at different conditions as shown in the figures were harvested, washed, bound to activated Concanavalin A-coated magnetic beads, and permeabilized. The bead–cell complexes were incubated overnight with the appropriate antibody as shown in the figures at 4 °C. Then, the complexes were washed three times, and the cells were resuspended in 100 µl pAG/MNase and incubated for 1 h at room temperature. The samples were then washed three times with digitonin buffer with protease inhibitors, resuspended in 150 µl digitonin buffer, and incubated for 5 min on ice. MNase was activated by adding calcium chloride, and the samples were incubated at 4 °C for 30 min. The reaction was stopped by adding 150 µl stop buffer, and the samples were incubated at 37 °C for 10 min to release the DNA fragments. The DNA was extracted using the DNA purification columns included in the CUT&RUN Assay Kit. qPCR was then performed using gene promoter-specific primers. Relative fold change was calculated as the ratio of immunoprecipitated DNA to IgG-precipitated DNA. The primer sequences and antibodies used for the CUT&RUN assays are listed in the Reagents and Tools Table.

## RNA preparation, complementary DNA (cDNA) preparation, reverse transcription (RT), and quantitative PCR (qPCR) analysis

Total RNA was extracted with TRIzol reagent (Invitrogen, Carlsbad, CA, USA) and purified with a RNeasy Mini Kit (Qiagen). According to the manufacturer's instructions, cDNA was synthesized using the M-MuLV First Strand cDNA Synthesis Kit (New England BioLabs, Ipswich, MA, USA). Then, qPCR was performed using gene-specific primers and the Power SYBR-Green Master Mix (Applied Biosystems, Foster City, CA, USA), per the manufacturer's instructions. The beta-actin (ACTB) gene served

as a control for normalization. The primer sequences for all genes analyzed in the study are provided in the Reagents and Tools Table.

## RNA sequencing and data analysis

Melanoma cell lines (A375 and SKMEL-103 cells) treated with BAY-850 (5 µM) and control DMSO for 48 h were used to prepare total RNA for gene-expression analysis on an Illumina HiSeq 2500 system. Total RNA was extracted using TRIzol® reagent (Invitrogen, Carlsbad, CA, USA) according to the manufacturer's instructions and purified on RNAeasy mini columns (Qiagen, Hilden, Germany) according to the manufacturer's instructions. Then, mRNA was purified from approximately 500 ng total RNA using oligo-dT beads and sheared by incubation at 94 °C. Following first-strand synthesis with random primers, second-strand synthesis was performed with dUTP to generate strand-specific libraries. The cDNA libraries were then end-repaired and A-tailed. Adapters were ligated, and second-strand digestion was performed using uracil-DNA-glycosylase. Indexed libraries that met appropriate cutoffs for both were measured by quantitative reverse transcription polymerase chain reaction (qRT-PCR) using a commercially available kit (KAPA Biosystems, Wilmington, MA, USA). The insert size distribution was determined using LabChip GX (PerkinElmer, Waltham, MA, USA) or an Agilent Bioanalyzer (Agilent Technologies, Santa Clara, CA, USA). Samples with a yield ≥0.5 ng/µL were used for sequencing on the Illumina HiSeq 2500 system (Illumina, San Diego, CA, USA). Images were converted into nucleotide sequences by the base-calling pipeline RTA 1.18.64.0 and stored in FASTQ format. For data analysis, the reads were first mapped to the human hg38 STAR (version 2.7.1a) with default parameters. Gene expression level was estimated using RSEM v1.3.3. Differentially expressed genes identified using DESeq2 v1.28 with default parameters. Genes are differentially expressed ($P$ value < 0.05) and were considered upregulated if the $P$ value < 0.05 and the $\log_2$ fold change >0 or downregulated if the $P$ value < 0.05 and the $\log_2$ fold change <0. The normalized gene expression data were used for downstream analyses, such as the heatmaps. To determine the function of the altered genes, KEGG was used for the pathway enrichment analyses. The RNA-seq data were submitted to Gene Expression Omnibus (GEO) (GEO accession number: GSE255437).

## Immunoblotting analysis

Whole-cell protein extracts were prepared using RIPA lysis buffer (Pierce) containing Protease Inhibitor Cocktail (Roche) and Phosphatase Inhibitor Cocktail (Sigma-Aldrich, St. Louis, MO). Lysed samples were centrifuged at 12,000 rpm for 15 min, and clarified supernatants were stored at −80 °C. Protein concentrations were determined using Bradford Protein Assay Reagent (Bio-Rad Laboratories, Hercules, CA, USA). Equal amounts of protein samples were electrophoresed on 10% or 12% sodium dodecyl sulfate (SDS) polyacrylamide gels and transferred onto polyvinylidene difluoride (PVDF) membranes (Millipore, Burlington, MA, USA) using a wet-transfer apparatus from Bio-Rad. The membranes were blocked with 5% skim milk and probed with primary antibodies in 5% BSA. After washing, the membranes were incubated with the appropriate horseradish peroxidase (HRP)-conjugated secondary antibodies (1:2000) (GE Healthcare Life

Sciences, Marlborough, MA, USA). The blots were developed using SuperSignal West Pico or Femto Chemiluminescent Substrate (Thermo Fisher Scientific). All antibodies used for immunoblotting are listed in the Reagents and Tools Table.

## Mouse subcutaneous-based tumorigenesis experiment with BAY-850 treatment

A375-MA2-F-Luc and SKMEL-103-F-Luc cells stably expressing firefly luciferase ($5 \times 10^6$) cells in 100 µl were mixed with 100 µl of Matrigel and were injected subcutaneously into 5–6-week-old NSG mice (stock No. 005557). When the tumor volumes reached ~80–100 mm³, the mice were treated with either vehicle (0.5% methyl cellulose in water) or BAY-850 (25 mg/kg body weight) intraperitoneally thrice a week until the end of the experimental period. Tumor volume was measured every week, and tumor size was calculated using the following formula: length × width² × 0.5 and plotted. In addition, The IVIS Spectrum in Vivo Imaging System was used to image mice (PerkinElmer, Waltham, MA, USA) every week. The total luminescence of tumor-bearing areas was measured using the Living Image in vivo imaging software (PerkinElmer). At the end of the experimental endpoint, the mice were sacrificed, and images of the lungs were captured using the IVIS Spectrum (PerkinElmer). Subcutaneous tumors from individual groups were harvested and imaged. All protocols for mouse experiments were approved by the Institutional Animal Care and Use Committee of the University of Alabama at Birmingham (UAB).

## Mouse subcutaneous-based tumorigenesis experiment with inhibitor treatment (combination)

A375 and SKMEL-103 cells ($5 \times 10^6$) in 100 µl were mixed with 100 µl of Matrigel and were injected subcutaneously into 5–6-week-old NSG mice (stock No. 005557). When the tumor volumes reached ~80–100 mm³, the mice were treated with either vehicle (0.5% methyl cellulose in water) or BAY-850 (25 mg/kg body weight) or Trametinib (0.5 mg/kg body weight) or combination of BAY-850 (25 mg/kg body weight) and Trametinib (0.5 mg/kg body weight) intraperitoneally thrice a week until the end of the experimental period. Tumor volume was measured every week, and tumor size was calculated using the following formula: length × width² × 0.5 and plotted. All protocols for mouse experiments were approved by the Institutional Animal Care and Use Committee of the University of Alabama at Birmingham (UAB).

## Mouse tumorigenesis experiment using melanoma PDX with inhibitor treatment

Melanoma PDXs (model IDs: TM01386; The Jackson Laboratory) were obtained from donor NSG, PDX-engrafted mice. PDXs were harvested after 6–8 weeks and implanted into 5–6-week-old NSG mice ($n = 3$) (stock no. 005557; The Jackson Laboratory). When the tumor volumes reached ~80–100 mm³, the mice were treated with either vehicle (0.5% methyl cellulose in water) or BAY-850 (25 mg/kg body weight) for the first set of the experiment. For the combination treatment, tumor volumes when reached ~80–100 mm³, the mice were treated with either vehicle (0.5% methyl cellulose in water) or BAY-850 (25 mg/kg body weight) or

Trametinib (0.5 mg/kg body weight) or combination of BAY-850 (25 mg/kg body weight) and Trametinib (0.5 mg/kg body weight) intraperitoneally thrice a week until the end of the experimental period. Tumor volume was measured every week, and tumor size was calculated using the following formula: length × width$^2$ × 0.5 and plotted. All protocols for mouse experiments were approved by the Institutional Animal Care and Use Committee of the University of Alabama at Birmingham (UAB).

## ATAD2 mRNA expression analysis of patient-derived melanoma patient samples

Gene expression in melanoma and normal skin samples was identified using cBioPortal and Gene Expression Profiling Interactive Analysis (GEPIA) (GEPIA; http://gepia.cancer-pku.cn) database. The Human Protein Atlas was used to analyze the protein expression of ATAD2 via IHC staining of both normal skin samples and melanoma patient samples. The Human Tissue Atlas Dataset was also used to show the three-year survival in high and low-ATAD2-expressing melanoma patient samples. The Talantov melanoma dataset (Talantov et al, 2005) was analyzed for ATAD2 expression in normal skin and melanoma patient samples and Riker melanoma dataset (Riker et al, 2008) was used for ATAD2 expression analysis in primary and metastatic melanoma patient samples using GEO2R and was plotted as boxplots using Prism. Statistical analysis was performed using unpaired Student's $t$ tests in the GraphPad Prism 7 software.

## Bioinformatics analysis of transcription factor binding on ATAD2 promoter

The 1 kb promoter sequence for ATAD2 was downloaded from the University of California, Santa Cruz (UCSC) genome browser (https://genome.ucsc.edu/cgi-bin/hgGateway). The promoter sequence was analyzed for identifying transcription factors regulating ATAD2 expression using PROMO 3.0 (https://alggen.lsi.upc.es/cgi-bin/promo_v3/promo/promoinit.cgi?dirDB=TF_8.3) software. Transcription factors predicted within a dissimilarity margin of 5% were shortlisted for further analysis.

## Bioinformatics analysis of previously published single-cell RNA-seq datasets (Rambow et al, Pozniak et al, and Karras et al)

The raw count matrix was downloaded following the instructions in the Data availability section of the paper. Raw count matrices were analyzed using R package Seurat V4.3.0. The raw count dataset was used to generate a Seurat object. The annotation information of the cells was added to the Seurat object. The gene signature score was added using the AddModuleScore function in the Seurat package. The VlnPlot2 function in the SeuratExtend package was used for the plotting. $P$ values were calculated using the Kruskal test.

## Statistical analysis

All experiments were conducted with at least three biological replicates. For all animal experiments, at least five replicates were used. The number of replicates is indicated in the figures and/or figure legends. Results for individual experiments quantitated are expressed as the mean ± standard error of the mean. For measurements of tumor growth in mice assays, values were first compared using the area under the curve method in the GraphPad Prism software, version 10.0, for Macintosh (GraphPad Software; https://www.graphpad.com), and $P$ values were calculated using two-tailed unpaired Student's $t$ tests in the GraphPad Prism software, version 10.0, for Macintosh. For the remaining experiments, $P$ values were calculated using two-tailed unpaired Student's $t$ tests in the GraphPad Prism software, version 10.0, for Macintosh. For GEPIA data, $P$ value was calculated using ANOVA (Analysis of Variance) for differential expression analysis between tumor and normal tissues. For RNAseq data, Quasi-Likelihood (QL) F-Tests were used for the statistical analysis. For analysis of previously published single-cell RNA-seq datasets (Rambow et al, Pozniak et al, and Karras et al), $P$ values were calculated using the Kruskal test.

## Data availability

RNA-Sequencing data of this study have been submitted to the National Center for Biotechnology Information (NCBI) Gene Expression Omnibus (GEO: Accession No. GSE255437). The link for this submission is https://www.ncbi.nlm.nih.gov/geo/query/acc.cgi?acc=GSE255437 and reviewer access code is mjubacgannodvax. All additional data discussed in the paper are available in the main text and expanded view.

The source data of this paper are collected in the following database record: biostudies:S-SCDT-10_1038-S44319-025-00660-w.

## Peer review information

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

## Acknowledgements

We gratefully acknowledge the following grants from the National Institutes of Health: R03CA230815 (RG), R03CA248913 (RG), R03CA292128 (RG), and R01CA233481 (RG), R03CA221926 (RG). We would also like to thank the UAB animal imaging facility.

## Author contributions

**Ashok Mari**: Data curation; Writing—original draft; Writing—review and editing. **Kevin Graciano**: Data curation; Writing—original draft; Writing—review and editing. **Raj Kumar**: Data curation; Writing—review and editing. **Emily Giles**: Data curation; Writing—review and editing. **Patrick T Ball**: Data curation; Writing—review and editing. **Revu V L Narayana**: Data curation; Writing—review and editing. **Romi Gupta**: Conceptualization; Supervision; Funding acquisition; Writing—original draft; Writing—review and editing.

Source data underlying figure panels in this paper may have individual authorship assigned. Where available, figure panel/source data authorship is

listed in the following database record: biostudies:S-SCDT-10_1038-S44319-025-00660-w.

## Disclosure and competing interests statement

The authors declare no competing interests.

# Expanded View Figures

**Figure EV1. ATAD2 is overexpressed in patient-derived melanoma samples and is regulated by MAPK pathways.**

(A) The Gene Expression Profiling Interactive Analysis (GEPIA) data was used for plotting mRNA expression of family IV of bromodomain-containing members. mRNA expression of BRPF1, BRPF2, BRPF3, BRD7, BRD9, and ATAD2b is shown in melanoma patient samples ($n = 461$) and normal skin samples ($n = 558$). (B, C) Human Protein Atlas showing ATAD2 protein expression intensity in different patient samples. (D) List of transcription factors with predicted DNA binding sites on the *ATAD1* promoter DNA sequence (1 kb upstream from the transcription start site) generated using PROMO search. (E) TCGA melanoma data showing the correlation between ATAD2 and E2F1 mRNA expression. (F) In the indicated melanoma cell lines ATAD2 mRNA expression was measured using RT-qPCR and plotted relative to primary human fibroblast cells. *ACTINB* was used as a normalization control (left) and ATAD2 protein expression in the shown cells was measured via western blotting. ACTINB was used as loading control (right). (G) The indicated cell lines were treated with 5μM of BAY-850 for 3 days, and viability was assessed by 3-(4,5-dimethylthiazol-2-yl)-2,5-diphenyltetrazolium bromide (MTT) assay. Relative cell viability in treated condition relative to DMSO-treated condition is presented, ns $P = 0.2244$, **$P = 0.0013$, ****$P = < 0.0001$, ****$P = < 0.0001$, from left to right. (A) $P$ value was calculated using ANOVA (Analysis of Variance) for differential expression analysis between tumor and normal tissues. (G) $P$ value was calculated using unpaired Student's $t$ test using three independent replicates.

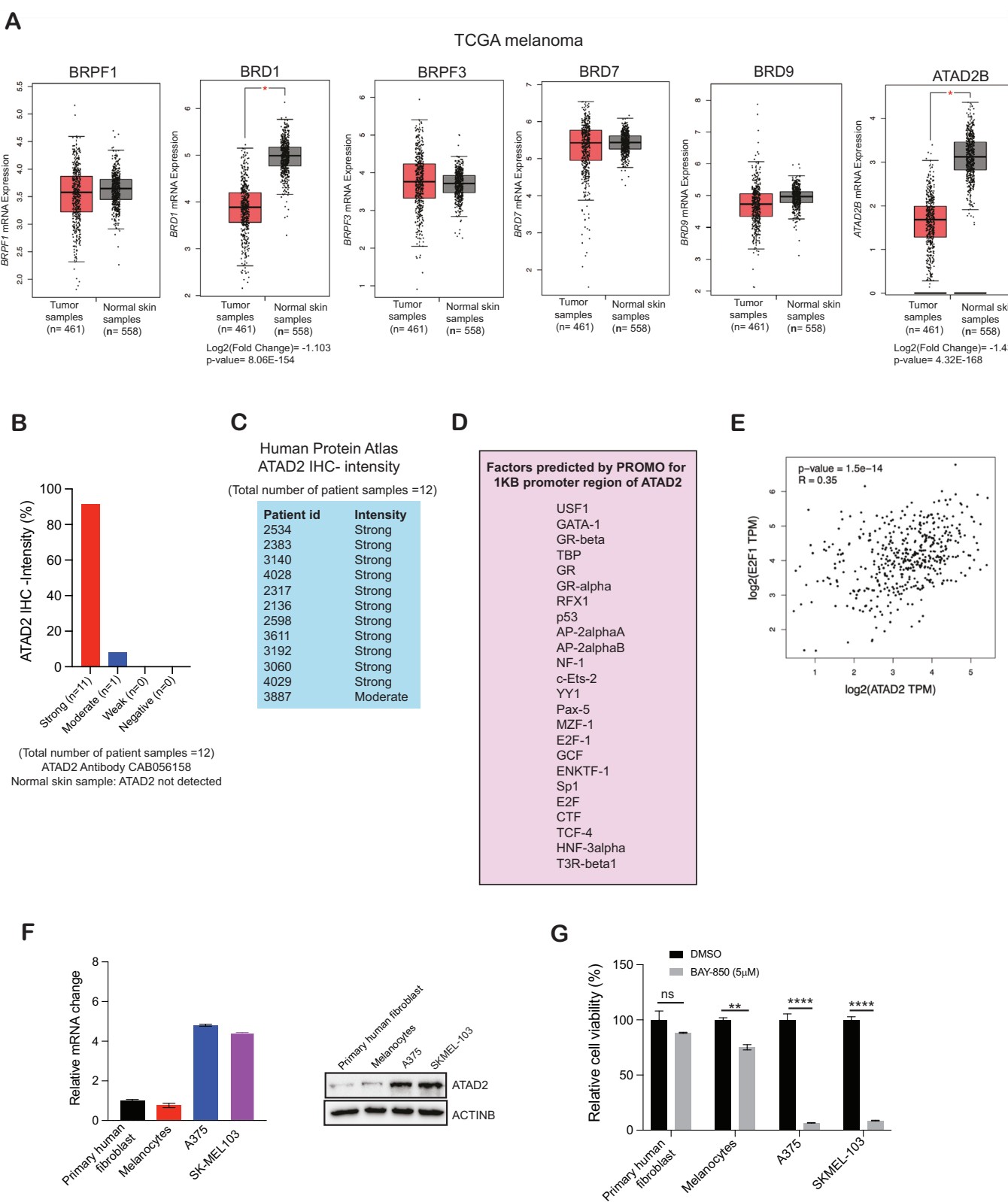

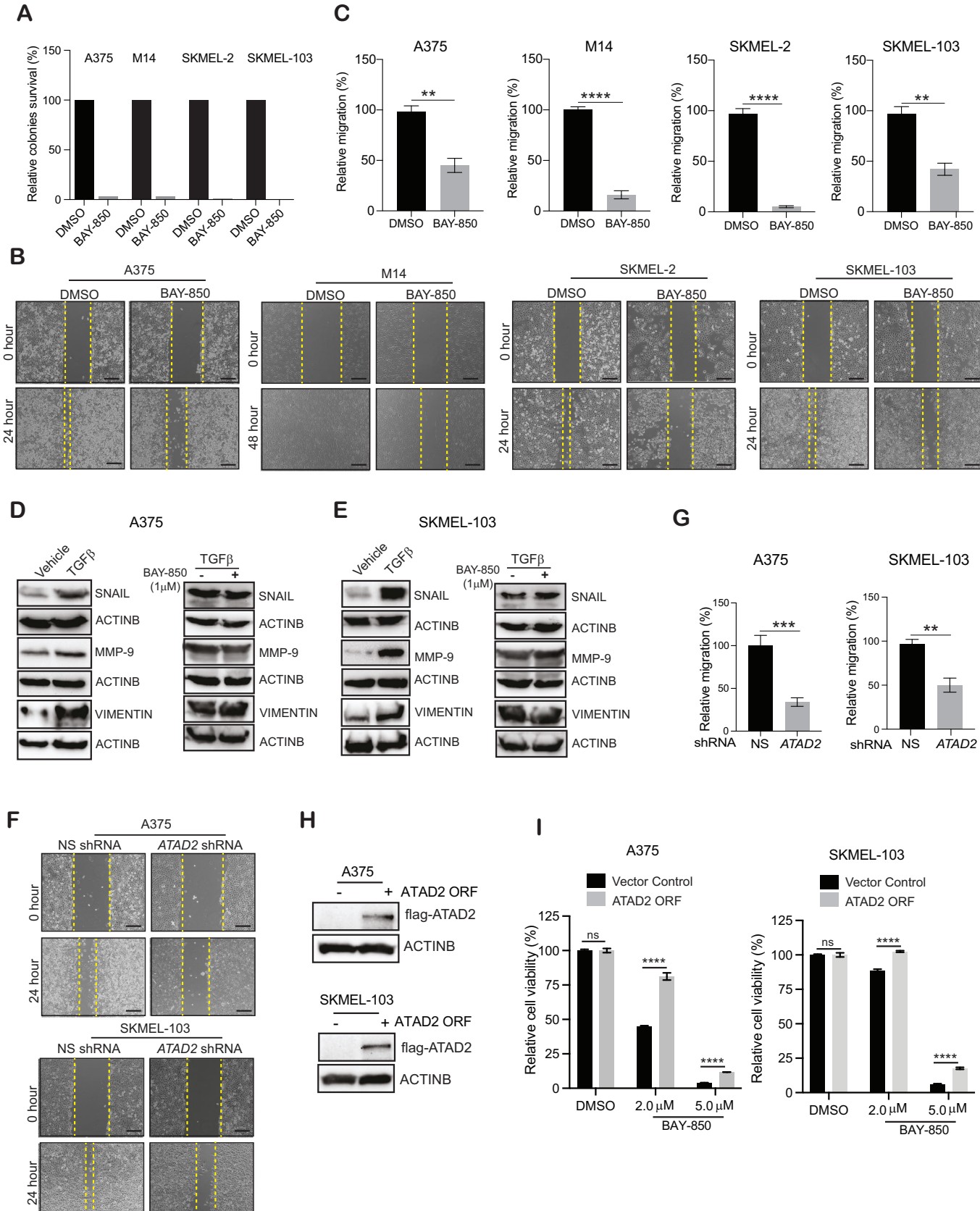

◀ **Figure EV2.   ATAD2 inhibition suppresses melanoma growth and progression.**

(**A**) The indicated melanoma cell lines were treated with 5 µM of BAY-850 for 2–4 weeks. Cell survival was measured using clonogenic assays. Relative percentage colony survival (%) shown in Fig. 2B is presented here. (**B**) Migration was analyzed in a wound-healing assay for melanoma cell lines treated with either DMSO or 5µM of BAY-850. Yellow dotted lines in the scratch assay images indicate the wound margins. Representative images are shown; scale bar, 200 µm. (**C**) Relative migration (%) in BAY-850 treated relative to the DMSO-treated cells is plotted for the experiment shown in (**B**), $**P = 0.0021$, $****P = < 0.0001$, $****P = < 0.0001$, $**P = 0.0040$, from left to right. (**D, E**) The indicated melanoma cell lines were serum starved for 12 h and were then exposed with Recombinant Human TGF-beta1 protein by adding to the media at concentration of 10 ng/mL for 48 h with or without 1 µM BAY-850 for 48 h. Indicated proteins were analyzed in shown condition for A375 and SKMEL-103 melanoma cell lines. ACTINB was used as the loading control. (**F**) Migration was analyzed in a wound-healing assay for melanoma cell lines expressing (NS) shRNAs or *ATAD2* shRNAs. Yellow dotted lines in the scratch assay images indicate the wound margins. Representative images are shown; scale bar, 200 µm. (**G**) Relative migration (%) in *ATAD2* shRNA expression cells relative to the control NS shRNA expressing cells shown in (**F**) is plotted. $***P = 0.0010$, $**P = 0.0076$, from left to right. (**H**) Indicated melanoma cell lines overexpressing either vector control or ATAD2-ORF were immune-blotted for the shown proteins confirming ATAD2 overexpression. (**I**) Indicated melanoma cell lines overexpressing either vector control or ATAD2-ORF were treated with BAY-850 at the shown concentration and cell viability was measured using 3-(4,5-dimethylthiazol-2-yl)-2,5-diphenyltetrazolium bromide (MTT) assay. Relative cell viability in the shown condition is presented, ns $P = > 0.9999$, $****P = < 0.0001$, $****P = < 0.0001$, ns $P = > 0.9999$, $****P = < 0.0001$, $****P = < 0.0001$, from left to right. (**B, F, I**) *P* value was calculated using unpaired Student's *t* test using three independent replicates.

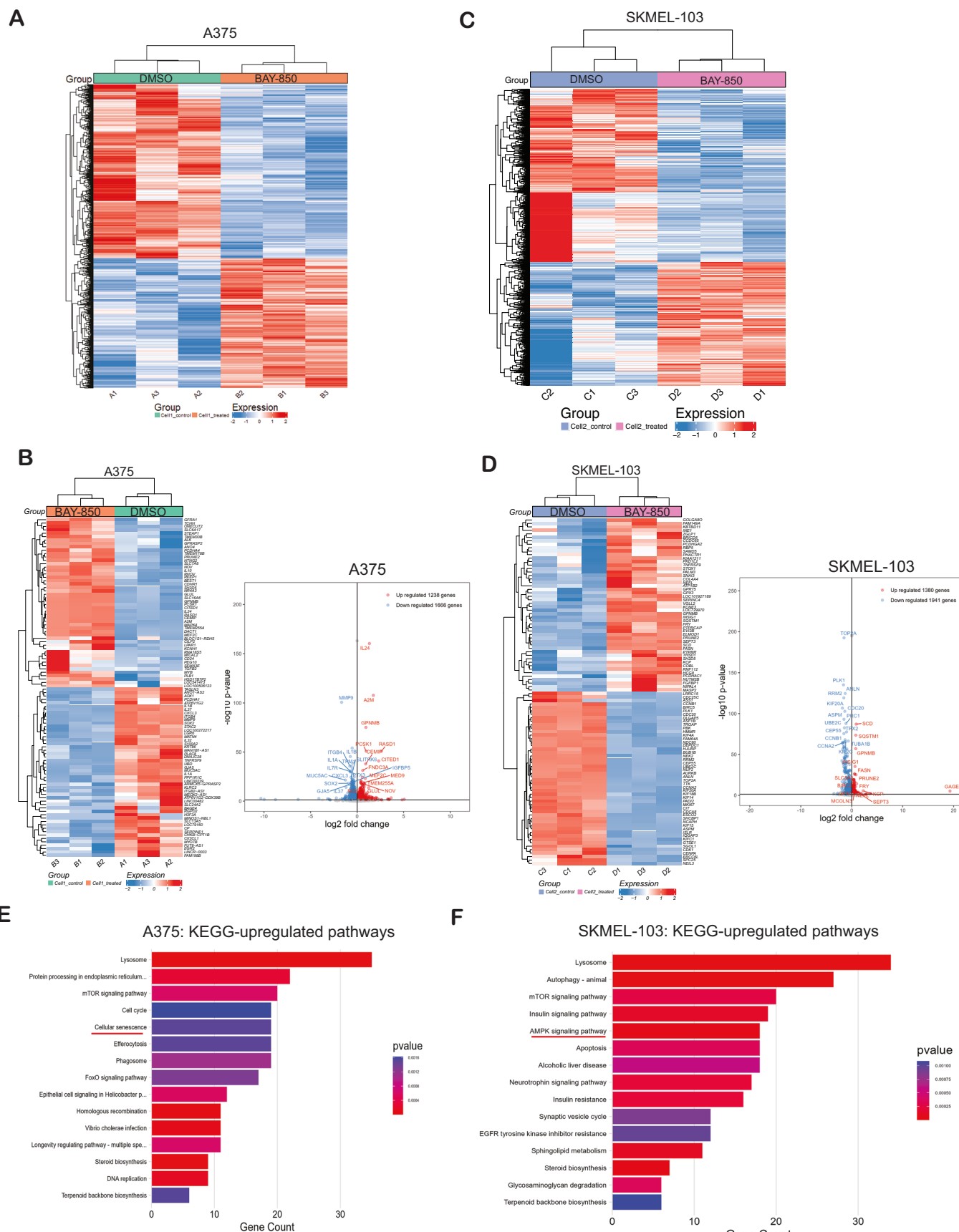

**Figure EV3. ATAD2 targeting activates tumor growth inhibitory pathways in melanoma cells.**

(A, C) Heatmap showing the genes that are upregulated and downregulated in A375 and SKMEL-103 cells upon treatment with BAY-850 (5 µM) for 48 h compared with the DMSO-treatment. (B, D) Heatmap showing top 50 upregulated and 50 downregulated genes (left) and volcano plot with top 15 upregulated and 15 downregulated genes (right) in A375 and SKMEL-103 cells upon BAY-850 treatment. (E, F) Pathways that were significantly upregulated upon BAY-850 treatment in A375 and SKMEL-103 cells based on gene expression changes was analyzed using KEGG pathway enrichment analysis and top 15 significantly altered pathway based on gene ratio and *P* values is presented. (A–D) *P* value was calculated using Quasi-Likelihood (QL) F-Tests.

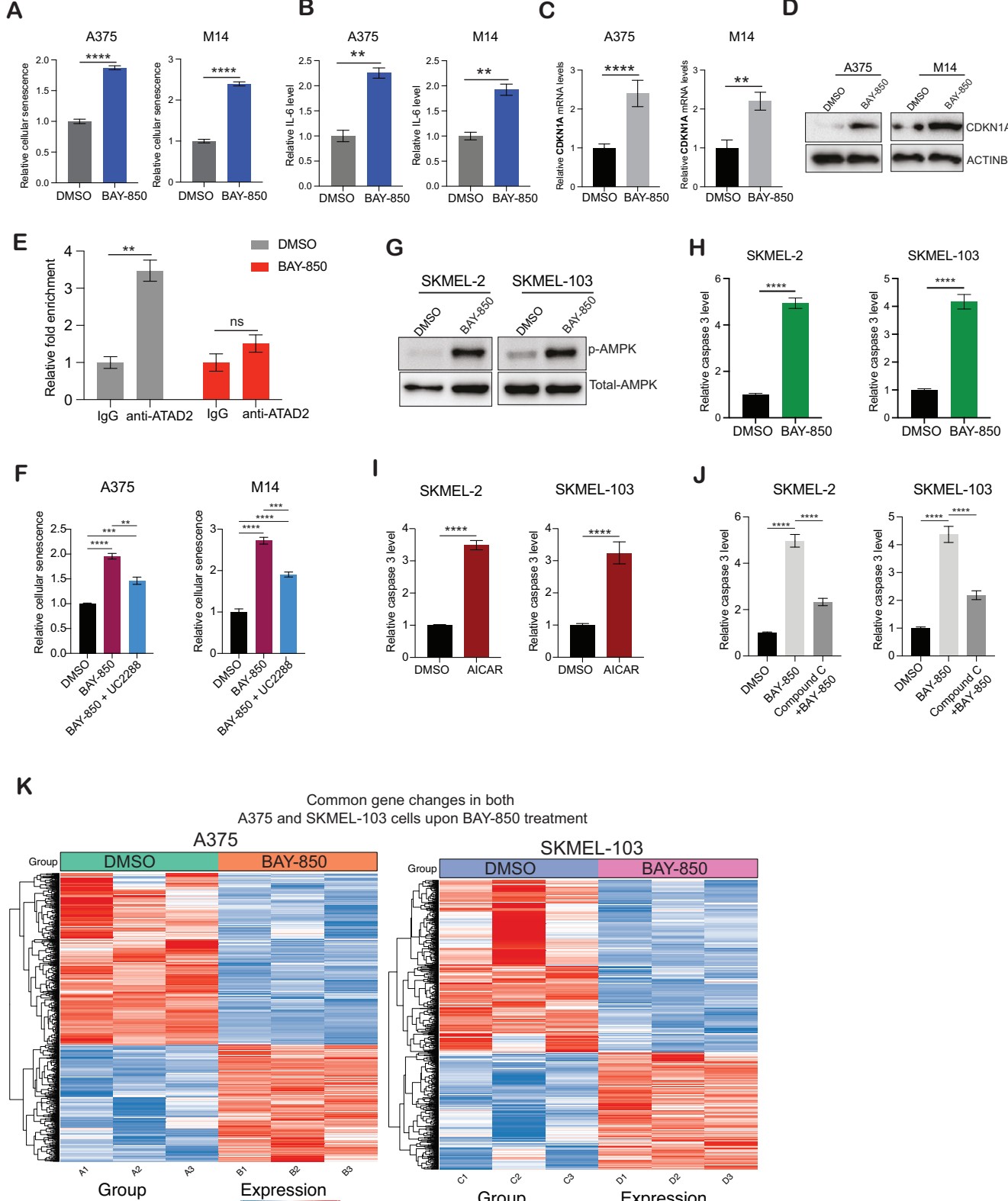

◀  **Figure EV4.   ATAD2 inhibits the growth of melanoma cells by regulating various tumor growth inhibitory pathways.**

(**A**) Senescence associated beta gal assay was performed in indicated melanoma cell lines upon treatment with either DMSO or 5 μM BAY-850 for 48 h. Relative cellular senescence in BAY-850 treated condition with respect to DMSO-treated condition is presented for the indicated melanoma cell lines. ****$P = <0.0001$, ****$P = <0.0001$, from left to right. (**B**) Senescence associated secretory factor IL-6 level was measured in indicated melanoma cell lines upon treatment with either DMSO or 5 μM BAY-850 for 48 h using ELISA based method. Relative IL6 level in BAY-850 treated condition with respect to DMSO treatment for each cell line is presented. **$P = 0.0012$, **$P = 0.0024$, from left to right. (**C, D**) The indicated melanoma cell lines (A375 and M14) were treated with either DMSO or ATAD2 inhibitor BAY-850 5 μM for 48 h. (**C**) *CDKN1A (p21)* mRNA expression was measured using RT-qPCR and plotted as the level in BAY-850-treated cells relative to that in DMSO-treated cells. *ACTINB* was used as a normalization control. ****$P = <0.0001$, **$P = 0.0018$, from left to right. (**D**) p21 protein expression was measured using western blot analysis under the indicated conditions. ACTINB was used as a loading control. (**E**) CUT-&-RUN analysis of ATAD2 binding on *CDKN1A* promoter in A375 treated with either DMSO or BAY-850 was performed. IgG was used as a negative control for CUT-&-RUN, and fold-enrichment plotted relative to IgG is shown. **$P = 0.0016$, ns $P = 0.1951$, from left to right. (**F**) The indicated melanoma cell lines were treated with either DMSO or 5 μM BAY-850 alone or 5 μM UC2288 and 5 μM BAY-850 in combination for 48 h, and cellular senescence was measured quantitatively using senescence-β-galactosidase activity assay kit (fluorescence, plate based # 25833; cell signaling) following the manufacturer's protocol and plotted. Relative cellular senescence in single and combination treatment condition with respect to DMSO control treated condition is plotted. ****$P = <0.0001$, ***$P = 0.0007$, **$P = 0.0017$, ****$P = <0.0001$, ****$P = <0.0001$, ***$P = 0.0002$, from left to right. (**G, H**) The indicated melanoma cell lines (SKMEL-2 and SKMEL-103) were treated with either DMSO or ATAD2 inhibitor BAY-850 5 μM for 48 h. (**G**) phospho-AMPK and total AMPK protein expression was measured using western blot analysis under the indicated conditions. (**H**) Apoptosis was measured under the indicated conditions, ****$P = <0.0001$, ****$P = <0.0001$, from left to right. (**I**) The indicated melanoma cell lines were treated with either DMSO or 1 mM AICAR for 48 h, and caspase 3 level was measured using Caspase 3 colorimetric assay kit (CASP3C, Sigma-Aldrich), following the manufacturer's protocol. Relative caspase 3 level in treatment condition with respect to DMSO control treated condition is plotted. ****$P = <0.0001$, ****$P = <0.0001$, from left to right. (**J**) The indicated melanoma cell lines were treated with either DMSO or 5 μM BAY-850 alone or 0.1 μM Compound C and 5 μM BAY-850 in combination for 48 h, and caspase 3 level was measured using Caspase 3 colorimetric assay kit (CASP3C, Sigma-Aldrich), following the manufacturer's protocol. Relative caspase 3 level in single and combination treatment condition with respect to DMSO control treated condition is plotted, ****$P = <0.0001$, ****$P = <0.0001$, ****$P = <0.0001$, ****$P = <0.0001$, from left to right. (**K**) Heatmap showing the common genes that are upregulated and downregulated in A375 and SKMEL-103 cells upon treatment with BAY-850 (5 μM) for 48 h compared with DMSO-treatment. (**A–C, E, F, H–J**) *P* value was calculated using unpaired Student's *t* test using three independent replicates.

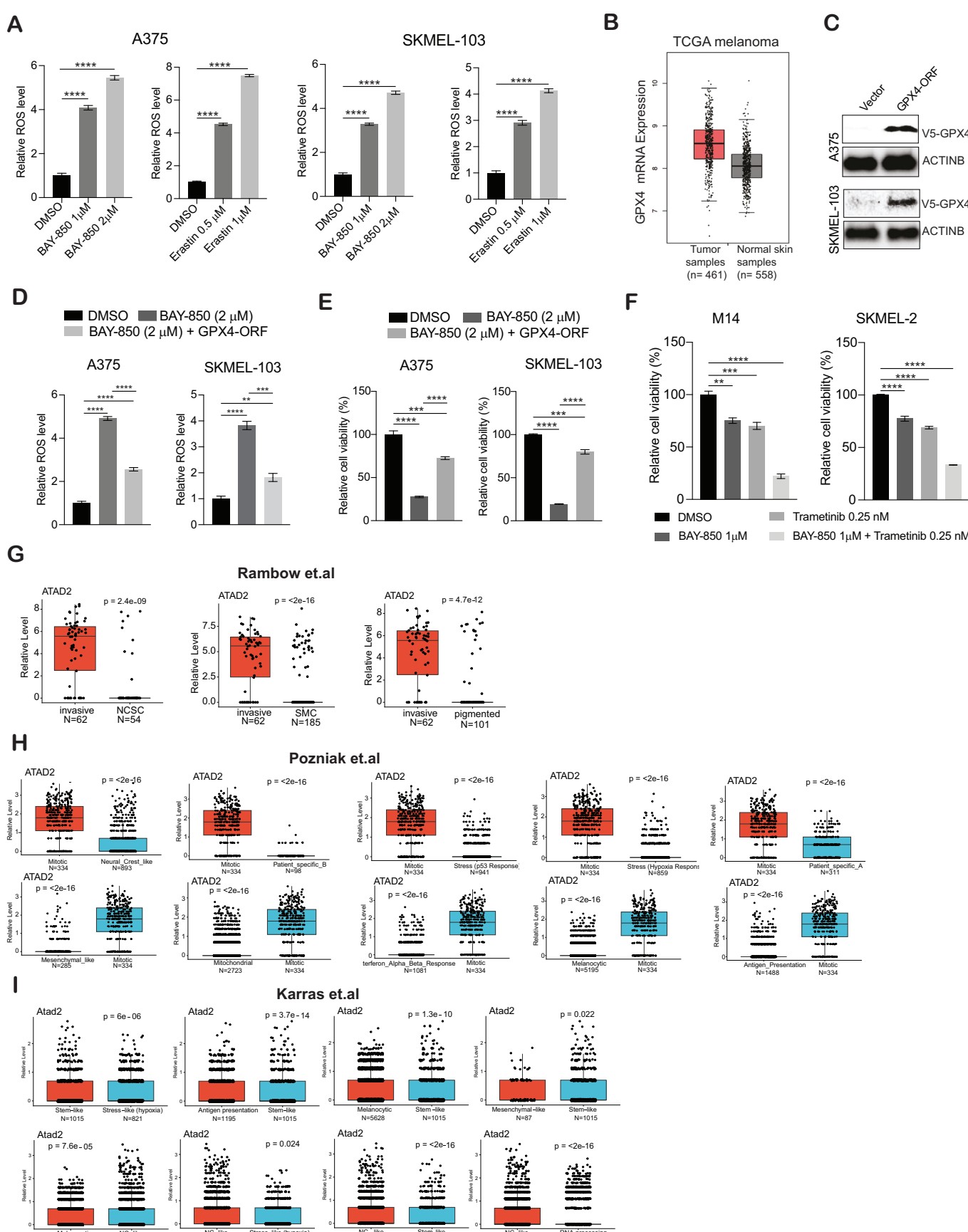

◀ **Figure EV5.  ATAD2 targeting induces ferroptosis in melanoma cells and GPX4 overexpression partially rescues ATAD2 targeting induced ferroptosis induction and cell viability.**

(A) The indicated melanoma cell lines were treated with either DMSO or BAY-850 or erastin at the shown concentration and cellular ROS levels was detected using the Cellular Reactive Oxygen Species Detection Assay Kit (ab186027, Abcam, Cambridge, UK), following the manufacturer's protocol. Relative ROS in BAY-850 or erastin treated condition or with respect to DMSO control treated condition is plotted. $****P = < 0.0001$, $****P = < 0.0001$, $****P = < 0.0001$, $****P = < 0.0001$, $****P = < 0.0001$, $****P = < 0.0001$, $****P = < 0.0001$, $****P = < 0.0001$, from left to right. (B) *GPX4* mRNA expression was plotted using Gene Expression Profiling Interactive Analysis (GEPIA). (C) Indicated melanoma cell lines overexpressing either vector control or GPX4-ORF were immune-blotted for the shown proteins confirming GPX4 overexpression. (D) Indicated melanoma cell lines overexpressing either vector control or GPX4-ORF were treated with BAY-850 at the shown concentration and cellular ROS levels was detected using the Cellular Reactive Oxygen Species Detection Assay Kit (ab186027, Abcam, Cambridge, UK), following the manufacturer's protocol. Relative ROS in the shown condition is plotted, $****P = < 0.0001$, $****P = < 0.0001$, $****P = < 0.0001$, $****P = < 0.0001$, $**P = 0.0046$, $***P = 0.0001$, from left to right. (E). Indicated melanoma cell lines overexpressing either vector control or GPX4-ORF were treated with BAY-850 at the shown concentration and cell viability was assessed by 3-(4,5-dimethylthiazol-2-yl)-2,5-diphenyltetrazolium bromide (MTT) assay. Relative percentage cell viability in the shown condition is presented, $****P = < 0.0001$, $***P = 0.0008$, $****P = < 0.0001$, $****P = < 0.0001$, $***P = 0.0004$, $****P = < 0.0001$, from left to right. (F) The indicated melanoma cell lines were treated with either DMSO or BAY-850 1 μM alone or trametinib 0.25 nM alone or combination of BAY-850 1 μM and trametinib 0.25 nM for 3 days, and viability was assessed by 3-(4,5-dimethylthiazol-2-yl)-2,5-diphenyltetrazolium bromide (MTT) assay. Relative percentage cell viability in the shown condition is presented, $**P = 0.0011$, $***P = 0.0007$, $****P = < 0.0001$, $****P = < 0.0001$, $****P = < 0.0001$, $****P = < 0.0001$, from left to right. (G–I) Analysis of the Rambow et al, (G), Pozniak et al, (H), and Karras et al, (I) datasets. N represents cell number. (A, D, E, F) *P* value was calculated using unpaired Student's *t* test using three independent replicates. (G–I) *P* value was calculated using Kruskal–Wallis test.

