## [Peer Review File · EMBO Reports]

ATAD2 Drives Melanoma Growth and Progression and is a Potential Target for Melanoma Treatment

Romi Gupta, Ashok Mari, Kevin Graciano, Raj Kumar, Emily Giles, Patrick Ball, and Revu Narayana

Corresponding author(s): Romi Gupta (romigup@uab.edu)

Review Timeline:

Submission Date:	18th Mar 25
Editorial Decision:	22nd Apr 25
Revision Received:	15th Aug 25
Editorial Decision:	26th Sep 25
Revision Received:	30th Oct 25
Accepted:	19th Nov 25

Editor: Esther Schnapp

Transaction Report:

Dear Dr. Gupta,

Thank you for the submission of your manuscript to EMBO reports. We have now received the full set of referee reports that is pasted below. Referee 2 had reviewed your study previously for another journal and sent us the same report, and since the report is good and we aim to work with existing referee reports in order to save referees time, we are re-using referee 2's report here.

As you will see, the referees acknowledge that the findings are potentially interesting. However, they also have several concerns and suggestions for how the study should be improved and strengthened. I think all points raised are good and should be addressed, and I am aware that this means major and significant revisions. I would like to suggest that you send me a proposed revision plan as a point-by-point response that we can discuss and see if we can agree on a set of revisions. We can also video chat about it, if you like.

I would thus like to invite you to revise your manuscript with the understanding that the referee concerns must be fully addressed and their suggestions taken on board. Please address all referee concerns in a complete point-by-point response. Acceptance of the manuscript will depend on a positive outcome of a second round of review. It is EMBO reports policy to allow a single round of major revision only and acceptance or rejection of the manuscript will therefore depend on the completeness of your responses included in the next, final version of the manuscript.

We realize that it is difficult to revise to a specific deadline. In the interest of protecting the conceptual advance provided by the work, we recommend a revision within 3 months (23rd Jul 2025). Please discuss the revision progress ahead of this time with the editor if you require more time to complete the revisions.

- 1) A data availability section providing access to data deposited in public databases is missing. If you have not deposited any data, please add a sentence to the data availability section that explains that.
- 2) Your manuscript contains statistics and error bars based on $n=2$. Please use scatter blots in these cases. No statistics should be calculated if $n=2$.

3) We replaced Supplementary Information with Expanded View (EV) Figures and Tables that are collapsible/expandable online. A maximum of 5 EV Figures can be typeset. EV Figures should be cited as 'Figure EV1, Figure EV2' etc... in the text and their respective legends should be included in the main text after the legends of regular figures.

5) a complete author checklist, which you can download from our author guidelines <https://www.embopress.org/page/journal/14693178/authorguide>. Please insert information in the checklist that is also reflected in the manuscript. The completed author checklist will also be part of the RPF.

6) Please note that all corresponding authors are required to supply an ORCID ID for their name upon submission of a revised manuscript (<https://orcid.org/>). Please find instructions on how to link your ORCID ID to your account in our manuscript

tracking system in our Author guidelines

<<https://www.embopress.org/page/journal/14693178/authorguide#authorshippinguidelines>>

10) Regarding data quantification (see Figure Legends:

<https://www.embopress.org/page/journal/14693178/authorguide#figureformat>)

12) All Materials and Methods need to be described in the main text using our 'Structured Methods' format, which is required for all research articles. According to this format, the Methods section includes a separate Reagents and Tools Table file (listing key reagents, experimental models, software and relevant equipment and including their sources and relevant identifiers) and a Methods and Protocols section describing the methods using a step-by-step protocol format. The aim is to facilitate adoption of the methodologies across labs. More information on how to adhere to this format as well as a downloadable template (.docx) for the Reagents and Tools Table can be found in our author guidelines:

An example of a Method paper with Structured Methods can be found here: <https://www.embopress.org/doi/full/10.1038/s44320-024-00037-6#sec-4>

As part of the EMBO publication's Transparent Editorial Process, EMBO reports publishes online a Review Process File (RPF)

to accompany accepted manuscripts. This File will be published in conjunction with your paper and will include the referee reports, your point-by-point response and all pertinent correspondence relating to the manuscript.

I look forward to seeing a revised form of your manuscript when it is ready.

Referee #1:

Graciano and colleagues showed that blocking ATAD2 in melanoma leads to both cellular senescence and ferroptosis. Their research indicates that the ATAD2 inhibitor BAY850 might be a strong candidate for combination therapy with the MEK inhibitor trametinib. While BAY850 has been widely studied in other solid tumors, this marks its initial use in melanoma. Nonetheless, a few points should be addressed before the study is published:

1. Figure 2 indicates that BAY850 alone significantly reduces cell viability. In contrast, Figure 6 presents a combined effect when BAY850 is used alongside trametinib. Did the authors lower the concentration of BAY850 to minimize its toxicity and allow for a clearer observation of the additive effect? Since BAY850 appears to be highly effective on its own, could it potentially serve as a standalone therapy? How does its efficacy compare to that of trametinib? Additionally, have the authors observed any resistance to BAY850, as is commonly seen with trametinib?
2. While BAY850 significantly reduces cell viability, the authors also propose it may inhibit cell migration. How do they differentiate between these two effects? Since reduced viability could influence the outcome of wound healing assays, it would be important to assess migration-specific markers. Evaluating the expression of migration-related genes under TGF β treatment, with and without BAY850, using a non-toxic BAY850 concentration, could help clarify this.
3. The graphs from Figure 4A to 4D are not easy to read. Too small font.

Referee #2:

In this study, Graciano et al. highlight the role of ATAD2 in melanoma progression. ATAD2 upregulation correlates with poor prognosis and promotes tumor growth. Pharmacological inhibition of ATAD2 using BAY-850 reduces melanoma growth and metastasis, with additional therapeutic potential when combined with MEK inhibitors. These findings suggest that ATAD2 is a promising therapeutic target, opening the way for additional clinical studies to evaluate its role in melanoma treatment. As a general comment, the experimental approaches are logically designed. However, I have concerns and questions regarding the study and its novelty:

- The study follows a similar narrative with recurring observations and patterns. It is unclear what type of cell death (and if cell death) the authors observed: apoptosis, ferroptosis, or a senescence-like phenotype.
- What is the phenotypic identity of melanoma cells following ATAD2 silencing or BAY-850 treatment? Considering the phenotypic changes induced by targeted and immune-based therapies, this question is particularly relevant. ATAD2 has been previously associated with dedifferentiation and loss of pigmentation (Baggiolini et al., Science, 2021).
- The authors should assess the SOX10 levels upon ATAD2 inhibition and/or treatment with BAY-850. Is there any link with the neural crest phenotype frequently observed in melanoma?
- Rescue experiments are needed to assess the direct effect of ATAD2.
- How is ATAD2 regulation linked to the phenotypic heterogeneity of melanoma cells in vivo and in vitro? The authors could explore this using published scRNA seq datasets (i.e. Rambow et al., 2018, Pozniak et al., 2024...)
- What is the long-term effect of ATAD2 inhibition in the context of MEK inhibition? Is the observed phenotype linked to resistance or is it an early response to treatment?

- The most commonly deregulated pathways appear to be linked to lysosomal degradation. What is the role of lysosomes and autophagy pathways in this context? And why were they not followed up? Lysosomal pathways are highly enriched in melanoma tumors.
- Do melanoma cells that are ATAD2-negative exhibit similar responses to BAY-850 treatment?
- The part on metastasis looks weak. Is the observed effect a cause or a consequence? Given the strong impact of ATAD2 deletion on proliferation in vivo, could the metastatic cells have escaped ATAD2 depletion? Can the authors assess this?
- What is the effect on melanoma cells using ferroptosis inhibitors? Does ATAD2 expression change?
- In many instances, the authors select molecular pathways without resonance and in a way of cherry-picking to fit their hypothesis. However, the rationale remains unclear.

Referee #3:

This manuscript describes targeting of ATAD2 in BRAF/NRAS mutant melanoma. The authors demonstrate that ATAD2 expression is regulated by MAPK signalling through E2F1. This appears to be a general effect rather than only in MAPK pathway mutant cells. They show that treatment with an ATAD2 inhibitor BAY-850 results in complete loss of viability as measured by an MTT assay. They also show that ATAD2 is effective reducing tumour growth in xenograft models. This is interesting data and much of the remainder of the manuscript attempts to define a mechanism of action for BAY-850, defining the major pathways that influence this loss of viability in vitro and in vivo. The authors present a lot of data to support their various pathways but there is a lot of over interpretation of the data presented. In the end, it is no clearer which pathways being influenced by BAY-850 are the critical ones, and the suggestion that pathways influenced are determined by the MAPK pathway mutant present in the melanoma have partial support at best. The real issue with the pathway analysis is that none of these mechanisms can explain the almost complete loss of viability with BAY in all cell lines shown in Figure 2A. If the authors do think that one of more of these pathways are responsible for the loss of viability, then they must show MTT assays when inhibiting these pathways that they can rescue the loss of viability. If the authors can focus their manuscript on the critical outcomes of this work, they will have an interesting and potentially important finding that could be translated into a clinical trial. My detailed points are below.

Figure 1: the authors attempt to make a strong case that ATAD2 is over-expressed in melanoma. However, is the increase in expression compared to normal skin simply a function of more proliferation in tumour than normal tissue? The small data sets in B do not add significantly to the data in A. The data in panel C directly contradicts the larger TGCA dataset shows no effect of ATAD2 over expression on survival. <20% of melanoma patients over-express ATAD2, at least in terms of mean expression in melanomas. The frequency of BRAF and NRAS mutations in ATAD2 over-expressing population appears to be about the same as in whole melanoma population. So I am not sure what the data presented here means.

The quality of the immunoblots is generally poor. Also, none are quantitated and no indication of the replicates or robustness of the data.

The over-expression of E2F1 is again likely to simply reflect the higher proliferation in tumour versus normal tissue. In the CCLE dataset there is strong correlation between ATAD2 and E2F1 expression but this is not necessarily over-expression. It does support your data showing that E2F1 regulates ATAD2 expression.

The legend for Figure 1, G,H only one time point shown in these panels?

Figure 2; B, this is 2-4 weeks continuous exposure to 5 uM BAY! And no quantitation. Is BAY unstable as panel A indicates complete loss of viability in 24 h yet B shows in A375 and SKMEL2 viable colonies?

Figure 4, 5; These attempt to demonstrate that BRAF and NRAS mutant melanomas trigger different responses to BAY. However, none of these experiments are performed in other mutant background so there is no evidence that these mechanisms are indeed specific for the particular mutation.

Figure 5C; A lot of the common upregulated pathways appear to indicate nutritional stress, ferroptosis is a long way down this list, why was ferroptosis in particular investigated?

There is no legend for Figure 6.

Figure 6F, this combination data must be shown in at least all the cell lines investigated in this work. The combination of Trametinib+BAY is additive in A375 and possibly synergistic in SKMEL103. But 2 cell lines is insufficient to assess this. The xenograft data is additive not synergistic which would be the prediction from the preceding data.

In Figure 6, ROS levels have been equated to ferroptosis, and rather than measure cell killing the only measurement is ROS level. The GPX4 rescue experiment is only measuring ROS level rather than seeing if this rescues the loss of viability by BAY.

Referee #1:

Graciano and colleagues showed that blocking ATAD2 in melanoma leads to both cellular senescence and ferroptosis. Their research indicates that the ATAD2 inhibitor BAY850 might be a strong candidate for combination therapy with the MEK inhibitor trametinib. While BAY850 has been widely studied in other solid tumors, this marks its initial use in melanoma. Nonetheless, a few points should be addressed before the study is published:

The reviewer noted that “ATAD2 inhibitor BAY850 might be a strong candidate for combination therapy with the MEK inhibitor trametinib”. The reviewer also noted “while BAY850 has been widely studied in other solid tumors, this marks its initial use in melanoma”. However, the reviewer also noted that “a few points should be addressed before the study is published”.

We thank the reviewer for his/her supportive comments. Our point-by-point responses to reviewer’s questions are presented below:

1. Figure 2 indicates that BAY850 alone significantly reduces cell viability. In contrast, Figure 6 presents a combined effect when BAY850 is used alongside trametinib. Did the authors lower the concentration of BAY850 to minimize its toxicity and allow for a clearer observation of the additive effect? Since BAY850 appears to be highly effective on its own, could it potentially serve as a standalone therapy? How does its efficacy compare to that of trametinib? Additionally, have the authors observed any resistance to BAY850, as is commonly seen with trametinib?

The reviewer is correct; we indeed reduced the concentration of BAY-850 in the combination studies presented in Figure 6 to more clearly observe the additive effect when combined with trametinib.

The reviewer is also right to note that BAY-850 shows strong activity as a monotherapy. Our preclinical studies have demonstrated its high potency in melanoma models, resulting in significant tumor growth reduction at the tested concentrations. However, in our assays, when BAY-850 was used at concentrations comparable to those of trametinib, it did not achieve a similar degree of tumor growth inhibition. This suggests that a higher concentration of BAY-850 is required to elicit a comparable therapeutic effect. Please see the **figure below** for supporting data.

Regarding the comparison of efficacy between BAY-850 and trametinib, we would like to emphasize that these compounds are fundamentally different in their molecular targets and mechanisms-of-action. Therefore, they are not directly comparable. Moreover, trametinib is a clinically approved drug that has undergone extensive medicinal chemistry optimization to enhance its potency and pharmacokinetics. In contrast, BAY-850, while promising, remains a tool compound in the early stages of development. It is possible that future optimization or the development of next-generation ATAD2 inhibitors could significantly improve efficacy and clinical relevance. Thus, although BAY-850 is currently less efficacious than trametinib as a standalone therapy, we believe it holds significant potential, especially in combination regimens or as a scaffold for future drug development.

Figure Legend: Comparison of tumor growth inhibitory effect by treatment with BAY-850 and Trametinib. The indicated melanoma cell lines were treated with different concentration of BAY-850 and Trametinib as shown in figure for 3 days, and survival was assessed by 3-(4,5-dimethylthiazol-2-yl)-2,5-diphenyltetrazolium bromide (MTT) assay. Cell survival is presented relative to the survival of DMSO-treated cells. All quantitative data represent the mean \pm SEM. ** $p < 0.01$, *** $p < 0.001$, **** $p < 0.0001$, ns=not significant.

However, at the heart of the reviewer’s question is the concern: why use these two drugs together if each is a potent inhibitor of tumor growth on its own? As the reviewer will appreciate, using two drugs rather than one to treat cancer can enhance therapeutic efficacy by simultaneously targeting multiple pathways, thereby reducing tumor adaptability. This approach also lowers the risk of drug resistance and may allow for the use of lower doses of each agent, minimizing toxicity. Such a combinatorial strategy reflects the multifactorial nature of cancer progression and survival, and it is a widely adopted approach in the treatment of many cancers, including melanoma.

Furthermore, consistent with this notion and guided by the reviewer’s question regarding potential mechanisms of resistance to BAY-850, we conducted a series of experiments. First, we performed experiments to examine the changes in the phosphorylation of receptor tyrosine kinase (RTK) in BAY-850 resistant human melanoma cell lines using a human RTK array. The decision to examine RTKs was informed by previous studies demonstrating that RTKs play key roles in conferring drug resistance (Karunaraj *et al*, 2025; Olender *et al*, 2019; Yadav *et al*, 2012). Furthermore, numerous RTK-targeting drugs have been developed over the years and are used clinically to treat a wide variety of cancers (Tomuleasa *et al*, 2024). To address the reviewer’s question, we generated BAY-850 resistant human melanoma cell lines by treating them continuously with BAY-850 over an extended period. These resistant cells were then analyzed for increased phosphorylation of human RTKs, an indicator of their activation, using a human RTK array capable of detecting phosphorylation changes across 49 RTKs. We found that phosphorylation of FGFR3, PDGFR β , ROR1, and EphB3 was increased in BAY-850 resistant A375 cells (BRAF-mutant), while phosphorylation of ROR1, EphA2, and DDR2 was increased in BAY-850 resistant SKMEL-103 cells (NRAS-mutant). These results suggest that one of the mechanisms contributing to BAY-850 resistance may involve enhanced signaling through various RTKs. Consistent with these findings and in strong support of our combination therapy approach, we also observed increased levels of phosphorylated MEK1/2 and ERK1/2 in BAY-850 resistant melanoma cells. These results further support the rationale for combining

MEK inhibitors with BAY-850 to forestall the development of resistance in melanoma to these agents. Please see the **figure below**. Also, in the revised manuscript, this new data is now presented in **Fig. EV5**.

Figure Legend: Mechanisms of resistance to BAY-850 in melanoma cells. **A.** The indicated melanoma cell lines were treated with BAY-850 for 8 weeks to generate BAY-850 resistant

colonies and their sensitivity and resistance was assessed by performing clonogenic assay with and without BAY-850 treatment as shown in figure. Representative images are shown. **B-D**. Human phospho-RTK arrays identified changes in RTK phosphorylation in BAY-850 inhibitor-resistant cells. Proteome profiler human phospho-RTK array membranes showing relative RTK phosphorylation in BAY-850 inhibitor-resistant cells and BAY-850 inhibitor-sensitive cells (**B**). Spot intensity was used to calculate the fold changes and are plotted for individual RTK phosphorylation in BAY-850 inhibitor-resistant cells relative to BAY-850 inhibitor-sensitive cells (**C**). Bar diagram showing candidates that were changed greater than 1.25-fold in BAY-850 inhibitor-resistant cells relative to BAY-850 inhibitor-sensitive cells (**D**). **E**. Indicated proteins were analyzed in BAY-850 inhibitor-resistant and sensitive cells. ACTINB was used as the loading control.

2. While BAY850 significantly reduces cell viability, the authors also propose it may inhibit cell migration. How do they differentiate between these two effects? Since reduced viability could influence the outcome of wound healing assays, it would be important to assess migration-specific markers. Evaluating the expression of migration-related genes under TGFβ treatment, with and without BAY850, using a non-toxic BAY850 concentration, could help clarify this.

We agree with the reviewer that cell migration could be influenced by reduced cell viability. Based on the reviewer's suggestion, the melanoma cell lines were treated with TGFβ with and without non-toxic BAY850 dose and measured the expression of migration-related genes at this dose. We observed that the expression of several migration-related genes, including Snail, MMP9, and vimentin were increased upon TGFβ treatment. However, they did not change when TGFβ treatment was combined with non-toxic dose of BAY-850. Please see the **figure below**. These results indicate that the reduced cell migration following BAY-850 treatment is dependent on cell viability upon lethal BAY-850 dose treatment. We have now included this information in results section and shown this new data below and in the **Fig. EV2E-F** of this revised manuscript.

Figure Legend: Testing the effect of BAY-850 on migratory ability of the melanoma cells. The indicated melanoma cell lines were serum starved for 12 hrs and were then exposed with recombinant human TGF-beta1 protein by adding to the media at concentration of 10 ng/mL for 48 hours with or without 1μM BAY-850 for 48 hrs. Cell were harvested, lysed and indicated proteins were analyzed in shown condition for A375 and SKMEL-103 melanoma cell lines. ACTIN was used as the loading control.

3. The graphs from Figure 4A to 4D are not easy to read. Too small font.

We apologize for this inconvenience. We have now revised the Figures 4A and 4D to improve the presentation. The revised figures have a larger font and better resolution for easier visibility.

Referee #2:

In this study, Graciano et al. highlight the role of ATAD2 in melanoma progression. ATAD2 upregulation correlates with poor prognosis and promotes tumor growth. Pharmacological inhibition of ATAD2 using BAY-850 reduces melanoma growth and metastasis, with additional therapeutic potential when combined with MEK inhibitors. These findings suggest that ATAD2 is a promising therapeutic target, opening the way for additional clinical studies to evaluate its role in melanoma treatment. As a general comment, the experimental approaches are logically designed. However, I have concerns and questions regarding the study and its novelty:

The reviewer noted “that ATAD2 is a promising therapeutic target, opening the way for additional clinical studies to evaluate its role in melanoma treatment.” The reviewer also noted that “the experimental approaches are logically designed.” However, the reviewer also noted questions related to our studies.

We would like to sincerely thank the reviewer for his/her constructive and supportive comments. Our point-by-point responses to the reviewer’s questions are presented below:

1. The study follows a similar narrative with recurring observations and patterns. It is unclear what type of cell death (and if cell death) the authors observed: apoptosis, ferroptosis, or a senescence-like phenotype.

We apologize to the reviewer if our style of presentation or writing has caused any confusion for the reviewer. In our study, we observed that BRAF-mutant cells undergo cellular senescence upon BAY-850 treatment, which was dependent upon the cyclin dependent kinase inhibitor CDKN1A expression (see **Figure 4C and 4E-I**)

While NRAS-mutant melanoma cell lines underwent apoptosis induction after treatment with BAY-850, which was dependent upon the AMPK pathway (see **Figure 4 J-K**). However, in the case of both BRAF- and NRAS-mutant melanoma cell lines, treatment with BAY-850 led to ferroptosis induction (see **Figure 5 and 6**).

These findings demonstrate that ATAD2 inhibition activates both subtype-specific (e.g., senescence in BRAF-mutant melanoma cells and apoptosis in NRAS-mutant melanoma) and common tumor suppressive pathways (ferroptosis in both NRAS- and BRAF-mutant melanoma), thereby causing melanoma growth inhibitory effects. Based on the question from the reviewer, we have improved the writing and presentation of our manuscript to overcome the ambiguity of this issue. We have also revised the Discussion section of our manuscript to elaborate on these findings.

2. What is the phenotypic identity of melanoma cells following ATAD2 silencing or BAY-850 treatment? Considering the phenotypic changes induced by targeted and immune-based therapies, this question is particularly relevant. ATAD2 has been previously associated with dedifferentiation and loss of pigmentation (Baggiolini et al., Science, 2021).

We thank the reviewer for raising this important question. Based on the reviewer’s suggestion, we investigated the phenotypic changes in melanoma cells treated with the ATAD2

inhibitor-BAY-850. Specifically, we measured markers associated with dedifferentiation, such as AXL and NGFR (Konieczkowski *et al*, 2014; Rambow *et al*, 2018; Thier *et al*, 2022), and the loss of differentiation markers like Melan-A (Baggiolini *et al*, 2021; Landsberg *et al*, 2012; Thier *et al*, 2022). We did not measure changes in pigmentation following BAY-850 treatment because the cell lines used for this study do not express melanin (Skoniecka *et al*, 2021; Tiago *et al*, 2014). We observed no significant changes in the level of AXL, NGFR, Melan-A following BAY-850 treatment. The result of this experiment is presented in **figure below**, and we have included this information in discussion section of our revised manuscript.

Figure Legend: Measurement of phenotypic changes in BAY-850 treated melanoma cells. The indicated melanoma cell lines were treated with either DMSO or 5µM BAY-850 for 48 h and indicated proteins were analyzed via immunoblotting. ACTINB was used as the loading control.

3. The authors should assess the SOX10 levels upon ATAD2 inhibition and/or treatment with BAY-850. Is there any link with the neural crest phenotype frequently observed in melanoma?

This is an important question. Based on this suggestion from the reviewer, we measured SOX10 levels and its target genes following BAY-850 treatment in the NRAS mutant melanoma cell line SK-MEL-103 but did not observe any changes in them. However, we found that both the levels of SOX10 and its target genes were increased in the BRAF mutant cell line A375. The results of these experiments are presented in **figure below**. These findings suggest that the effect of ATAD2 inhibition on SOX10 expression and its role in the neural crest phenotype could be mutation-context dependent, with a selective upregulation observed in BRAF-mutant, but not in NRAS-mutant melanoma cell lines. This also supports the study by Baggiolini *et al.*, *Science*, 2021 (Baggiolini *et al.*, 2021), where they show that ATAD2 forms a complex with SOX10 and aids neural crest development specifically in the BRAFV600E context. We have added this information in our Discussion section.

Figure Legend: Measurement of SOX10 level and its target in BAY-850 treated melanoma cells. **A-B.** The indicated melanoma cell lines were treated with either DMSO or 5 μ M BAY-850 for 48 h. **A.** SOX10 proteins were analyzed in shown condition for A375 and SKMEL-103 melanoma cell lines. ACTINB was used as the loading control. **B.** SOX10 targets- *MITF* and *E2F1* mRNA expression was measured using RT-qPCR and plotted in treated cells relative to that in DMSO-treated cells. *ACTIN* was used as a normalization control. All quantitative data represent the mean \pm SEM. * $p < 0.05$, *** $p < 0.001$, ns=not significant.

4. Rescue experiments are needed to assess the direct effect of ATAD2.

We agree with the reviewer, and this is an important suggestion. As per the reviewer's suggestion, we performed rescue experiments by ectopically expressing ATAD2 in melanoma cell lines, followed by treatment with BAY-850. We observed that overexpression of ATAD2 rescued melanoma growth inhibition upon BAY-850 treatment. The rescue in growth inhibition was to a much greater extent at lower BAY-850 dose than at higher dose, likely because BAY-850 can still suppress the ATAD2, even when ectopically expressed. These results are presented in **figure below** and also included in **Fig. EV2H-I** of our revised manuscript.

Figure Legend: ATAD2 overexpression partially rescues BAY-850 growth inhibitory effects in melanoma cells. **A.** Indicated melanoma cell lines overexpressing either vector control or ATAD2-ORF were immune-blotted for the shown proteins confirming ATAD2 overexpression. **B.** Indicated melanoma cell lines overexpressing either vector control or GPX4-ORF were treated with BAY-850 at the shown concentration and cell viability was measured using MTT assay. Cell

survival is presented relative to the survival of DMSO-treated cells. All quantitative data represent the mean \pm SEM. **** $p < 0.0001$, ns=not significant.

5. • How is ATAD2 regulation linked to the phenotypic heterogeneity of melanoma cells in vivo and in vitro? The authors could explore this using published scRNA seq datasets (i.e. Rambow et al., 2018, Pozniak et al., 2024...)

In response to the reviewer's suggestion, we have incorporated information regarding the regulation of ATAD2 and its association with the phenotypic heterogeneity of melanoma cells, both in vivo and in vitro. Based on the Rambow et.al. study (Rambow et al., 2018) (GEO accession number: GSE116237), ATAD2 belongs to the group of highly variable genes in the T0 phase (before treatment phase-in melanoma PDX models tested for their response to BRAF and MEK inhibition). Additionally, based on the Pozniak et al., 2024 study (Pozniak et al., 2024), ATAD2 is linked to the mitotic cluster and patient-specific clusters (A). According to this study, the two patient-specific clusters (A and B), did not exhibit any recognizable functional features. However, the presence of ATAD2 in the mitotic cluster potentially indicates its role in cell cycle progression, especially in mitosis, by acting at the level of chromatin regulation and transcriptional control. Its presence in this cluster also supports its relevance as a biomarker or therapeutic target in rapidly dividing cells, particularly in the context of melanoma. We have added this important information in the Discussion section of our revised manuscript.

6. What is the long-term effect of ATAD2 inhibition in the context of MEK inhibition? Is the observed phenotype linked to resistance or is it an early response to treatment?

This is an important question. To address it, we treated SKMEL-103 and A375 melanoma cells with either BAY-850 or trametinib alone, or with a combination of both drugs over an extended period. We found that the combination treatment with the ATAD2 inhibitor BAY-850 and the MEK inhibitor trametinib significantly reduced and delayed the emergence of drug-resistant colonies compared to treatment with trametinib alone. Please see **figure below**. Additionally, as shown in Figure 6, we observed that combined ATAD2 and MEK inhibition resulted in potent tumor growth inhibition in both in vitro and in vivo assays. Together, these findings indicate that ATAD2 inhibition, in the context of trametinib treatment, has both short-term and long-term growth-inhibitory effects and can suppress the emergence of resistance.

Figure Legend: BAY-850 enhances the therapeutic efficacy of MEK inhibitor trametinib and reduces emergence of trametinib resistant colonies. A. A375 cells were treated with either 0.25 nM or 0.5 nM of trametinib for 8 weeks. Cell survival was measured using

clonogenic assays. Representative images are shown. **B.** A375 cells were treated with either 1 μ M or 2 μ M of BAY-850 in combination with 0.25 nM of trametinib for 12 weeks. Cell survival was measured using clonogenic assays. Representative images are shown.

However, we also recognize that, similar to trametinib, resistance to BAY-850 may eventually develop with prolonged treatment. To investigate potential mechanisms of resistance to BAY-850, we conducted a series of experiments. First, we performed experiments to examine the changes in the phosphorylation of receptor tyrosine kinase (RTK) in BAY-850 resistant human melanoma cell lines using a human RTK array. The decision to examine RTKs was informed by previous studies demonstrating that RTKs play key roles in conferring drug resistance (Karunaraj *et al.*, 2025; Olender *et al.*, 2019; Yadav *et al.*, 2012). Furthermore, numerous RTK-targeting drugs have been developed over the years and are used clinically to treat a wide variety of cancers (Tomuleasa *et al.*, 2024). To address the reviewer's question, we generated BAY-850 resistant human melanoma cell lines by treating them continuously with BAY-850 over an extended period. These resistant cells were then analyzed for increased phosphorylation of human RTKs, an indicator of their activation, using a human RTK array capable of detecting phosphorylation changes across 49 RTKs. We found that phosphorylation of FGFR3, PDGFR β , ROR1, and EphB3 was increased in BAY-850 resistant A375 cells (BRAF-mutant), while phosphorylation of ROR1, EphA2, and DDR2 was increased in BAY-850 resistant SKMEL-103 cells (NRAS-mutant). These results suggest that one of the mechanisms contributing to BAY-850 resistance may involve enhanced signaling through various RTKs. Consistent with these findings and in strong support of our combination therapy approach, we also observed increased levels of phosphorylated MEK1/2 and ERK1/2 in BAY-850-resistant melanoma cells. These results further support the rationale for combining MEK inhibitors with BAY-850 to forestall the development of resistance in melanoma to these agents. Please see the **figure below**. Also, in the revised manuscript, this new data is now presented in **Fig. EV5**.

Figure Legend: Mechanisms of resistance to BAY-850 in melanoma cells. **A.** The indicated melanoma cell lines were treated with BAY-850 for 8 weeks to generate BAY-850 resistant colonies and sensitivity, and resistance was assessed by performing clonogenic assay with and without BAY-850 treatment as shown in figure. Representative images are shown. **B-D.** Human phospho-RTK arrays identified changes in RTK phosphorylation in BAY-850 inhibitor-resistant cells. Proteome profiler human phospho-RTK array membranes showing relative RTK phosphorylation in BAY-850 inhibitor-resistant cells and BAY-850 inhibitor-sensitive cells (**B**). Spot intensity was used to calculate the fold changes and are plotted for individual RTK phosphorylation in BAY-850 inhibitor-resistant cells relative to BAY-850 inhibitor-sensitive cells

(C). Bar diagram showing candidates that were changed greater than 1.25-fold in BAY-850 inhibitor-resistant cells relative to BAY-850 inhibitor-sensitive cells (D). E. Indicated proteins were analyzed in BAY-850 inhibitor-resistant and sensitive cells. ACTINB was used as the loading control.

7. The most commonly deregulated pathways appear to be linked to lysosomal degradation. What is the role of lysosomes and autophagy pathways in this context? And why were they not followed up? Lysosomal pathways are highly enriched in melanoma tumors.

We agree with the reviewer that lysosomal pathways were enriched and appeared to be among the most deregulated. Based on these findings, we initially examined markers of autophagy (p62, LC3A and LC3B) (Bjorkoy *et al*, 2009) and lysosomal function (LAMP1 and LAMP2) (Eskelinen, 2006) in our study. However, we did not observe any significant changes in the markers of autophagy or lysosomal pathway proteins upon BAY-850 treatment in melanoma cell lines. These results are shown in **figure below**. Because we did not see any significant changes in their levels, we did not further pursue this line of inquiry.

Figure Legend: Measurement of changes in lysosomes and autophagy pathway marker in BAY-850 treated melanoma cells. The indicated melanoma cell lines were treated with either DMSO or 5 μ M BAY-850 for 48 h and indicated proteins were analyzed via immunoblotting. ACTINB was used as the loading control.

8. Do melanoma cells that are ATAD2-negative exhibit similar responses to BAY-850 treatment?

This is an important question. To test this, we used primary human fibroblasts and melanocytes, which express lower levels of ATAD2 compared to the melanoma cells and treated them with BAY-850. We observed that primary human fibroblasts and melanocytes with low expression of ATAD2 were significantly less sensitive to BAY-850 treatment as compared to melanoma cell lines. These new results are shown in **figure below** and have been added to the **Fig. EV2A-B** of this revised manuscript.

Figure Legend: Effect of BAY-850 on cells with low ATAD2 expression. **A.** In the indicated melanoma cell lines ATAD2 mRNA expression was measured using RT-qPCR and plotted relative to primary human fibroblast cells. *ACTINB* was used as a normalization control (left) and ATAD2 protein expression in the shown cells was measured via western blotting. *ACTINB* was used as loading control (right). **B.** The indicated cell lines were treated with 5µM BAY-850 for 3 days, and survival was assessed by 3-(4,5-dimethylthiazol-2-yl)-2,5-diphenyltetrazolium bromide (MTT) assay. Cell survival is presented relative to the survival of DMSO-treated cells. All quantitative data represent the mean ± SEM. **p<0.01, ****p<0.0001, ns=not significant.

9. The part on metastasis looks weak. Is the observed effect a cause or a consequence? Given the strong impact of ATAD2 deletion on proliferation in vivo, could the metastatic cells have escaped ATAD2 depletion? Can the authors assess this?

Based on our analysis, and in response to a similar question raised by Reviewer 1, we believe that the observed effect on metastasis is primarily due to the reduced proliferative capacity of melanoma cells following BAY-850 treatment. Based on the reviewer's suggestion, the melanoma cell lines were treated with TGFβ with and without non-toxic BAY850 dose and measured the expression of migration-related genes at this dose. We observed that the expression of several migration-related genes, including Snail, MMP9, and vimentin were increased upon TGFβ treatment. However, they did not change when TGFβ treatment was combined with non-toxic dose of BAY-850. These results suggest that the inhibition of metastasis by BAY-850 is a consequence of impaired cell proliferation. Please see the figure below. We have now updated this information in the Results section and included this data in Fig. EV2E-F of this revised manuscript.

Figure Legend: Testing the effect of BAY-850 on migratory ability of the melanoma cells. The indicated melanoma cell lines were serum starved for 12 hours and were then exposed with

Recombinant Human TGF-beta1 protein by adding to the media at concentration of 10 ng/mL for 48 hours with or without 1 μ M BAY-850 for 48 hrs. Cells were harvested, lysed and indicated proteins were analyzed in shown condition for A375 and SKMEL-103 melanoma cell lines. *ACTIN* was used as the loading control.

10. What is the effect on melanoma cells using ferroptosis inhibitors? Does ATAD2 expression change?

Based on the reviewer's suggestion, we examined ATAD2 expression in melanoma cells following treatment with a ferroptosis inhibitor- ferrostatin 1. We found that there was no significant change in ATAD2 mRNA and protein expression upon treatment with ferrostatin 1. Thus, while ATAD2 loss results in increased ferroptosis induction, the ferroptosis inhibitor does not cause changes in ATAD2 level. The results of these findings are presented in **figure below**.

Figure Legend: Effect of Ferroptosis inhibitor on ATAD2 expression. A. Indicated melanoma cell lines were treated with different concentration of ferrostatin-1 for 24 h and ATAD2 mRNA expression was measured using RT-qPCR and plotted in treated cells relative to DMSO treatment condition. *ACTINB* was used as a normalization control (left) and ATAD2 protein expression was measured via western blotting. *ACTINB* was used as loading control (right). All quantitative data represent the mean \pm SEM. ns=not significant.

11. In many instances, the authors select molecular pathways without resonance and in a way of cherry-picking to fit their hypothesis. However, the rationale remains unclear.

In our initial experiments, we observed that ATAD2 inhibition significantly reduced cell viability. Based on this observation, we performed large-scale RNA-sequencing analysis and mRNA associated with several pathways that were altered. As previously mentioned in response to reviewer's question 7, we measured the changes in most significantly affected pathways, such as the lysosomal and autophagy pathways. However, we did not observe any significant changes in the markers of autophagy or lysosomal pathway proteins upon BAY-850 treatment in melanoma cell lines. These results are shown in **figure below**. Because we did not see any significant changes in their levels, we did not further pursue this line of inquiry.

Figure Legend: Measurement of changes in lysosomes and autophagy pathway markers in BAY-850 treated melanoma cells. The indicated melanoma cell lines were treated with either DMSO or 5 μ M BAY-850 for 48 h and indicated proteins were analyzed via immunoblotting. ACTINB was used as the loading control.

The strongest effects were observed in cell death and apoptotic pathways. Therefore, we focused our investigation on cell death pathways commonly enriched in both NRAS- and BRAF-mutant melanoma cells.

Among the top commonly upregulated pathways, we observed and validated ferroptosis as a significantly enriched cell death pathway in both BRAF- and NRAS-mutant melanoma following BAY-850 treatment (Please see Figure 5 and 6). The role of ATAD2 in regulating ferroptosis in cancer cells has never been explored and is novel. Additionally, previous studies have shown that ferroptosis induction in cancer cells is a fundamental mechanism that can limit tumor growth and promote cell death. Proper regulation of this pathway is essential for maintaining cellular homeostasis, and its dysregulation is a well-established hallmark of cancer. Moreover, ferroptosis has emerged as a promising therapeutic strategy in cancer treatment, with novel approaches being developed to activate this pathway for clinical benefit (Pu *et al*, 2022) . Our findings also suggest that BAY-850 induces ferroptosis, representing a potential new therapeutic avenue that could be exploited for the treatment of melanoma.

Referee #3:

This manuscript describes targeting of ATAD2 in BRAF/NRAS mutant melanoma. The authors demonstrate that ATAD2 expression is regulated by MAPK signalling through E2F1. This appears to be a general effect rather than only in MAPK pathway mutant cells. They show that treatment with an ATAD2 inhibitor BAY-850 results in complete loss of viability as measured by an MTT assay. They also show that ATAD2 is effective reducing tumour growth in xenograft models. This is interesting data and much of the remainder of the manuscript attempts to define a mechanism of action for BAY-850, defining the major pathways that influence this loss of viability in vitro and in vivo.

The authors present a lot of data to support their various pathways but there is a lot of over interpretation of the data presented. In the end, it is no clearer which pathways being influenced by BAY-850 are the critical ones, and the suggestion that pathways influenced are determined by the MAPK pathway mutant present in the melanoma have partial support at best. The real issue with the pathway analysis is that none of these mechanisms can explain the almost complete loss of viability with BAY in all cell lines shown in Figure 2A. If the authors do think that one of more of these pathways are responsible for the loss of viability, then they must show MTT assays when inhibiting these pathways that they can rescue the loss of viability. If the authors can focus their manuscript on the critical outcomes of this work, they will have an interesting and potentially important finding that could be translated into a clinical trial. My detailed points are below.

We thank the reviewer for acknowledging the importance and relevance of the data presented in our manuscript. We also thank the reviewer for his/her time and thorough review of our manuscript and providing detailed comments to further improve and strengthen our manuscript.

We fully agree that the tumor suppressive effects of ATAD2 inhibition are likely from multiple mechanisms, and that a single pathway alone cannot fully account for the growth inhibitory effects observed with ATAD2 inhibition.

In our study, we observed that BRAF-mutant cells undergo cellular senescence upon BAY-850 treatment, which was dependent upon the cyclin dependent kinase inhibitor CDKN1A expression (see **Figure 4C and 4E-I**). While we found that NRAS-mutant melanoma cell lines underwent apoptosis induction after treatment with BAY-850, which was dependent upon AMPK pathway (see **Figure 4J-K**). However, in the case of both BRAF- and NRAS-mutant melanoma cell lines, treatment with BAY-850 led to ferroptosis induction (see **Figures 5-6**).

These findings demonstrate that ATAD2 inhibition activates both subtype-specific (e.g., senescence in BRAF-mutant melanoma cells and apoptosis in NRAS-mutant melanoma) and common tumor suppressive pathways (ferroptosis in both NRAS- and BRAF-mutant melanoma), thereby causing melanoma growth inhibitory effects. Based on the question from the reviewer, we have improved the writing and presentation of our manuscript to overcome the ambiguity of this issue. We have also revised the Discussion section of our manuscript to elaborate on these findings.

In addition, as the reviewer will note, we have included several additional experiments to more precisely assess the impact of various pathways in mediating the effects of ATAD2 inhibition. We first investigated whether inhibition of CDKN1A expression using its inhibitor UC2288 (Wettersten *et al*, 2013) could rescue CDKN1A-dependent senescence induction upon BAY-850 treatment in BRAF-mutant melanoma cells. We observed that treatment with UC2288 led to a significant rescue of senescence induction in A375 cells upon treatment with BAY-850. This data is shown in **figure below** and has now been added in **Fig. EV3E** of the revised manuscript.

Figure Legend: Measurement of changes in senescence in cell treated with BAY-850 and p21 inhibitor UC2288. The indicated melanoma cell lines were treated with either DMSO or 5 µM BAY-850 alone or 5 µM UC2288 and 5 µM BAY-850 in combination for 48 h, and cellular senescence was measured quantitatively using senescence-β-galactosidase activity assay kit (fluorescence, plate-based Cat No# 25833; Cell Signaling Technology) following the manufacturer’s protocols and plotted. Relative cellular senescence in single and combination treatment condition with respect to DMSO control treated condition is plotted. All quantitative data represent the mean ± SEM. **p<0.01, ***p<0.0001, ****p<0.0001.

Next, we tested whether inhibition of AMPK signaling using Compound C (Lu *et al*, 2019) could rescue AMPK-dependent apoptosis induction following BAY-850 treatment in NRAS-mutant SKMEL-103 cells. We observed that Compound C treatment led to a significant rescue of apoptosis in SKMEL-103 cells. This data is shown in **figure below** and has now been added in **Fig. EV3G** of the revised manuscript.

Figure Legend: Measurement of changes in caspase 3 level in cell treated with BAY-850 and AMPK inhibitor Compound C. The indicated melanoma cell lines were treated with either

DMSO or 5 μ M BAY-850 alone or 0.1 μ M Compound C and 5 μ M BAY-850 in combination for 48 h, and caspase 3 activity was measured quantitatively using using Caspase 3 colorimetric assay kit (CASP3C, Sigma-Aldrich), following the manufacturer's protocols and plotted. All quantitative data represent the mean \pm SEM. *** p <0.001, **** p <0.0001.

In our study, we also demonstrate that in both NRAS- and BRAF-mutant melanoma cells, overexpression of GPX4 resulted in a significant rescue of ferroptosis induction and improved cell viability. Please see **figure below**. This data is also presented in **Fig. EV4E-G** of the revised manuscript.

Figure Legend: Measurement of changes in ROS level and viability in cell treated with BAY-850 with GPX4 overexpression. **A.** Indicated melanoma cell lines overexpressing either vector control or GPX4-ORF were immune-blotted for the shown proteins confirming GPX4 overexpression. **B.** Indicated melanoma cell lines overexpressing either vector control or GPX4-ORF were treated with BAY-850 at the shown concentration and cellular ROS levels was detected using the Cellular Reactive Oxygen Species Detection Assay Kit (ab186027, Abcam, Cambridge, UK), following the manufacturer's protocols. Relative ROS in BAY-850 or GPX4 overexpressing and BAY-850 treated condition with respect to DMSO control treated condition is plotted. **C.** Indicated melanoma cell lines overexpressing either vector control or GPX4-ORF were treated with BAY-850 at the shown concentration and cell viability was assessed by 3-(4,5-dimethylthiazol-2-yl)-2,5-diphenyltetrazolium bromide (MTT) assay. Cell survival is presented relative to the survival of DMSO-treated cells. All quantitative data represent the mean \pm SEM. ** p <0.01, *** p <0.001, **** p <0.0001.

These results demonstrate that the BAY-850 treatment in BRAF and NRAS mutant melanoma cells inhibits both distinct (senescence in BRAF mutant and apoptosis in NRAS mutant melanoma) and common pathways (ferroptosis).

Our point-by-point response to reviewer's additional questions are presented below:

Figure 1: the authors attempt to make a strong case that ATAD2 is over-expressed in melanoma. However, is the increase in expression compared to normal skin simply a function of more proliferation in tumour than normal tissue? The small data sets in B do not add significantly to the data in A. The data in panel C directly contradicts the larger TCGA dataset shows no effect of ATAD2 over expression on survival. <20% of melanoma patients over-express ATAD2, at least

in terms of mean expression in melanomas. The frequency of BRAF and NRAS mutations in ATAD2 over-expressing population appears to be about the same as in whole melanoma population. So I am not sure what the data presented here means.

This is an important question, and we understand the concern of the reviewer. Therefore, as a first step we asked if inhibition of proliferation reduces the level of ATAD2. To test this, we used Camptothecin that is known to inhibit cell proliferation (Rudolf *et al*, 2011; Rudolf *et al*, 2010). We found that ATAD2 expression did not change after Camptothecin treatment confirming that overexpression of ATAD2 is not dependent on proliferation. Please see **figure below**.

Figure Legend: Measurement of changes ATAD2 level in Camptothecin treated melanoma cells. The indicated melanoma cell lines were treated with either DMSO or Camptothecin (1μM for 24 hours) and indicated proteins were analyzed via immunoblotting. ACTINB was used as the loading control.

Additionally, we have already shown in our manuscript that ATAD2 is directly regulated by MAP kinase pathway and through the transcription factor E2F1, which further supports a direct role of MAP kinase signaling in stimulating the expression of ATAD2 (See Fig 1. D-L). It is also worth noting that melanoma cells show a general tendency for increased MAP kinase pathway due to other alterations beyond BRAF and NRAS mutations. Thus, MAP kinase mediated ATAD2 overexpression is still valid in the cells that show higher MAP kinase pathway activity. Furthermore, we do not claim that MAP kinase → E2F1 pathway is the sole regulator of ATAD2 in melanoma and other pathways and regulators may also be involved. We have made this point clear in Discussion section of the revised manuscript.

Furthermore, we would like to note that the experimental approach that we present here is a standard experimental approach that we have taken to showcase the overexpression of ATAD2 in melanoma and for establishing its mechanism of action. The same approach has been taken by multiple independent research groups in several previously published studies (Lee *et al*, 2024; Liu *et al*, 2022; Wang *et al*, 2023).

In regard to the data presented in panel C of Figure 1, it represents ATAD2 protein levels, not mRNA expression. If the reviewer is referring to mRNA data from TCGA, then the two datasets are not directly comparable and should not be interpreted as contradictory.

Promoted by the reviewer's comment, we analyzed the TCGA mRNA data by using top 25% expressors and bottom 25% expressors of ATAD2 and found that although as noted by the reviewer, not significant but overexpression of ATAD2 trended towards poor survival in patients. Please see **figure below**.

Figure legend: Effect of ATAD2 expression on patient survival- TCGA. Kaplan-Meier overall survival curve of TCGA SKCM patients belonging either to the patient's group with high and low ATAD2 mRNA expression.

Furthermore, a previously published study by Baggiolini et al. (Baggiolini *et al.*, 2021) identified ATAD2 as one of the top 25 epigenetic regulators in melanoma patient samples, with its expression strongly correlating with poor survival, which further support our data that ATAD2 overexpression show association with poor prognosis in melanoma. Please see figure below.

Figure legend: Effect of ATAD2 expression on patient survival- Baggiolini et al. (Science, 2021). Kaplan-Meier overall survival curve of TCGA SKCM patients belonging either to the patient's group with high levels of ATAD2 expression (ATAD2HI) or with low expression levels (ATAD2LO), log-rank p value reported. Figure taken from Baggiolini et al. (Science, 2021).

Thus, overall, our results and previously published studies support our findings that ATAD2 is overexpressed in melanoma and shows some association with poor survival. This information has been added in the revised manuscript.

2. The quality of the immunoblots is generally poor. Also, none are quantitated and no indication of the replicates or robustness of the data.

We apologize for the quality and have improved the presentation wherever we can in the revised manuscript. For example, we have repeated the Western blots shown in **Fig. 1E** and have included them in the revised version of the manuscript. We also show this in **Figure below (top)**.

To further strengthen the reliability of our findings, we have also performed ATAD2 immunoblots in triplicate to demonstrate the robustness of the data. Please see **figure below (bottom)**. While ATAD2 is the primary focus of this study, it is not feasible to present all Western blots in triplicate within the manuscript. However, the blots shown are representative of experiments that were performed multiple times to ensure consistency and reproducibility. It is important to note that Western blotting provides a semi-quantitative measure of protein expression, therefore, we have refrained from using this as a quantitative approach.

Figure legend: Effect of trametinib on ATAD2 expression in melanoma (replicates). The indicated melanoma cell lines were treated with either DMSO or 5 μ M BAY-850 for 24 and 48 h and indicated proteins were analyzed via immunoblotting. ACTINB was used as the loading control. Data for Figure 1E is shown at the top and data at the bottom shows the ATAD2 western blots in replicates. Replicate 3 is presented in Figure 1E.

3. The over-expression of E2F1 is again likely to simply reflect the higher proliferation in tumour versus normal tissue. In the CCLE dataset there is strong correlation between ATAD2 and E2F1 expression but this is not necessarily over-expression. It does support your data showing that E2F1 regulates ATAD2 expression.

Once again, similar to ATAD2, we show that E2F1 levels are not a direct effect of proliferation. Here, we asked if inhibition of proliferation reduces the level of E2F1. To test this, we treated melanoma cells with Camptothecin that is known to inhibit cell proliferation (Rudolf

et al., 2011; Rudolf *et al.*, 2010) and measured E2F1 levels. We found that E2F1 expression did not change after Camptothecin treatment confirming that overexpression of E2F1 is not dependent on proliferation. Please see figure below. We also cite several studies that have already established the E2F1 is overexpressed in melanoma and regulates its growth and proliferation (Alla *et al.*, 2010; Dar *et al.*, 2011; Liu *et al.*, 2025).

Figure Legend: Measurement of changes E2F1 level in Camptothecin treated melanoma cells. The indicated melanoma cell lines were treated with either DMSO or Camptothecin (1 μ M for 24 hours) and indicated proteins were analyzed via immunoblotting. ACTINB was used as the loading control. H2AX immunoblot shown here after Camptothecin treatment is used for both ATAD2 and E2F1 related questions asked by the reviewer.

Additionally, using both patient datasets and functional assays, we confirm that ATAD2 expression is directly dependent on E2F1. In our experiments, we show that E2F1 binds to ATAD2 promoter region, and its knockdown causes a reduction in ATAD2 mRNA and protein levels (see **Figure 1I-K**). These results further support the notion that E2F1 acts as a transcriptional regulator of ATAD2 in melanoma.

Furthermore, prompted by the reviewer, we have added the E2F1 and ATAD2 positive correlation data by analyzing the TCGA data in the revised manuscript (see **Fig. EV1E**). Please also see figure below. It is to be noted that this further supports our data. We thank the reviewer for bringing this to our attention.

Figure Legend: Correlation analysis for E2F1 transcription factor with ATAD2 indicating its potential role in stimulating ATAD2 transcription in melanoma. A correlation analysis between the mRNA levels of the E2F1 transcription factors and *ATAD2* mRNA was performed

using GEPIA on the TCGA dataset. Pearson correlation coefficient (R) and p-values for correlation analysis is shown.

4. The legend for Figure 1, G,H only one time point shown in these panels?

We have now included two time points for these experiments. Please see **figure below** and revised Figure 1 G, H.

Figure Legend: Effect of inhibition of MAPK pathway on E2F1 expression in melanoma. **A-B.** The indicated melanoma cell lines were treated with either DMSO or the MAPK inhibitor trametinib (200 nM) for 24 and 48 h. **(A)** *E2F1* mRNA expression was measured using RT-qPCR and plotted as the level in trametinib-treated cells relative to that in DMSO-treated cells. ACTINB was used as a normalization control. **(B)** E2F1 protein expression was measured using western blot analysis under the indicated conditions. ACTINB was used as a loading control. All quantitative data represent the mean \pm SEM. * $p < 0.05$, ** $p < 0.01$, *** $p < 0.001$

5. Figure 2; B, this is 2-4 weeks continuous exposure to 5 μ M BAY! And no quantitation. Is BAY unstable as panel A indicates complete loss of viability in 24 h yet B shows in A375 and SKMEL2 viable colonies?

Based on the reviewer's suggestion, we have now repeated this experiment shown in Figure 2B and present new data and including the quantification. Please refer to **Fig. EV2C** for the updated data. The results demonstrate a significant long-term survival effect on melanoma cells following BAY-850 treatment. For Figure 2B, cells were treated with fresh inhibitor every week to ensure continuous exposure over an extended period, better simulating clinical treatment conditions. Figure 2A represents a short-term viability assay, in which cells were treated once, and viability was measured after 72 hours. Together, these complementary assays highlight both the immediate and prolonged effects of BAY-850 on melanoma cell viability.

6. Figure 4, 5; These attempt to demonstrate that BRAF and NRAS mutant melanomas trigger different responses to BAY. However, none of these experiments are performed in other mutant background so there is no evidence that these mechanisms are indeed specific for the particular mutation.

We thank the reviewer for this thoughtful comment. While it is not feasible to perform all experiments across all genotypes within a single manuscript, we appreciate the importance of this point. We have noted this limitation in the Discussion section of our revised manuscript, and future studies will aim to further investigate the role of ATAD2 in other genotypes as well in greater detail.

7. Figure 5C; A lot of the common upregulated pathways appear to indicate nutritional stress, ferroptosis is a long way down this list, why was ferroptosis in particular investigated?

We agree that several pathways were found to be upregulated in our analysis. While we tested some of these pathways, such as autophagy and lysosomal pathway, they did not show consistent results in our secondary functional assays and were therefore not pursued further. Please see the figure below.

Figure Legend: Measurement of changes in lysosomes and autophagy pathway markers in BAY-850 treated melanoma cells. The indicated melanoma cell lines were treated with either DMSO or 5 μ M BAY-850 for 48 h and indicated proteins were analyzed via immunoblotting. ACTINB was used as the loading control.

Additionally, the role of ATAD2 in regulating nutritional stress pathways is well established. For example, ATAD2 has been shown to promote glycolysis and tumor progression in clear cell renal cell carcinoma by regulating the transcriptional activity of c-Myc ((Wu *et al*, 2023). In lung adenocarcinoma, ATAD2 expression has been reported to increase [18 F]fluorodeoxyglucose uptake via the AKT–GLUT1/HK2 pathway (Sun *et al*, 2019). Previous studies have shown that both AKT (Porstmann *et al*, 2005) and GLUT1/HK2 regulated glucose metabolism play a role in fatty acid metabolism and steroid synthesis (Parhofer, 2015; Xiao *et al*, 2022). Furthermore, under severe hypoxic conditions, proteolysis of ATAD2 has been shown to induce chemoresistance in cancer cells by inhibiting cell cycle progression during the S phase (Haitani *et al*, 2022). These studies collectively indicate that ATAD2 plays a role in regulating nutritional stress responses in cancer cells.

However, to date, no published studies have directly linked ATAD2 to ferroptosis. Our decision to investigate ferroptosis was motivated not only by the novelty of the approach but also by the compelling results obtained in our study. Further supporting this rationale are previous

findings demonstrating that ferroptosis is a fundamental mechanism that can limit tumor growth and promote cell death. Proper regulation of ferroptosis is essential for maintaining cellular homeostasis, and its dysregulation is a well-established hallmark of cancer. Moreover, ferroptosis has emerged as a promising therapeutic strategy, with novel approaches being developed to activate this pathway for clinical benefit. Our findings suggest that BAY-850 induces ferroptosis, representing a potential new therapeutic avenue for the treatment of melanoma. We have now included this rationale in the results and discussion section of the revised manuscript.

8. *There is no legend for Figure 6.*

We apologize for this oversight and have now included the relevant information in the revised manuscript.

9. *Figure 6F, this combination data must be shown in at least all the cell lines investigated in this work.*

As per reviewer's suggestion we tested the effect of combination treatment in other cell lines and have found similar results. Please see the **data below**. These results are also included in Figure 6F and **Fig. EV4H** of the revised manuscript.

Figure Legend: Co-targeting of ATAD2 with trametinib cause potent melanoma tumor growth inhibition. The indicated melanoma cell lines were treated with either DMSO or BAY-850 1 μ M alone or trametinib 0.25 nM alone or combination of BAY-850 1 μ M and trametinib 0.25 nM for 3 days, and survival was assessed by 3-(4,5-dimethylthiazol-2-yl)-2,5-diphenyltetrazolium bromide (MTT) assay. Cell survival is presented relative to the survival of DMSO-treated cells. All quantitative data represent the mean \pm SEM. * $p < 0.05$, ** $p < 0.01$, *** $p < 0.001$, **** $p < 0.0001$.

10. *The combination of Trametinib+BAY is additive in A375 and possibly synergistic in SKMEL103. But 2 cell lines is insufficient to assess this. The xenograft data is additive not synergistic which would be the prediction from the preceding data.*

We thank the reviewer for this valuable observation. We agree that the combination of trametinib

and BAY-850 appears additive in A375 and potentially synergistic in SKMEL-103 based on our data. However, as the reviewer correctly notes, the use of only two cell lines limits the generalizability of these findings. Furthermore, consistent with the reviewer's observation, the xenograft data support an additive rather than a synergistic effect, aligning more closely with the A375 profile. Nonetheless, the data showcase that the combination of trametinib+BAY-850 is more effective than either drug alone. We have revised the manuscript to describe the effect as additive in Discussion section of the revised manuscript.

11. In Figure 6, ROS levels have been equated to ferroptosis, and rather than measure cell killing the only measurement is ROS level. The GPX4 rescue experiment is only measuring ROS level rather than seeing if this rescues the loss of viability by BAY.

In our initial experiments, we used ROS levels as a marker for ferroptosis induction. However, in response to the reviewer's suggestion, we have now included additional data showing the rescue of cell viability alongside ROS measurements upon GPX4 overexpression in BAY-850 treated condition. These results are presented below and included in **Fig. EV4E-G** of the revised manuscript. These new experiments support our results that loss of ATAD2 promotes ROS-dependent ferroptosis induction and tumor growth inhibition via downregulation of GPX4. Overexpression of GPX4 significantly rescue both ROS dependent ferroptosis induction and tumor growth inhibition upon BAY-850 treatment in melanoma cells. This data is presented in **figure below**. We have also updated the manuscript to include these findings and believe this addition addresses the reviewer's concern.

Figure Legend: Measurement of changes in ROS level and viability in cell treated with BAY-850 with GPX4 overexpression. **A.** Indicated melanoma cell lines overexpressing either vector control or GPX4-ORF were immune-blotted for the shown proteins confirming GPX4 overexpression. **B.** Indicated melanoma cell lines overexpressing either vector control or GPX4-ORF were treated with BAY-850 at the shown concentration and cellular ROS levels was detected using the Cellular Reactive Oxygen Species Detection Assay Kit (ab186027, Abcam, Cambridge, UK), following the manufacturer's protocols. Relative ROS in BAY-850 or GPX4 overexpressing and BAY-850 treated condition is plotted with respect to DMSO control treated condition. **C.** Indicated melanoma cell lines overexpressing either vector control or GPX4-ORF were treated with BAY-850 at the shown concentration and cell viability was assessed by 3-(4,5-dimethylthiazol-2-yl)-2,5-diphenyltetrazolium bromide (MTT) assay. Cell survival is presented relative to the survival of DMSO-treated cells. All quantitative data represent the mean \pm SEM. ** $p < 0.01$, *** $p < 0.001$, **** $p < 0.0001$.

References

- Alla V, Engelmann D, Niemetz A, Pahnke J, Schmidt A, Kunz M, Emmrich S, Steder M, Koczan D, Putzer BM (2010) E2F1 in melanoma progression and metastasis. *J Natl Cancer Inst* 102: 127-133
- Baggiolini A, Callahan SJ, Montal E, Weiss JM, Trieu T, Tagore MM, Tischfield SE, Walsh RM, Suresh S, Fan Y *et al* (2021) Developmental chromatin programs determine oncogenic competence in melanoma. *Science* 373: eabc1048
- Bjorkoy G, Lamark T, Pankiv S, Overvatn A, Brech A, Johansen T (2009) Monitoring autophagic degradation of p62/SQSTM1. *Methods Enzymol* 452: 181-197
- Dar AA, Majid S, de Semir D, Nosrati M, Bezrookove V, Kashani-Sabet M (2011) miRNA-205 suppresses melanoma cell proliferation and induces senescence via regulation of E2F1 protein. *J Biol Chem* 286: 16606-16614
- Eskelinen EL (2006) Roles of LAMP-1 and LAMP-2 in lysosome biogenesis and autophagy. *Mol Aspects Med* 27: 495-502
- Haitani T, Kobayashi M, Koyasu S, Akamatsu S, Suwa T, Onodera Y, Nam JM, Nguyen PTL, Menju T, Date H *et al* (2022) Proteolysis of a histone acetyl reader, ATAD2, induces chemoresistance of cancer cells under severe hypoxia by inhibiting cell cycle progression in S phase. *Cancer Lett* 528: 76-84
- Karunaraj P, Scheele R, Wells ML, Rathod R, Abrahamson S, Taylor LC, Gokulu IS, Chowdhury L, Kazmi A, Song W *et al* (2025) A Hotspot Phosphorylation Site on SHP2 Drives Oncoprotein Activation and Drug Resistance. *bioRxiv*
- Konieczkowski DJ, Johannessen CM, Abudayyeh O, Kim JW, Cooper ZA, Piris A, Frederick DT, Barzily-Rokni M, Straussman R, Haq R *et al* (2014) A melanoma cell state distinction influences sensitivity to MAPK pathway inhibitors. *Cancer Discov* 4: 816-827
- Landsberg J, Kohlmeyer J, Renn M, Bald T, Rogava M, Cron M, Fatho M, Lennerz V, Wolfel T, Holzel M *et al* (2012) Melanomas resist T-cell therapy through inflammation-induced reversible dedifferentiation. *Nature* 490: 412-416
- Lee J, You C, Kwon G, Noh J, Lee K, Kim K, Kang K, Kang K (2024) Integration of epigenomic and transcriptomic profiling uncovers EZH2 target genes linked to cysteine metabolism in hepatocellular carcinoma. *Cell Death Dis* 15: 801
- Liu Y, Luo D, Lu Y, Tan L (2025) E2F transcription factor 1 as a potential prognostic biomarker and promotes tumor proliferation in skin cutaneous melanoma. *Pathol Res Pract* 269: 155875
- Liu ZG, Su J, Liu H, Yang XJ, Yang X, Wei Y, Zhu XY, Song Y, Zhao XC, Guo HL (2022) Comprehensive bioinformatics analysis of the E2F family in human clear cell renal cell carcinoma. *Oncol Lett* 24: 351
- Lu J, Meng Z, Cheng B, Liu M, Tao S, Guan S (2019) Apigenin reduces the excessive accumulation of lipids induced by palmitic acid via the AMPK signaling pathway in HepG2 cells. *Exp Ther Med* 18: 2965-2971
- Olender J, Wang BD, Ching T, Garmire LX, Garofano K, Ji Y, Knox T, Latham P, Nguyen K, Rhim J *et al* (2019) A Novel FGFR3 Splice Variant Preferentially Expressed in African American Prostate Cancer Drives Aggressive Phenotypes and Docetaxel Resistance. *Mol Cancer Res* 17: 2115-2125
- Parhofer KG (2015) Interaction between Glucose and Lipid Metabolism: More than Diabetic Dyslipidemia. *Diabetes Metab J* 39: 353-362

Porstmann T, Griffiths B, Chung YL, Delpuech O, Griffiths JR, Downward J, Schulze A (2005) PKB/Akt induces transcription of enzymes involved in cholesterol and fatty acid biosynthesis via activation of SREBP. *Oncogene* 24: 6465-6481

Pozniak J, Pedri D, Landeloos E, Van Herck Y, Antoranz A, Vanwynsberghe L, Nowosad A, Roda N, Makhzami S, Bervoets G *et al* (2024) A TCF4-dependent gene regulatory network confers resistance to immunotherapy in melanoma. *Cell* 187: 166-183 e125

Pu F, Chen F, Zhang Z, Shi D, Zhong B, Lv X, Tucker AB, Fan J, Li AJ, Qin K *et al* (2022) Ferroptosis as a novel form of regulated cell death: Implications in the pathogenesis, oncometabolism and treatment of human cancer. *Genes Dis* 9: 347-357

Rambow F, Rogiers A, Marin-Bejar O, Aibar S, Femel J, Dewaele M, Karras P, Brown D, Chang YH, Debiec-Rychter M *et al* (2018) Toward Minimal Residual Disease-Directed Therapy in Melanoma. *Cell* 174: 843-855 e819

Rudolf E, Rudolf K, Cervinka M (2011) Camptothecin induces p53-dependent and -independent apoptogenic signaling in melanoma cells. *Apoptosis* 16: 1165-1176

Rudolf K, Cervinka M, Rudolf E (2010) Dual inhibition of topoisomerases enhances apoptosis in melanoma cells. *Neoplasia* 57: 316-324

Skoniecka A, Cichorek M, Tyminska A, Pelikant-Malecka I, Dziewiatkowski J (2021) Melanization as unfavorable factor in amelanotic melanoma cell biology. *Protoplasma* 258: 935-948

Sun T, Du B, Diao Y, Li X, Chen S, Li Y (2019) ATAD2 expression increases [18F]Fluorodeoxyglucose uptake value in lung adenocarcinoma via AKT-GLUT1/HK2 pathway. *BMB Rep* 52: 457-462

Thier B, Zhao F, Stupia S, Bruggemann A, Koch J, Schulze N, Horn S, Coch C, Hartmann G, Sucker A *et al* (2022) Innate immune receptor signaling induces transient melanoma dedifferentiation while preserving immunogenicity. *J Immunother Cancer* 10

Tiago M, de Oliveira EM, Brohem CA, Pennacchi PC, Paes RD, Haga RB, Campa A, de Moraes Barros SB, Smalley KS, Maria-Engler SS (2014) Fibroblasts protect melanoma cells from the cytotoxic effects of doxorubicin. *Tissue Eng Part A* 20: 2412-2421

Tomuleasa C, Tigu AB, Munteanu R, Moldovan CS, Kegyes D, Onaciu A, Gulei D, Ghiaur G, Einsele H, Croce CM (2024) Therapeutic advances of targeting receptor tyrosine kinases in cancer. *Signal Transduct Target Ther* 9: 201

Wang Q, Bode AM, Zhang T (2023) Targeting CDK1 in cancer: mechanisms and implications. *NPJ Precis Oncol* 7: 58

Wettersten HI, Hee Hwang S, Li C, Shiu EY, Wecksler AT, Hammock BD, Weiss RH (2013) A novel p21 attenuator which is structurally related to sorafenib. *Cancer Biol Ther* 14: 278-285

Wu Z, Ge L, Song Y, Deng S, Duan P, Du T, Wu Y, Zhang Z, Hou X, Ma L *et al* (2023) ATAD2 promotes glycolysis and tumor progression in clear cell renal cell carcinoma by regulating the transcriptional activity of c-Myc. *Discov Oncol* 14: 79

Xiao X, Luo Y, Peng D (2022) Updated Understanding of the Crosstalk Between Glucose/Insulin and Cholesterol Metabolism. *Front Cardiovasc Med* 9: 879355

Yadav V, Zhang X, Liu J, Estrem S, Li S, Gong XQ, Buchanan S, Henry JR, Starling JJ, Peng SB (2012) Reactivation of mitogen-activated protein kinase (MAPK) pathway by FGF receptor 3 (FGFR3)/Ras mediates resistance to vemurafenib in human B-RAF V600E mutant melanoma. *J Biol Chem* 287: 28087-28098

Dear Romi,

Thank you for your patience while your revised manuscript was re-reviewed at EMBO reports. We have now received the full set of referee reports that is pasted below.

As you will see, while the referees overall support the publication of your work, both referees 1 and 2 also point out that the data suggesting that ATAD2 plays distinct roles in BRAF- and NRAS-mutant melanoma cells should be strengthened. Referee 2 also asks for a clear phenotypic characterization. I would like to encourage you to add more data along the lines suggested by the referees. Please let me know in case you would like to discuss this further. All remaining referee concerns will need to be addressed also in a detailed point-by-point response.

A few editorial requests will also need to be addressed:

- Please add up to 5 keywords to the ms file.
- The author credits need to be removed from the ms file. All credits need to be entered during online ms submission.
- In the author checklist, all questions regarding the statistics need to be answered. Please send us a new, completed checklist.
- The 6 EV tables that are uploaded all seem to be datasets and should be called such and updated in all places to Dataset EV1-EV6 (in the file names and ms callouts).
- Please correct EXPERIMENTAL METHODOLOGY to METHODS.

Figure Legends - Comments

- Please note that the exact p values are not provided in the legends of figures 1B, D, G, I, K; 2A, D, F, I; 3B, C, E, H, J; 4E, F, G, I, K; 5D, E; 6A, C, D, F, H, I, J; EV2 B, D, G, I; EV3 E-G; EV4 C, F, G, H. Please provide exact values as reasonable.
- Please indicate the statistical test used for data analysis in the legends of figures 1A, B, D, F, G, I, K; 2A, D, F, I; 3B, C, E, H, J; 4A-D; E, F, G, I, K; 5C, D, E; 6A, C, D, F, H, I, J; EV1 A; EV2 B, D, G, I; EV3 C, D, E-G; EV4 B, C, F, G, H
- Please note that the yellow dotted lines are not defined in the legend of figures 2E, J. This needs to be rectified.

I would like to suggest some minor changes to the abstract that needs to be written in present tense:

Melanoma is a highly metastatic form of skin cancer for which current therapies offer limited benefits. Here we show that the histone reader ATAD2 is overexpressed in melanoma and predicts poor prognosis, and that the MAP kinase pathway via the transcription factor E2F1 stimulates the expression of ATAD2. Genetic or pharmacological inhibition of ATAD2 suppresses the growth and metastasis of BRAF and NRAS mutant melanoma. Mechanistically, ATAD2 inhibition induces senescence in BRAF mutant melanoma while activates the AMPK pathway in NRAS mutant melanoma (** This part needs to be strengthened by more data **). Additionally, ATAD2 inhibition induces ferroptosis in both BRAF and NRAS mutant melanoma by downregulating the ferroptosis suppressor GPX4, and the ferroptosis inhibitor erastin blocks melanoma growth (** the last part of the sentence is confusing, as the opposite would be expected, please rewrite **). Combining the ATAD2 inhibitor BAY-850 with the MEK inhibitor trametinib enhances melanoma growth inhibition. Our data identify ATAD2 as a key driver of melanoma and provide a rationale for potentially targeting ATAD2 in conjunction with the MAPK pathway to treat melanoma.

EMBO press papers are accompanied online by A) a short (1-2 sentences) summary of the findings and their significance, B) 2-3 bullet points highlighting key results and C) a synopsis image that is exactly 550 pixels wide and 200-600 pixels high (the height is variable). The synopsis image should provide a sketch of the major findings, like a graphical abstract. Please note that text needs to be readable at the final size. Please send us this information along with the final manuscript.

I am looking forward to receiving a newly revised ms as soon as possible.

Kind regards,
Esther

Referee #1:

Overall, my previous comments have been addressed. I have only one remaining major point that should be considered to strengthen the conclusion that ATAD2 plays distinct roles depending on whether melanoma cells are BRAF- or NRAS-mutated (CDKN1A vs. AMPK inhibition). This is indeed very interesting, but to support such a general statement, it would be important to expand the panel of melanoma cell lines used in the study. If I am not mistaken, only 2 NRAS-mutated and 2 BRAF-mutated lines were included. At least 3 per group should be analyzed, and ideally a broader panel would be necessary to be able to make such a general conclusion.

Minor comments regarding the text:

On page 12, the transition between the BRAF-mutated and NRAS-mutated melanoma cell data is not clear. Why did the authors examine AMPK specifically? The connection between these sections should be made smoother.

The word "novel" is used frequently throughout the manuscript. I would avoid this term.

In the discussion, the role of ATAD2 in cellular differentiation is mentioned. It could be helpful to make this paragraph a bit clearer.

ATAD2 overexpression was indeed previously shown to drive dedifferentiation and acquisition of a malignant state by melanocytes that were not yet melanoma cells. This happened by aberrant re-expression of a NC-related signature and the activation of MYC-related pathways. The new data presented here suggest that ATAD2 inhibition does not affect the differentiation state of an already malignant cell.

Finally, the earlier study did not demonstrate that ATAD2 directly regulates SOX10 expression, but rather that ATAD2 enables SOX10 to bind neural crest-related genes.

Referee #2:

In the revised version, the authors have addressed several concerns regarding novelty and have better characterized the role of ATAD2 in melanoma. However, some important issues remain:

Although potential phenotypic changes are described in the text, no corresponding data are presented. This raises concerns about the relevance of these observations. A clear phenotypic characterization, along with the effect of BAY, is important, as current targeted therapeutic approaches show variable efficacy across different cell populations.

The section on metastasis does not appear relevant in this context. I recommend either removing it or moving it to the Supplementary Figures, as the observed effects are more likely attributable to reduced proliferation.

The data do not support a link between the genetic makeup of melanoma cells and distinct roles of ATAD2 in a convincing way. I would suggest being more careful on the interpretation and instead drawing more general conclusions about the versatility of different cell death mechanisms.

Before a final decision can be made, it would be advisable for the authors to address these concerns.

Referee #3:

The authors have attempted to address the comments and questions from my initial review. The manuscript is certainly stronger for the revisions made. I am not completely convinced by their arguments about the different mechanisms, but the data presented supports their concept broadly. The ChIP experiments are good evidence for the direct relationships suggested. In response to their efforts to demonstrate that inhibiting proliferation does not influence ATAD2 expression; the data presented in the rebuttal is not part of the manuscript and I appreciate their efforts to prove their case. However, camptothecin blocks cells in S/G2 phase checkpoint arrest induced in response to DNA damage. This is after the G1 phase E2F1-dependent gene expression regulation that is being indicated by the authors. The ChIP data is direct evidence of E2F1 regulating ATAD2 expression and provides strong support for the relationship being proposed here.

Editors comment

We thank the Editor for forwarding the comments from the reviewers and for requesting editorial corrections. We appreciate the acknowledgment that “the referees overall support the publication of your work.” Our point-by-point response to the editorial requests is provided below.

- Please add up to 5 keywords to the ms file.

As suggested, we have added 5 keywords to our manuscript. These are added below the abstract in the revised manuscript file.

- The author credits need to be removed from the manuscript file. All credits need to be entered during online manuscript submission.

We have removed credits of the authors from the manuscript and added it to online manuscript submission.

- In the author checklist, all questions regarding the statistics need to be answered. Please send us a new, completed checklist.

We have answered all question regarding statistics in author checklist and also provided this completed checklist.

- The 6 EV tables that are uploaded all seem to be datasets and should be called such and updated in all places to Dataset EV1-EV6 (in the file names and ms callouts).

As recommended, we have updated file name from EV tables to dataset EV. Please see revised manuscript.

- Please correct EXPERIMENTAL METHODOLOGY to METHODS.

We have now included this correction.

Please provide exact values as reasonable for Figure Legends and indicate the statistical test used for data analysis in the legends of figures.

We have now included exact values for each panel of a figure in Figure Legends and indicated the statistical test used for data analysis.

- Please note that the yellow dotted lines are not defined in the legend of figures 2E, J. This needs to be rectified.

We have now rectified the yellow lines in the images.

I would like to suggest some minor changes to the abstract that needs to be written in present tense:

As recommended, we have included the suggested changes and also written the abstract in present tense.

EMBO press papers are accompanied online by A) a short (1-2 sentences) summary of the findings and their significance, B) 2-3 bullet points highlighting key results and C) a synopsis image that is exactly 550 pixels wide and 200-600 pixels high (the height is variable). The synopsis image should provide a sketch of the major findings, like a graphical abstract. Please note that text needs to be readable at the final size. Please send us this information along with the final manuscript.

We will send the following information as suggested along with the manuscript-

A) a short (1-2 sentences) summary of the findings and their significance

As requested, we have added this. Please see below-

ATAD2 drives melanoma progression, and its inhibition induces ferroptosis to suppress tumor growth, supporting its potential as a therapeutic target for more effective melanoma treatment.

B) 2-3 bullet points highlighting key results

As requested, we have added this. Please see below-

- ATAD2 is overexpressed in melanoma, driven by the MAPK–E2F1 pathway, and is linked to poor prognosis.
- Inhibiting ATAD2 suppresses tumor growth and metastasis in melanoma by inducing ferroptosis via downregulation of GPX4.
- Combining ATAD2 inhibitor BAY-850 with MEK inhibitor trametinib enhances anti-tumor effects, supporting a dual-targeted therapeutic strategy.

C) a synopsis image that is exactly 550 pixels wide and 200-600 pixels high (the height is variable).

Reviewer's comment-

Referee #1:

Overall, my previous comments have been addressed. I have only one remaining major point that should be considered to strengthen the conclusion that ATAD2 plays distinct roles depending on whether melanoma cells are BRAF- or NRAS-mutated (CDKN1A vs. AMPK inhibition). This is indeed very interesting, but to support such a general statement, it would be important to expand the panel of melanoma cell lines used in the study. If I am not mistaken, only 2 NRAS-mutated and 2 BRAF-mutated lines were included. At least 3 per group should be analyzed, and ideally a broader panel would be necessary to be able to make such a general conclusion.

We thank the reviewer for noting that “Overall, my previous comments have been addressed”. The reviewer also noted that our findings that ATAD2 plays distinct roles depending on whether melanoma cells are BRAF- or NRAS-mutated (CDKN1A vs. AMPK inhibition) is interesting. We thank the reviewer for his/her kind comments and acknowledging our effort in addressing his/her previous comments.

The reviewer however noted “At least 3 per group should be analyzed, and ideally a broader panel would be necessary to be able to make such a general conclusion.”

As the reviewer will see we have spent a substantial effort and resources to address the original concerns of all three reviewers. We believe that, although interesting, a more thorough future study on genotype specific effects is required. This would be a substantial undertaking and would be suitable as an independent study of its own. We currently do not have resources to complete such a large set of studies, and I hope the reviewer appreciate our limitations and previous effort to address all the concerns.

However, we fully recognize the reservation of the reviewer in this regard and to address this issue, we have explicitly acknowledged this limitation in the Discussion section of this revised manuscript and noted that a more thorough study will be required to address this in a detailed manner.

We once again thank the reviewer for his/her time and effort in reviewing our manuscript, which because of the reviewer's suggestion has substantially improved.

Minor comments regarding the text:

On page 12, the transition between the BRAF-mutated and NRAS-mutated melanoma cell data is not clear. Why did the authors examine AMPK specifically? The connection between these sections should be made smoother.

We apologize for the lack of a smooth transition in the previous version of the manuscript. We have now revised the text to ensure a clearer and more cohesive transition between the BRAF-

mutated and NRAS-mutated melanoma cell lines. Additionally, we have clarified the rationale for investigating AMPK in the context of our study. Please see the revised manuscript.

The word "novel" is used frequently throughout the manuscript. I would avoid this term.

We agree with the reviewer's comment and have removed the word "novel" from the manuscript.

In the discussion, the role of ATAD2 in cellular differentiation is mentioned. It could be helpful to make this paragraph a bit clearer.

We thank the reviewer for this helpful suggestion. Based on the feedback, we have revised the paragraph in the Discussion section to improve clarity.

Referee #2:

In the revised version, the authors have addressed several concerns regarding novelty and have better characterized the role of ATAD2 in melanoma. However, some important issues remain:

We thank reviewer for noting that “the authors have addressed several concerns regarding novelty and have better characterized the role of ATAD2 in melanoma”. The reviewer also noted that some important issues remain and listed them below. Our point-by-point response to reviewers remaining issues are presented below:

Although potential phenotypic changes are described in the text, no corresponding data are presented. This raises concerns about the relevance of these observations. A clear phenotypic characterization, along with the effect of BAY, is important, as current targeted therapeutic approaches show variable efficacy across different cell populations.

It seems the major concerns of the reviewer for above is related to the type of cell death. We would like to mention that the central finding of our study is the induction of ferroptosis in melanoma cells upon ATAD2 inhibition. We have restructured the manuscript in a way to strengthen the overall impact and clarity of the study and to address reviewer 2 comments.

However, reviewer is correct in asserting that therapeutic approaches can show variable responses across different cell population and thus a more thorough future study on genotype specific effects is required. Based on this suggestion, we tested whether ATAD2 expression is associated with specific genotypic states or functional states. To this end, we analyzed three datasets, Rambow et al. (PMID: 30017245), Pozniak et al. (PMID: 38181739), and Karras et al. (PMID: 36131018) for ATAD2 expression patterns. Our analysis revealed that, in the Rambow et al. dataset, ATAD2 expression was significantly enriched with the invasive state compared to other states (NCSC, pigmented, and SMC). Similarly, in the Pozniak et al. dataset, ATAD2 expression was enriched in the mitotic state compared to other functional states. Finally, in the Karras et al. dataset, ATAD2 expression was enriched in neural crest (NC) and stem-like states compared to other cell states. These results are presented below and shown in the figure EV5 in this revised manuscript. Taken together, these findings support a role for ATAD2 in tumor growth, progression and treatment resistance.

Figure legend: Analysis of the Rambow et al., Pozniak et al., and Karras et al. datasets were performed to determine cellular states in which is associated with. N represents cell number. P-values were calculated using Kruskal-Wallis test.

The section on metastasis does not appear relevant in this context. I recommend either removing it or moving it to the Supplementary Figures, as the observed effects are more likely attributable to reduced proliferation.

We agree with this suggestion of the reviewer and accordingly have moved this result to the Supplementary Figure EV2 of the manuscript. Please see the revised figures in the updated manuscript.

The data do not support a link between the genetic makeup of melanoma cells and distinct roles of ATAD2 in a convincing way. I would suggest being more careful on the interpretation and instead drawing more general conclusions about the versatility of different cell death mechanisms.

Before a final decision can be made, it would be advisable for the authors to address these concerns.

We acknowledge the reservation of the reviewer about this data. Based on the feedback from the reviewer and related comments from other reviewers', we have now restrained our interpretation of these results and also reshuffled our manuscript to move some of this data to the supplementary figures. Promoted by this comment of the reviewer, we have also revised the Discussion section of our manuscript to address the limitation of our results related to genotype-specific mechanisms.

We hope the reviewer will find the responses satisfactory and can now support the publication of our manuscript.

Referee #3:

The authors have attempted to address the comments and questions from my initial review. The manuscript is certainly stronger for the revisions made. I am not completely convinced by their arguments about the different mechanisms, but the data presented supports their concept broadly. The ChIP experiments are good evidence for the direct relationships suggested. In response to their efforts to demonstrate that inhibiting proliferation does not influence ATAD2 expression; the data presented in the rebuttal is not part of the manuscript and I appreciate their efforts to prove their case. However, camptothecin blocks cells in S/G2 phase checkpoint arrest induced in response to DNA damage. This is after the G1 phase E2F1-dependent gene expression regulation that is being indicated by the authors. The ChIP data is direct evidence of E2F1 regulating ATAD2 expression and provides strong support for the relationship being proposed here.

We thank the reviewer for mentioning that “the authors have attempted to address the comments and questions from my initial review”. The reviewer also stated that “the manuscript is certainly stronger for the revision made”. The reviewer also liked our ChIP studies.

However, the reviewer noted that “I am not completely convinced by their arguments about the different mechanisms” but also noted that “but the data presented their concept broadly”

We have noted reservations of reviewer 3 and other reviewers in regard to genotype specific mechanisms associated with ATAD2. Therefore, we have reorganized our manuscript to largely focus on the common mechanism and moved the other data to supplementary section of the manuscript.

We believe that, although interesting, a more thorough future study on genotype specific effects is required. This would be a substantial undertaking and would be suitable as an independent study.

Guided by the reviewer, we have revised the discussion section of the manuscript to acknowledge this limitation of our study.

Dr. Romi Gupta
University of Alabama at Birmingham
Department of Biochemistry and Molecular Genetics
Birmingham, Alabama
United States

Dear Romi,

I am very pleased to accept your manuscript for publication in the next available issue of EMBO reports. Thank you for your contribution to our journal.
